

# Extreme Heat and Wildfire Emissions Enhance Volatile Organic Compounds: Insights on Future Climate

Christian Mark Salvador[1,#],  Jeffrey Wood[2], Emma Cochran[2], Hunter Seubert[2], Bella Kamplain[2], Sam Overby[2], Kevin Birdwell[1], Lianhong Gu[1], Melanie Mayes[1,#]

[1]*Environmental Sciences Division, Oak Ridge National Laboratory, Oak Ridge, TN, USA*
[2]*School of Natural Resources, University of Missouri, Columbia, MO, USA*

*Correspondence to:* Christian Mark Salvador (salvadorcg@ornl.gov) and Melanie Mayes (mayesma@ornl.gov)

**Abstract.** Climate extremes are projected to cause unprecedented deviations in the emission and transformation of volatile organic compounds (VOCs), which trigger feedback mechanisms that will impact the atmospheric oxidation and formation of aerosols and clouds. However, the response of VOCs to future conditions such as extreme heat and wildfire events is still uncertain. This study explored the modification of the mixing ratio and distribution of several anthropogenic and biogenic VOCs in a temperate oak–hickory–juniper forest as a response to increased temperature and transported biomass burning plumes. A chemical ionization mass spectrometer was deployed on a tower at a height of 32 m in rural central Missouri, United States, for the continuous and in situ measurement of VOCs from June to August of 2023. The maximum observed temperature in the region was 38°C, and during multiple episodes the temperature remained above 32°C for several hours. Biogenic VOCs such as isoprene and monoterpene followed closely the temperature daily profile but at varying rates, whereas anthropogenic VOCs were insensitive to elevated temperature. During the measurement period, wildfire emissions were transported to the site and substantially increased the mixing ratios of acetonitrile and benzene, which are produced from burning of biomass. An in-depth analysis of the mass spectra revealed more than 250 minor compounds, such as formamide and methylglyoxal. The overall volatility, O:C, and H:C ratios of the extended list of VOCs responded to the changes in extreme heat and the presence of combustion plumes. Multivariate analysis also clustered the compounds into five factors, which highlighted the sources of the unaccounted-for VOCs. Overall, results here underscore the imminent effect of extreme heat and wildfire on VOC variability, which is important in understanding future interactions between climate and atmospheric chemistry.



## 1. Introduction

Future global climate, with continuing greenhouse gas emissions such as $CO_2$ from the burning of fossil fuels, is expected to have warmer temperatures that impact critical atmospheric processes. Global averaged surface air temperature is projected to exceed 1.5°C relative to 1850–1900 by the year 2030, regardless of the emission scenarios. Looking further to the future, 2081 to 2100 will experience an additional increase of 0.2°C–1.0°C and 2.4°C–4.8°C in low and high emissions scenarios, respectively (Lee et al., 2021). The heating of the atmosphere in the future will have severe effects on several atmospheric components and processes. For instance, a series of models have shown that warming due to greenhouse gas emissions will induce an increase in the global annual average mixing ratios of particles with less than 2.5 µm diameter ($PM_{2.5}$) (Park et al., 2020), which will have grave implications for air quality, climate, and human cardiovascular health. By 2050, the elevated temperature is projected to increase $PM_{2.5}$ by 2–3 µg m$^{-3}$ in the summer of the eastern United States as a consequence of faster oxidation rates and elevated production of organic aerosols (Shen et al., 2017). There is thus an urgent need to elucidate the impact of extreme heat on atmospheric processes, including the emission and transformation of organic compounds, to understand future aerosol-generating scenarios.

One potential effect of overall atmospheric warming is the alteration wildfire events' frequency around the globe (Varga et al., 2022; Sarris et al., 2014; Ruffault et al., 2018). At elevated temperatures, evaporation of soil moisture and generation of more fuel from drying vegetation are more pronounced, thus inducing more wildfire events. Beyond the $CO_2$ emissions, wildfires generate thousands of carbonaceous compounds that impact global climate air quality and human health (Schneider et al., 2024a). With the elevated prevalence of wildfires with prolonged duration, extreme wildfire events are expected to impact the future mixing ratio and distribution of atmospheric chemical compounds that influence relevant processes such as aerosol and cloud formation. For instance, global-scale airborne measurements showed increased tropospheric ozone in air masses influenced by biomass-burning (BB) events (Bourgeois et al., 2021). Long-term analysis of wildfire events in Western Canada (2001–2019) also indicated an increase in the average ozone mixing ratio (~2 ppb), particularly during events with high mixing ratios of atmospheric aerosols from combustion (Schneider et al., 2024b). Ozone enhancement will lead to elevated atmospheric oxidation capacity that can initiate more secondary pollutant formation.

Among the chemical components of the atmosphere, volatile organic compounds (VOCs) are expected to respond to extreme heat and wildfire emissions. VOCs, particularly the unsaturated compounds, interact with oxidants such as hydroxyl (OH) and nitrate ($NO_3$) radicals, which subsequently create ozone and oxidized molecules (Hakola et al., 2012; Ramasamy et al., 2016; Spirig et al., 2004; Vermeuel et al., 2023). Further reaction products such as highly oxidized molecules also participate in the formation of particles that subsequently act as cloud condensation nuclei (Chen et al., 2022; Hallquist et al., 2009). The emission and transformation of VOCs highly depend on environmental parameters such as temperature, relative humidity, and solar radiation, but the degree of changes under future climate is still uncertain (i.e., suppression or enhancement) (Daussy and Staudt, 2020). For instance, a global estimate of isoprene emissions with future temperature and land-cover drivers was 889 Tg yr$^{-1}$, substantially higher compared to



that expected using current climatological and land-cover conditions (522 Tg yr$^{-1}$) (Wiedinmyer et al., 2006).
However, $CO_2$, which is expected to rise in future climate, can substantially decrease the emission of isoprene from
vegetation (Lantz et al., 2019a). On the other hand, empirical results and modeling efforts suggest that future elevated
temperatures could suppress the impact of $CO_2$ on isoprene emissions, thus increasing the uncertainty of future
climate's influence on the emission of isoprene (Lantz et al., 2019b; Sahu et al., 2023). The complexity of the
interaction between abiotic factors of the future and the emission of VOCs should be fully understood to better predict
future air quality and climate scenarios.

In this work, we conducted a field campaign in the summer of 2023 to quantify the variability of VOCs over a
temperate oak–hickory–juniper (*Quercus–Carya–Juniperus*) forest in the Ozark Border Region of central Missouri.
The primary goal of the campaign was to examine the influence of temperature on VOCs. However, we were also able
to incorporate opportunistic analyses of smoke plumes that reached our site because of extreme wildfire activity in
Canada. We deployed a high-resolution chemical ionization mass spectrometer to continuously measure VOC
concentrations. The mass resolution of the technique (6000 m/Δm) provided an extended list of VOCs, beyond the
usual routinely evaluated compounds (e.g., methanol, isoprene, and monoterpene). The Ozark Plateau (Wiedinmyer
et al., 2005), and this site in particular, is a known hotspot for biogenic VOC (BVOC) emissions. Given these strong
emitters of BVOCs and the evident transport of anthropogenic VOCs (AVOCs) into the forest, the study area proved
to be a good test bed for measurement of the overall response of VOCs to abiotic stress in a way that simulates possible
future atmospheric conditions. The results presented here provide important information to assess possible future
feedback loops of vegetation and atmospheric chemistry to regional- and/or global-scale climate changes.



## 2. Experimental Designs

### 2.1 Site Description and Meteorological Data

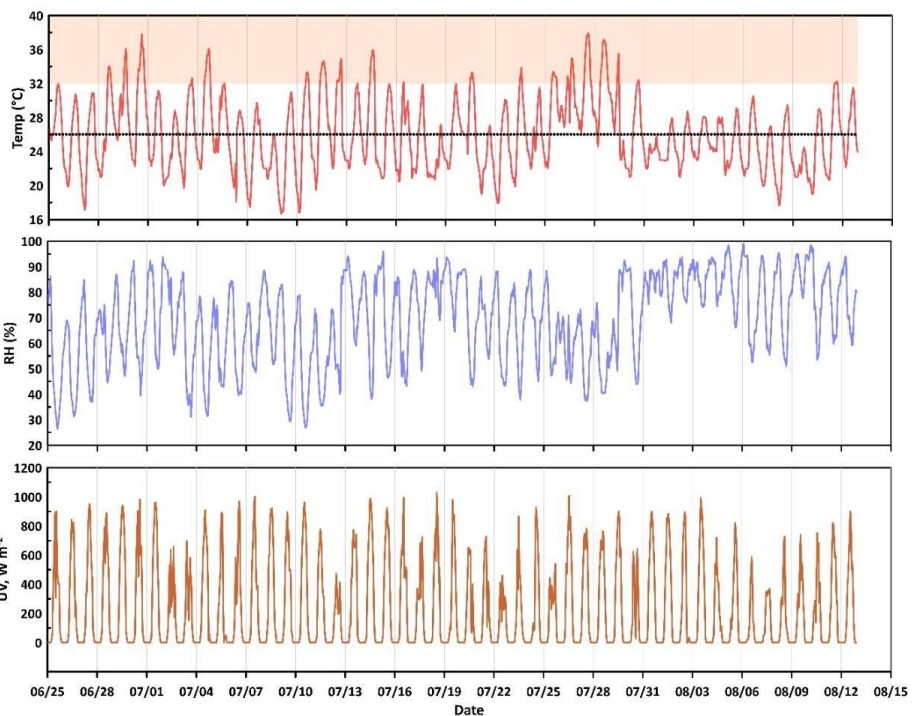

*Figure 1. Time series profile of (top) temperature, (mid) relative humidity, RH, and (bottom) global solar radiation, UV, at the temperate mixed deciduous forest in Missouri. The dotted line in the temperature plot is the average value during the measurement duration, and the shaded filled area denotes the extreme temperature conditions (>32°C).*

Measurements were conducted at the Missouri Ozark AmeriFlux (MOFLUX) site (latitude 38.7441, longitude −92.2000) in central Missouri, United States. The MOFLUX site is registered with the AmeriFlux (ID: US-MOz) and PhenoCam networks (ID: missouriozarks). The campaign was conducted during the summer of 2023, between June 25 and August 12. The site is situated in the Baskett Wildfire Research and Education Area. The primary sources of BVOCs were oaks (white and black), sugar maple, shagbark hickory, and eastern red cedar (Geron et al., 2016). The subtropical/mid-latitude continental characteristics of the area provide a warm and humid overall climate for the forest. Long-term measurements of meteorological parameters (1981–2010) at a nearby airport (~10 km) indicated that the average temperatures for January and July were −1°C and 25.2°C (National Climatic Data Center citation). Typical annual precipitation is fairly evenly distributed through the annual cycle and averages 1082 mm. More information regarding the site is provided elsewhere (Gu et al., 2015).

Figure 1 shows the time series profile of hourly averages of temperature and relative humidity collected from Columbia Regional Airport (38.817, −92.221), approximately 8.5 km from the MOFLUX site. Global solar radiation data were measured at a weather site in Ashland, MO (38.722, −92.253), 5.22 km from the MOFLUX tower. The data



were accessed using the MesoWest online website (https://mesowest.utah.edu/) provided by the Department of
Atmospheric Sciences, University of Utah. The average (absolute min-max) temperature, relative humidity (RH),
global solar radiation, and wind speed (not shown in the figure), were 26°C (16–38°C), 69.01% (26.43–99.02),
W m$^{-2}$ (0–1028 W m$^{-2}$), and 3.2 m s$^{-2}$ (0–11.27 m s$^{-2}$) during the time of VOC measurements. The diurnal profiles
of the meteorological conditions are provided in the supplement. During the weeks of July 4 and July 11, 64 and 100%
cumulative percent area reported abnormally dry conditions (D0, US Drought Monitor Category). Drought data were
accessed from the U.S. Drought Monitor (https://droughtmonitor.unl.edu/). Drought is a critical event at MOFLUX,
as such environmental stress induced the highest ecosystem isoprene emission ever recorded for a temperate forest in
2011 (53.3 mg m$^{-2}$ h$^{-1}$) (Potosnak et al., 2014). Smoke mixing ratios (in mg m$^{-3}$) were estimated from the High-
Resolution Rapid Refresh (HRRR) 3 km weather model for Missouri at 6 hour intervals for the duration of the VOC
data measurement period. Values ranged from 0 to 10 mg m$^{-3}$ during 80% of the measurement dates (overall average
was 7.33 mg m$^{-3}$) but reached a maximum of 175 mg m$^{-3}$ on July 16 in association with drift from large Canadian
wildfires.
**2.2 VOC Measurement and Identification**
VOCs were measured using a proton transfer reaction time of flight mass spectrometer (PTR-ToF-MS 6000 X2)
(Ionicon Analytik Ges.m.b.H., Innsbruck, Austria). A detailed description of the general mechanism of the PTR-ToF-
MS can be found elsewhere (Yuan et al., 2017). Briefly, hydronium ions are utilized to charge the VOCs through a
non-dissociative proton transfer in the reaction chamber of the instrument. This technique can identify a wide range
of compounds (e.g., carboxylic acids, carbonyls, and aromatic hydrocarbons) if the target compound has a proton
affinity higher than water (691 kJ/mol). The protonation occurs as follows:

$$H_3O^+ + VOC \rightarrow H_2O + VOC{-}H^+ \quad (1)$$

The PTR-ToF-MS was calibrated regularly using a 110 ppb mixture of gases (isoprene, limonene, benzene, toluene,
ethylbenzene, dichlorobenzene, trichlorobenzene, and trimethylbenzene, Restek Corp). The linear calibration curve
consisted of eleven data points, with mixing ratios ranging between 1.89 and 50.9 ppb. The same compounds were
used to calculate the mixing ratio of other compounds using the transmission efficiency and first-order kinetic reaction.
The PTR-ToF-MS was operated with 2.6 mbar and 80°C drift tube pressure and temperature, with an E/N value of
~119 Townsend. The mass range was set up to 500 m/z with a time resolution of 100 ms. The single spectrum time
was set to calculate the fluxes of the VOC, the results of which will be reported in subsequent works. One of the
limitations of the PTR-ToF-MS technique is that it cannot distinguish isomers (e.g., α-pinene, β-pinene, and limonene)
because of their identical exact mass (Blake et al., 2009). Instrument blank was measured hourly using a series of
switching valves and Ultra Zero grade air (Airgas).



Ambient air was sampled from the MOFLUX tower. The air was drawn at the top of the tower using a ½ in. OD PFA
tube (McMaster-Carr) and a GAST compressor/vacuum pump with a mass flow controller (Alicat Scientific, Inc) set
at 20 L min$^{-1}$.

High-resolution peak analysis, chemical formula identification, and data quantification were performed using the
IONICON data Analyzer (IDA). IDA identified more than 1000 ions, which were subsequently reduced to 275 peaks
with more than 5 parts per trillion (ppt) mixing ratios above the average blank data. Here, *mixing ratio* is defined as
the ratio of the moles of target analyte to the moles of all of atmospheric gases (i.e., $N_2$ and $O_2$). This can be expressed
as the following equation:
$$R_i = \frac{n_i}{n_\Sigma - n_i} \approx \frac{n_i}{n_\Sigma} \quad (2)$$
Where $R_i$ is the mixing ratio, $n_i$ is the moles of gas analyte, and $n_\Sigma$ is the total moles of atmospheric gases. The amount
of organic gases in the atmosphere is significantly lower than the total gases. The chemical identification procedure
was complemented by an analysis using ChemCalc, which also provided the theoretical masses and degree of
saturation (Patiny and Borel, 2013).
**2.3 Source and Process Signature Analysis of VOCs using Multivariate Analysis**
Determination of the source signature or emission profile of the VOCs is critical in assessing the dominant
anthropogenic and biogenic activities that impact the atmospheric reactivity from VOCs. Here, multivariate analysis
was applied to the observed VOC mixing ratios using non-negative matrix factorization (NNMF). Because NNMF
requires no uncertainty for the calculation procedure, it has an advantage over positive matrix factorization, which is
typically implemented for a mixture of organic compounds in the gas and particle phase (Salvador et al., 2022). NNMF
is expressed as
$$A_{m \times n} = W_{m \times k} H_{k \times n} + \sigma_{m \times n} \quad (3)$$

where $A$ is the input matrix with dimensions of $m$ and $n$ containing non-negative elements, $W$ and $H$ are species
fingerprint and coefficient matrices, $k$ is the lowest rank approximation or the optimal factor, and $\sigma$ is the residual
between the left and right sides of the equation. The VOC mixing ratio data with a matrix of $196 \times 274$ dimensions
was employed as the input for the NNMF routine program in MATLAB. The NNMF was applied for a 10-factor series
with 30 replicates, 1000 iterations, and a multiplicative update algorithm. The five-factor solution was the optimal
number used for the analysis based on the calculated root mean square of the residuals and the variability of the major
tracers across the factors.



## 3. Results and Discussion

### 3.1 General Overview of the Major VOCs

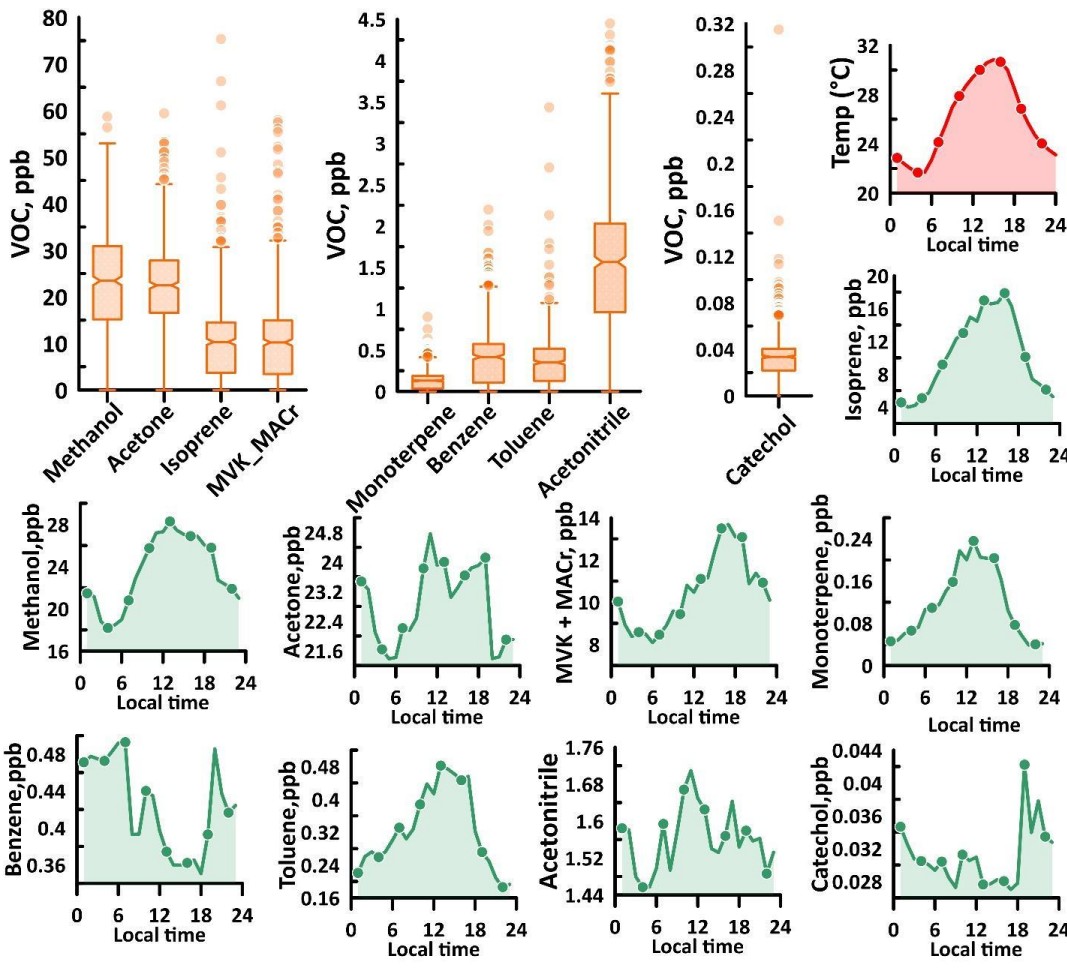

***Figure 2.*** *Average mixing ratio in ppb (top left) and diurnal profile of some of the major VOCs at MOFLUX. Time reported here is the local daylight time. The center lines of the box and whisker plots are the mean mixing ratio. Box edges are quartiles, and lower (upper) corresponds to 25th (75th) . Whiskers represent 1.5 times the interquartile range. Symbols outside the box plot are outliers. Diurnal profiles have a unit of ppb mixing ratio. MVK and MACr are methyl vinyl ketone and methacrolein. The average diurnal profile of temperature (top right) is also provided for reference.*

Several VOCs were detected in the ambient air throughout the three-month measurement period. Figure 2 shows the average mixing ratio of the dominant VOCs observed in the temperate forest. Among the VOCs, methanol and acetone recorded the highest mixing ratios. Methanol and acetone are the most abundant nonmethane organic gases in the troposphere and are emitted by terrestrial plants during growth stages (Bates et al., 2021; Hu et al., 2013; Wells et al., 2014). Mean mixing ratios of the methanol and acetone were 23 ppb, consistent with a prior study done in MOFLUX, in which half-hour averages of methanol ranged between 1.9 and 26 ppb (Seco et al., 2015). Here, the maximum



hourly average mixing ratio of methanol reached as high as 59 ppb, which occurred at 6:00 pm on the 30th of June.
Methanol also showed a diurnal profile with a daily peak at noon, which was an indication of a photochemical source.
Besides the terrestrial emissions of methanol, the secondary production of methanol from organic peroxyradicals (e.g.,
$CH_3O_2$) contributes substantially to the methanol budget (Bates et al., 2021).

Also shown in Figure 2 are the average mixing ratios of isoprene and its primary oxidation products, methyl vinyl
ketone and methacrolein (MVK+MACr). Isoprene is the most dominant BVOC, contributing around 50% to the total
global budget (Guenther et al., 2012). Isoprene substantially influences the surface ozone concentration and secondary
organic aerosol formation, which is attributed to isoprene's reactivity to ozone, OH, and nitrate ($NO_3$) radicals
(Wennberg et al., 2018). Besides the photochemical oxidation of isoprene, MVK and MACr have other sources, such
as BB and gasoline vehicular emissions (Ling et al., 2019). Isoprene also generates MVK during nighttime through
the dominant β-$RO_2$ isomer formation pathway (Ng et al., 2017). Isoprene's emission rate at MOFLUX was previously
reported as one of the highest for canopy-scale emissions (53.3 mg m$^{-2}$ h$^{-1}$) (Potosnak et al., 2014). This was evident
in our measurement, where the average mixing ratio of isoprene during the intensive observation period was 10.32
ppb, and MVK + MACr had a similar mean mixing ratio. The isoprene mixing ratio reached as much as 75 ppb, which
occurred at 1:00 pm on July 4. Observed isoprene mixing ratios were substantially elevated compared to other similar
temperate forests in the United Kingdom (~8 ppb) (Ferracci et al., 2020), deciduous forest in Michigan, USA (~1.5
ppb) (Kanawade et al., 2011), and mixed temperate forest in Canada (~0.01 ppb) (Fuentes and Wang, 1999). For
MVK+MACr, prior measurements in similar environments reported mixing ratios below 2.0 ppb (Safronov et al.,
2019; Shtabkin et al., 2019; Montzka et al., 1995) highlighting the intense production of MVK+MACr at MOFLUX.
Interestingly, the most elevated mixing ratio of MVK+MACr (58 ppb) occurred on a different day (6/28) and later in
the night (8:00 pm); this result was attributed to other sources of MVK and MACr. Nevertheless, the MVK showed a
similar diurnal profile with isoprene, which suggested that photochemical oxidation of isoprene was the dominant
source of MVK+MACr observed in MOFLUX. Also, diurnal profiles, as indicated in Figure 2, showed that
MVK+MACr still persisted even with the reduction of the isoprene at nighttime. This was attributed to the longer
atmospheric lifetime and lesser reactivity of MVK+MACr.

Monoterpene, a critical contributor to ozone and secondary aerosol formation (Salvador, 2020; Salvador et al., 2020b),
is composed of several organic species such as α-pinene, β-pinene, limonene, δ-carene, ocimene, and sabinene, and
its distribution varies significantly based on the vegetation species. At MOFLUX, monoterpenes had an average
mixing ratio of less than 0.2 ppb, as shown in Figure 2. Throughout the measurement duration, the maximum mixing
ratio of monoterpene was 0.9 ppb. This ambient level is similar to a prior measurement at the MOFLUX site (Seco et
al., 2015), as well as observations of monoterpene in other temperate forests in Wisconsin, USA, and Wakayama,
Japan (Vermeuel et al., 2023; Ramasamy et al., 2016). Interestingly, the diurnal profile of monoterpene at MOFLUX
had a daytime peak, which is not typical compared to other observations of monoterpene with nighttime enhancements
(Gentner et al., 2014; Stewart et al., 2021; Salvador et al., 2020a). Regions dominated by emissions of α-pinene, β-
pinene, and limonene typically have a nighttime peak, whereas daytime enhancements are observed for areas with



sabinene and ocimene (Hakola et al., 2012; Gentner et al., 2014; Jardine et al., 2015; Borsdorf et al., 2023). Either of
the latter two VOCs or a combination thereof might be the main monoterpenes impacting the chemical reactivity at
MOFLUX leading to aerosol formation in the forest. Particle size distribution analysis (see Figure S2) at our temperate
forest indicated no evident particle formation events. Relatively large particles (i.e., particle diameter . 50 nm) were
observed with no apparent aerosol growth. Prior study also showed less frequent new particle formation events,
particularly during the influence of southerly air masses rich in BVOCs (Yu et al., 2014). The most probable reason
for the presence of these large particles was the isoprene-rich condition of the temperate forest that impacted the
aerosol nucleation, even with enough monoterpene and ozone available for particle formation. Prior plant chamber
analysis indicated that the suppression of new particle formation was dependent on the ratio of isoprene carbon to
monoterpene carbon (Kiendler-Scharr et al., 2009). The mixing of isoprene and monoterpene also impacts the
atmospheric oxidation capacity, in which isoprene scavenges the OH radicals (Mcfiggans et al., 2019). Recent studies
also showed that the mixing of isoprene to monoterpene reduced $C_{20}$ dimers that drive aerosol formation at mixed
biogenic precursor systems (Heinritzi et al., 2020). At MOFLUX, the median ratio of isoprene carbon to monoterpene
carbon was 42, which is significantly higher compared to measurements in forests in Alabama (Lee et al., 2016),
Michigan (Kanawade et al., 2011), the Amazon (Greenberg et al., 2004), and Finland (Spirig et al., 2004). Ratios
above 20 completely limit the formation of aerosols, which is consistent with the observations at MOFLUX.

Besides biogenic VOCs, several anthropogenic-related VOCs were detected in the temperate forest. The site is about
5 km away from a major highway, which possibly contributed to the diversity of VOCs at MOFLUX. During the
measurement period, benzene, a VOC usually emitted from automobile exhausts, had a mean mixing ratio of
0.42 ppb, with a maximum of 2.2 ppb. Benzene had mixing ratio peaks consistent with the traffic (8:00 and 20:00)
with no evident noontime peak. Similar to biogenic precursors, benzene can also initiate particle formation events,
particularly at low NOx conditions (Ng et al., 2007; Li et al., 2016). The mixing of the biogenic (e.g., isoprene and
monoterpene) and anthropogenic VOCs (e.g., benzene) at MOFLUX can introduce unaccounted-for molecular
interactions (Voliotis et al., 2021) that can influence the formation of aerosols in the forest. Toluene, another important
aromatic VOC from urban emissions, was also observed at a significant amount at the site (~0.3 ppb, mean) with a
max mixing ratio of 3.4 ppb. The noontime peak of the toluene daily cycle was unexpected because it usually tracks
with traffic conditions. Interference of para-Cymene fragmentation in the drift tube of the PTR-ToF-MS at mass 93
(Ambrose et al., 2010) might have impacted the observed concentrations at MOFLUX.

Typical gas phase BB tracers were also observed in substantial amounts in MOFLUX. Acetonitrile, one of the
prominent BB markers (Huangfu et al., 2021), had mean and maximum mixing ratios of 1.56 ppb and 4.45 ppb,
respectively.  Such values are beyond the mixing ratio range (0.047 to 1.08 ppb) of acetonitrile recorded in Asian, US,
and European regions (Huangfu et al., 2021), highlighting the severe impact of BB in the atmospheric VOC
distribution and reactivity of MOFLUX. Acetonitrile did not follow a typical daily cycle, which is consistent with the
sporadic nature of the emissions and transport. Another prominent BB marker measured at the site was catechol, an
aromatic compound directly emitted from combustion processes. At MOFLUX, catechol had a mean level of 30 ppt




but increased significantly to 300 ppt on some days. Catechol had a minor peak during the daytime, which can be
attributed to the photochemical processing of phenol (Finewax et al., 2018), another aromatic VOC emitted during
BB events.
**3.2 Impact of Extreme Temperatures on VOCs**
During some parts of the measurement period, mid-Missouri experienced extreme temperature conditions
that impacted the physiochemical processes of the vegetation and the atmosphere. During the measurement period,
the average temperature was 26°C, and the highest hourly value was 38°C. The average temperature was close to the
reported long-term mean temperature in the region; however, the period of measurement exhibited extreme
temperatures that impacted VOC emissions. Diurnal profile temperature showed a daily peak occurring at 15:00,
which typically had a 29.9°C mean temperature. The extreme temperature, defined by an hourly mean temperature
over 32 °C, was based on the projected climate scenarios that temperature will increase by 2–4°C by 2100 (Collins et
al., 2013). The extreme temperature occurred for more than 100 hours (see Figure S1 for histogram). The strong impact
of the elevated temperature in the region ultimately altered the vegetation's physiological functions.

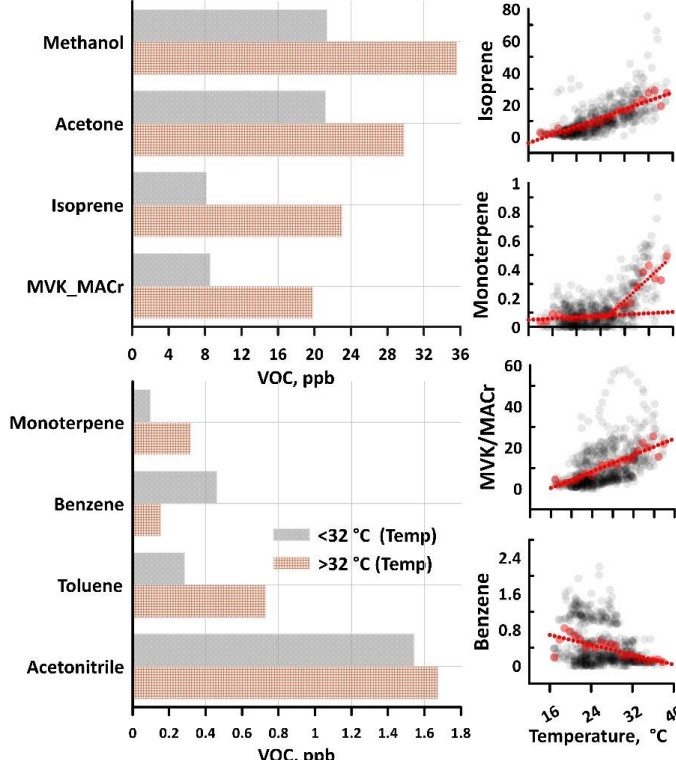


*Figure 3. (Left) Comparison of VOC mixing ratios for temperatures below and above 32°C. Catechol, not shown here, showed no*
*evident difference between the two conditions (~30 ppt). (Right) The correlation analysis of temperature with biogenic VOCs and*
*benzene mixing ratios (in ppb). Correlation analysis of other major VOCs is provided in the supplement. Black symbols are hourly*
*data, whereas the red lines indicate the best-fit line of the binned mixing ratio of VOCs according to 1.0°C of temperature. Note*
*that monoterpene has two best fit lines that showcase the response of monoterpene at different temperature regimes.*





BVOCs have different responses to elevated temperatures. Isoprene, for instance, was observed to follow (r = 0.95)
closely the temperature profile in the region (Figure 3). Linear regression of the temperature and isoprene indicated
that an increase of 1.0°C results in 1.32 ppb of isoprene. Moreover, the isoprene mean mixing ratio at elevated
temperatures was 23 ppb, which was thrice compared to conditions below 32°C. Monoterpene at MOFLUX also
showcased a complex response to temperature. Below 27.8°C, the monoterpene was insensitive to the temperature
(0.0023 ppb/°C, r = 0.42) but showed a direct response at enhanced temperatures (0.0392 ppb/°C, r = 0.92). The ten-
fold increase in the dependence of monoterpenes on extreme temperature had several implications for the distribution
and chemical reactivity in the forest. The non-linear pattern of monoterpene was consistent with the profile of ocimene
and sabinene when exposed to a temperature range between 28 and 40°C (Jardine et al., 2017), supporting the initial
assessment of the possible dominant monoterpenes at MOFLUX. However, we are not discounting the potential
contribution of monoterpenes (e.g., limonene) that are insensitive to changes in temperature. Moreover, the non-linear
response of monoterpene to temperature also impacted the aerosol formation events at MOFLUX. A normal
distribution of the average ratio of isoprene carbon to monoterpene carbon binned per 1.0°C was calculated (see Figure
S4). Even though the values exceeded ratios in which aerosol formation is suppressed, it was interesting that an
optimum temperature existed at which the distribution of BVOCs would result in a maximum inhibition of aerosol
formation.

MVK and MACr produced from the oxidation of isoprene showed a strong association with temperature (1.0 ppb/°C,
r = 0.95). MVK and MACr reached a 20 ppb average mixing ratio during extreme temperature conditions. This result
is twice the ratio at low temperature in the forest, similar to the observed pattern with isoprene. This was consistent
with a previous chamber study, which showed that the observed yields of MVK+MACr increased to 17–22% at
enhanced temperatures (70°C), compared to 9–11% at 30°C. Several possible causes can be attributed to such
observations. First, the higher mixing ratio of precursor isoprene yielded more MVK+MACr in the atmosphere.
Moreover, several of the reaction mechanisms during the oxidation of isoprene are temperature dependent (e.g., 1,6-
and 1,5-H shift isomerization reactions of isoprene), which further augmented the formation of the first-generation
products of isoprene (Navarro et al., 2013).

Anthropogenic tracers generated from transportation and BB showed little to no dependence on temperature. For
benzene (−0.027 ppb/°C) and xylene (−0.0069 ppb/°C), an indirect relationship with temperature was recorded. Such
results were attributed to the reduction of the height of the primary boundary layer at nighttime, which enhanced the
mixing ratio of such AVOCs. With colder temperatures during nighttime, a negative correlation between temperature
and AVOCs was expected. Remarkably, the toluene mixing ratio (0.73 ppb) doubled at higher temperatures, unlike
the benzene and xylene. This result further affirmed the initial claim that the compound occurring at mass 93 stemmed
from the fragmentation of a monoterpene with direct association with temperature. Combustion markers such as
acetonitrile (r = 0.53) and catechol (r = 0.017) also did not follow the trend of temperature, which is consistent with
the infrequent emissions of BB plumes.





Overall, extreme temperature conditions had a mixed impact on the VOCs observed in the temperate forest. Urban
and combustion markers showed insensitivity to temperature variation. On the other hand, BVOCs such as isoprene,
MVK+MACr, and monoterpene showed linear responses but at varying rates. The alteration of VOC distribution due
to enhanced temperature has imminent implications on the formation of secondary aerosols, particularly under future
climate with expected elevated temperatures. Recent laboratory chamber studies have shown that unexpected
interaction of individual VOCs during the oxidation process produced intermediates and products that impacted the
yields, volatility, and other physiochemical properties of aerosols (Voliotis et al., 2021; Takeuchi et al., 2022; Chen
et al., 2022). This has a serious impact on the projection of secondary aerosol formation in the future, considering the
cross-reactions between intermediate products from different VOCs are not yet accounted for in secondary aerosol
simulations in regional and global climate models. Based on the results here, isoprene at MOFLUX is expected to
increase more as the temperature increases compared to monoterpene. Thus, careful consideration of the oxidant
chemistry and product speciation will provide valuable new insights into the impact feedback loop between aerosols
and climate in temperate forests.
**3.3 Transport of Emissions from Forest Fires**
In 2023, severe wildfires that were initiated by summer lightning storms occurred over several boreal forests
in Canada, which resulted in burning of more than 156,000 km$^2$ of cumulative area that accounted for at least 1.7% of
Canada's land area (Wang et al., 2023). Between May and September of 2023, carbon emissions from fires reached
more than 638 Tg C based on satellite observations (Byrne et al., 2023). Two air pollution episodes (June 24 to
July 1 and July 12 to 19) resulting from these wildfires affected the field measurement at MOFLUX. Figure 4 shows
the smoke concentration measured at MOFLUX. The two pollution episodes had different levels of smoke, the second
period having stronger enhancements compared to the first. Wildfire emissions during the first episode were
substantially transported to Europe, whereas the second impacted the USA to a considerable extent (Wang et al.,
2023). A wildfire that occurred between July 12 and 19 primarily near Fort Nelson, Northwest Canada, was transported
to the MOFLUX site. Back trajectory analysis (see Figure 4) indicated that the plumes arriving at the site during the
same period originated from the northwest, suggesting a significant long-range transport of combustion products to
the MOFLUX temperate forest. Atmospheric dispersion of the smoke in Missouri is presented in Figure S5.

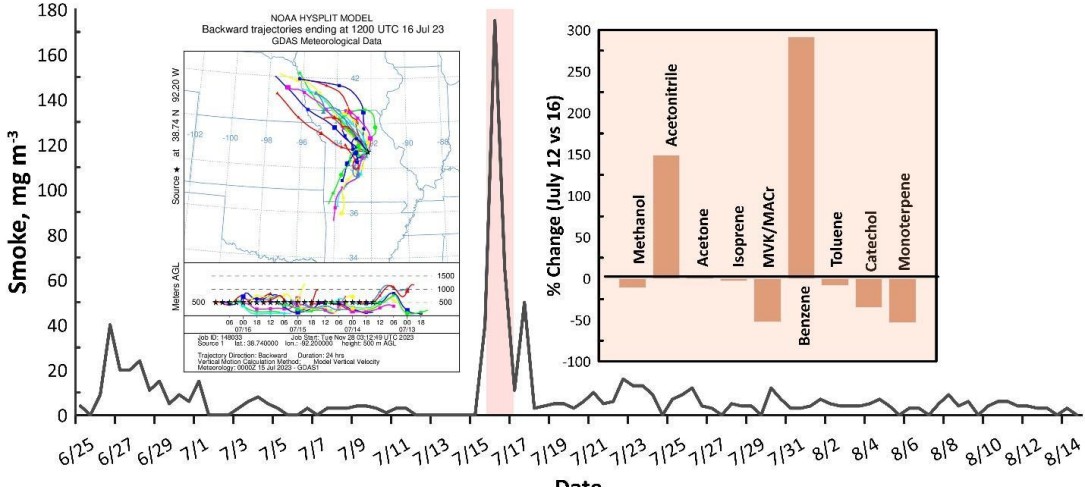

**Figure 4.** *Smoke profile observed during the field measurement. The red highlighted area is the period with intense transport of*
*BB plumes. (Inset Left) Backward air parcel trajectory analysis of plumes arriving on the 16th of July was calculated using the*
*Hybrid Single-Particle Lagrangian Integrated Trajectory (HYSPLIT) Model (Stein et al., 2015). (Inset Right) Percent change of*
*the mixing ratios of major VOCs measured during days with no combustion event (July 12) and with significant transport of BB*
*markers (July 16).*
Among the major VOCs, acetonitrile and benzene appeared to be associated with the transport of the combustion
plumes. Figure 4 shows a comparison of the VOC mixing ratios during the impact of combustion plume (July 16) and
non-BB event day (July 12), together with smoke mixing ratio observed at MOFLUX. These two VOCs had day
average mixing ratios of 2.15 (acetonitrile) and 0.34 (benzene) ppb, corresponding to increases of 139% and 269%,
respectively, compared to non-BB days. The source of benzene shifted from transport emissions to BB, highlighting
the diverse anthropogenic activities impacting the variability of benzene in temperate forests. Interestingly,
unsaturated BB markers like benzene can contribute to enhancement of atmospheric ozone levels (Bourgeois et al.,
2021). The ozone forming potential (OFP) of benzene increased to 0.421 ppb during the transport of wildfire
emissions, compared to 0.107 ppb observed on July 12, highlighting the influence of transported combustion plumes
on the overall chemical reactivity in the forest (see Text S1 for the calculation of OFP).
**3.4 Expanded List of VOCs and Their Response to Enhanced Temperature and Long-Range Transport of**
**Combustion Emissions**
Beyond the major VOCs discussed above, more than 250 compounds with a mass to charge ratio (m/z) of at least 40
and with mixing ratios 5 ppt above the blank measurements were identified. The proposed molecular ion formulas are
listed in the Table S1. The compounds have a wide variety of molecular compositions, with 14, 23, 5, 3, and 2 max
numbers of carbon, hydrogen, oxygen, nitrogen, and sulfur, respectively, with a median formula of $C_4H_6O$ and a mode
of 2 degrees of unsaturation. The numbers of VOCs according to atomic content were as follows: 36 $C_xH_y$, 93 $C_xH_yO_w$,
17 $C_xH_yN_z$, 60, $C_xH_yO_wN_z$, and 10 $C_xH_yO_wS_v$, where v, y, x, y, and z are positive integers. The median oxygen-to-
carbon ratio (O:C) and hydrogen-to-carbon ratio (H:C) of all the identified ions were 0.2 and 1.4. The O:C ratio was
similar to a measurement in a boreal forest in southern Finland, which can be explained by the similarity of the



measurement technique applied (i.e., Vocus PTR-ToF-MS), capturing fewer oxygenated compounds compared to
other ionization techniques (i.e., Br and $NO_3$ instead of the hydronium ion) (Huang et al., 2021). The volatility of the
extended list of VOCs was assessed by estimating the effective saturation mass mixing ratio ($C_{sat}$). The
parameterization of the volatility, based on the number of carbon, oxygen, and nitrogen atoms (Donahue et al., 2011;
Mohr et al., 2019), was calculated using the following equation:

$$log\left(C_{sat}\right) = (n_{O*} - n_C)b_C - (n_O - 3n_N)b_O - 2\frac{(n_O - 3n_N)n_C}{(n_C + n_O - 3n_N)}b_{CO} - n_N b_N \qquad (4)$$
where $n_{0*} = 25$, $b_C = 0.475$, $b_O = 0.2$, $b_{CO} = 0.9$, and $b_N = 2.5$. The terms $n_C$, $n_O$, and $n_N$ are the number of carbon,
oxygen, and nitrogen atoms, respectively. The average log saturation mixing ratio for all the compounds was
7.50 µg m$^{-3}$, and 100 and 136 compounds were classified as intermediate and volatile organic compounds,
respectively. Log $C_{sat}$ values below 3 µg m$^{-3}$ were recorded for three compounds (i.e., $C_6H_3NO_3$, $C_{10}H_{10}O_3$, and
$C_{12}H_{23}NO_3$), which categorized them as semivolatile VOCs.

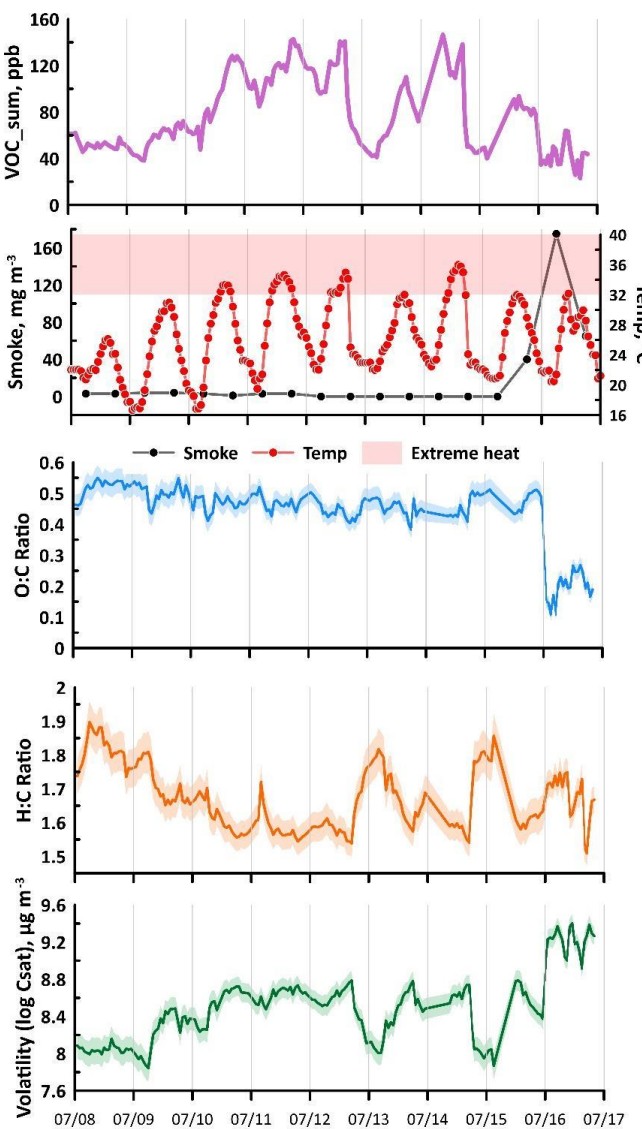

**Figure 5.** *Time series profile of the sum of VOC mixing ratio, smoke, and temperature observed at MOFLUX, weighted O:C, H:C ratios, and the volatility during the intensive operational period between July 8 to 17. The shaded regions of O:C, H:C, and volatility are the weighted standard deviations.*

We analyzed July 8 through 17 to develop a deeper understanding of how the extended list of VOCs was influenced by extreme heat and combustion plumes. Note that in these analyses, the concentrations of acetone, isoprene, and MVK+MACr were not included to focus on the extended list of VOCs. During this period, the average VOC mixing ratio was 78 ppb. Figure 5 also shows the profile of the total VOC mixing ratios, which depicts a mixing ratio range between 23 to 147 ppb. Figure 5 illustrates the profiles of the weighted average of the O:C ratio, H:C ratio, and volatility of the VOCs. The O:C ratio was consistent, ranging between 0.4 and 0.55. However, the apparent transport



of the combustion plume to the site decreased the ratio to less than 0.3, indicating the dominance of the less oxygenated
compounds in the atmosphere as a result of BB. The H:C ratio (1.8) was increased at the start period, characterized
by low temperatures, which signifies the presence of highly unsaturated compounds such as aromatics. As the
temperature increased, H:C values (1.5) decreased except during the period with biomass burning emissions (1.75),
implying the alteration of VOC distribution. Lastly, elevated temperature resulted in the emission of more volatile
compounds, as the weighted average volatility reached 8.5. The atmosphere over MOFLUX was further enriched with
volatile compounds during the passage of the combustion plume, as the mean log $C_{sat}$ reached 9 µg m$^{-3}$.

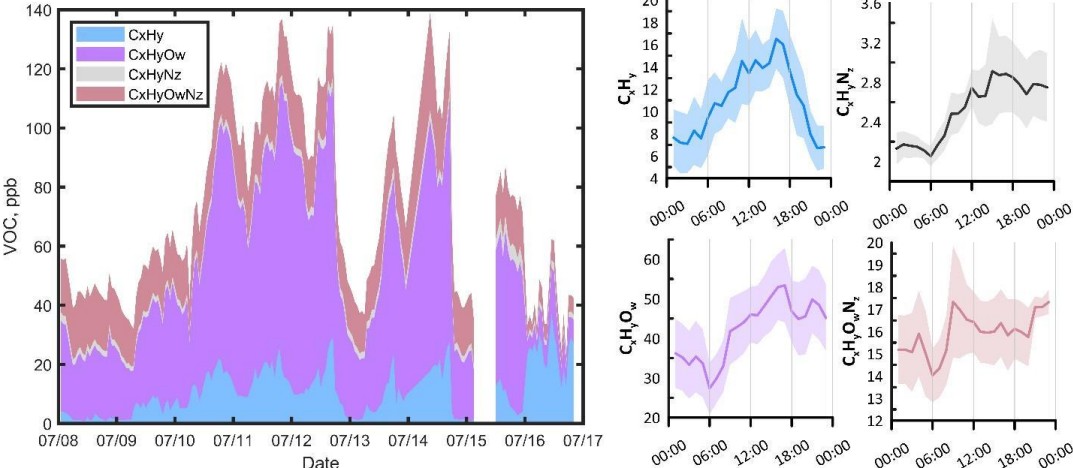


*Figure 6. (Left) Time series and (right) diurnal profile of the clustering based on the atomic content of the VOCs during the*
*intensive observation period with enhanced temperature and combustion plume transport at MOFLUX. $C_xH_yO_wS_v$ compounds were*
*not included due to low mixing ratio compared to other categories.*
Figure 6 shows the profile of the extended list of VOCs, clustered according to atomic content. The decreasing order
of average concentration was as follows: $C_xH_yO_w$ (41 ppb), $C_xH_yO_wN_z$ (15 ppb), $C_xH_y$ (11 ppb), $C_xH_yN_z$ (2.4 ppb),
$C_xH_yO_wS_v$ (0.48 ppb). Hydrocarbons ($C_xH_y$) had evident enhancement at elevated temperature, as well as during the
later hours of the transport of the BB compounds on the 16[th] of July. Oxygenated hydrocarbons ($C_xH_yO_w$) had a
delayed response to temperature, in which peak concentration occurred around 18:00. Such categories also showed
increased concentration during the initial hours of combustion plume impact at MOFLUX. $C_xH_yN_z$ compounds such
as acetonitrile showed clear augmentation during the initial hours of the plume transport but exhibited a less sensitive
response to changes in temperature. For $C_xH_yO_wN_z$ and $C_xH_yO_wS_v$ compounds, temperature and BB had little to no
effect on either group, except for the clear reduction during the latter hours of combustion plume passage in the
MOFLUX temperate forest. The changes in the distribution of the extended list of VOCs, beyond the major
compounds, validated the substantial influence of temperature and BB on the overall chemical reactivity of the
atmosphere.

Several compounds among the extended list showed an enhanced mixing ratio at high temperatures (>32°C). Besides
the major compounds such as isoprene and monoterpene, VOCs such as formic acid ($CH_2O_2$), acetic acid ($C_2H_4O_2$),



isocyanic acid (HCNO), acrolein ($C_3H_4O$), furan ($C_4H_4O$), methylglyoxal ($C_3H_4O_2$), glycolic acid ($C_2H_4O_3$), and
propanethiol ($C_3H_8S$) exhibited at least 100% increases at enhanced temperature conditions. Formic and acetic acid,
as two of the most dominant acids in the atmosphere, are key VOCs in aerosol growth, cloud precipitation, and
rainwater acidity. Formic acid is primarily formed through photochemical production but can be emitted directly from
vegetation, which is a temperature-dependent process (Millet et al., 2015). Given the anticipated increased
temperatures due to extensive fossil fuel burning, addressing the enhancements of formic and acetic acid will advance
our knowledge of future chemistry–climate interactions.
**3.5 Source Apportionment of VOCs Measured during Extreme Temperature and Biomass Burning**

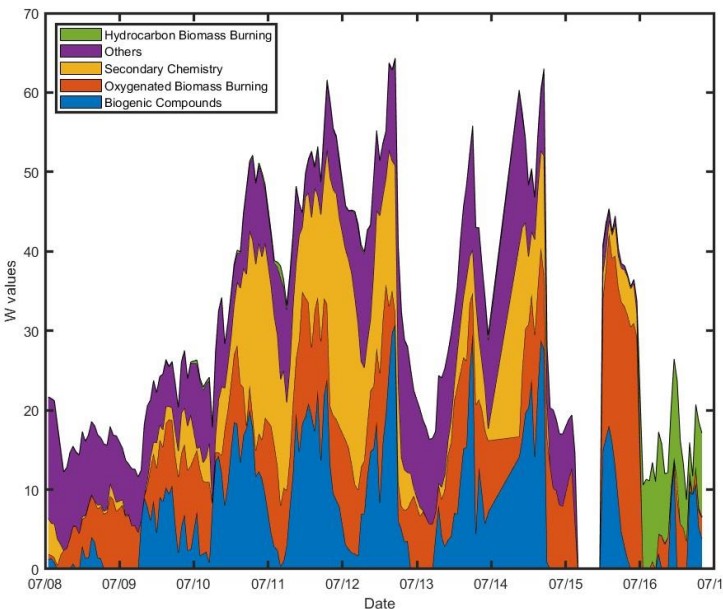


*Figure 7. Stacked profile of the non-negative factors of species fingerprints from all the VOCs measured at MOFLUX.*
To systematically investigate the pattern and contributions of the extended list of VOCs, a NNMF routine was applied
to study the prominent sources of the VOCs in the forest between July 8 and 17. Based on the dominant tracers,
response to the range of temperatures, impact of combustion plume, and diurnal variations, five important categories
were identified from the NNMF analysis: (i) Biogenic-related Compounds, (ii) Secondary Chemistry, (iii) Oxygenated
BB compounds (O-BB), (iv) Hydrocarbon BB compounds (H-BB), and (v) Others. The "others" factor, with a
substantial number of contributing compounds, remains unidentified.

The biogenic factor, which consisted primarily of isoprene and monoterpene, followed the profile of temperature,
which supports the attribution to biogenic sources. Two BB factors were accounted for, which were separated based
on the chemical composition of the gases contributing to each factor. The median formulas for O-BB and H-BB were



$C_4H_5NO$ and $C_6H_8$, with a mode of 1 and 2 degrees of unsaturation, respectively. This is consistent with the analysis
based on atomic content: oxygenated hydrocarbons ($C_xH_yO_w$) and $C_xH_yN_z$ compounds were persistent in the early
hours of the combustion plume, whereas pure hydrocarbons ($C_xH_y$) showed evident enhancement in the later hours of
July 16. Also, the saturation mixing ratio of the O-BB and H-BB factors were 7.27 and 8.45 µg m$^{-3}$: these values place
both factors under the volatile category (log $C_{sat} > 6$ µg m$^{-3}$). The prominent compounds under the O-BB factor were
acetonitrile ($C_2H_3N$), formamide ($CH_3NO$), maleic acid ($C_4H_4O_2$), hydroxyfuranone ($C_4H_4O_3$), butyramide or
dimethylacetamide ($C_4H_9NO$), benzonitrile ($C_7H_5N$), and furaldehyde ($C_5H_5O_2$), which were all previously detected
in field- and lab-scale measurements of combustion plumes (Jain et al., 2023; Stockwell et al., 2015; Coggon et al.,
2019; Salvador et al., 2022). The H-BB factor was populated by unsaturated hydrocarbons such as butadiene ($C_4H_6$),
butene ($C_4H_8$), pentenes ($C_5H_{10}$), benzene ($C_6H_6$), hexadiene ($C_6H_{10}$), and ethylbenzene ($C_8H_{10}$), although cyclic
hydrocarbons are not discounted. Interestingly, monoterpene at $C_{10}H_{17}^+$ and its fragment at $C_7H_8^+$ had substantial
contributions from the H-BB factor during this period. This is unlikely due to the expected biogenic emission in the
forest, although it came second with 34% contribution compared to 66% from the H-BB factor. However, several
prior studies have shown that monoterpene can also originate from anthropogenic activities (Coggon et al., 2021),
particularly BB events (Wang et al., 2022). With the enhancement of monoterpene and other unsaturated hydrocarbons
(e.g., butenes and ethylbenzene) during combustion plumes, several changes in atmospheric reactivity are expected,
such as photochemical ozone production, scavenging of OH radicals, and suppression or enhancement of aerosol
formation.

The secondary chemistry factor had a median chemical formula of $C_3H_4O_2$ with two degrees of unsaturation. Among
the factors identified, this factor had the highest oxygen content, with a median and max of 2 and 5, respectively.
Similar to the oxygenated hydrocarbon group, this factor had a diurnal profile characterized by evening enhancement
(~20:00). Also, the secondary factor is marked by compounds such as ethenone ($C_2H_2O$), acrolein ($C_3H_4O$), acetic
acid ($C_2H_4O_2$), MVK+MACr ($C_4H_6O$), hydroxyacetone ($C_3H_6O_2$), and acetylacetone ($C_5H_8O_2$). Due to its nighttime
peak as well as the oxidized nature of the major VOCs at this factor, it is assumed that these products are generated
from the nighttime oxidation using $NO_3$ radicals of the organics in the forest. Isoprene also generates MVK during
nighttime through the dominant β-$RO_2$ isomer formation pathway (Ng et al., 2017). During the transport of BB plumes,
the secondary factor had a relatively low increase in signal compared to both BB factors, which shows that oxidation
compounds were generally locally generated with little to no contribution from long-range transport. BB tracers
dominated the air mass of MOFLUX during the transport of the combustion plume, which drastically affected the
atmospheric chemistry of the area.
**4. Summary and Atmospheric Implications**
Critical VOCs, which have important contributions to several atmospheric processes, were continuously measured in
a temperate deciduous and juniper forest in the midwestern US during the summer of 2023 using PTR-ToF-MS.
During the measurement period, the forest included several sources of biogenic compounds and was influenced by
short- and long-range transport of anthropogenic emissions. Extreme heat and wildfire emissions impacted the



atmospheric conditions of the forest during the field measurement; such emissions are vital phenomena that provide
insights into future climate. Typical VOCs in the forest, consisting of methanol, acetone, isoprene, monoterpene,
MVK+MACr, benzene, toluene, acetonitrile, and catechol, had an average total mixing ratio of 69 ppb, which
highlights the strong effect of such VOCs on atmospheric reactivity in a temperate forest.
Among the VOCs, isoprene had one of the highest recorded mixing ratios (10 ppb), next to methanol (23 ppb) and
acetone (22 ppb). Enhanced temperature induced the emission of isoprene to a greater extent than did UV, based on
the late afternoon diurnal peak that coincides with temperature. At extreme temperature conditions, isoprene was
observed at twice the typical level, with a 1.32 ppb increase per 1.0°C. At the same time, for monoterpene, which was
suspected to be primarily ocimene or sabinene based on its daytime peak, a three-fold enhancement at extreme
temperatures was observed. The large gap between the mixing ratios of isoprene and monoterpene suppressed the
formation of aerosols due to the scavenging of OH radicals and reduction of $C_{20}$ dimers. AVOCs such as benzene
(0.42 ppb) and acetonitrile (1.52 ppb) responded much less to changes in temperature compared to the BVOCs. The
varying enrichment of the major VOCs and their response to extreme temperatures had a serious impact on the
potential aerosol formation and chemical reactivity. New studies indicated that the coexistence of multiple precursor
VOCs can generate unexplored molecular-scale interactions, which is critical as current and future VOC distributions
are expected to be widely different compared to current conditions.
Besides the role of elevated temperature, the duration of the VOC measurement at MOFLUX was marked by sporadic
transport of plumes from wildfires in Canada. The impending warming of the atmosphere is projected to potentially
increase the frequency and duration of wildfires due to drier seasons; however, increases could be significantly
affected regionally by accompanying changes in atmospheric circulation. In MOFLUX, the profiles of the major VOCs
such as benzene and acetonitrile changed notably with respect to combustion plumes, as their concentrations were
enhanced by more than 100%. Because benzene is an important precursor of ozone and aerosol formation, the
variability of such AVOCs should be included in the simulation of future atmospheric processes.
Beyond the major VOCs, analysis of the whole mass spectra revealed more than 250 other compounds, the mixing
ratios of which sum to as much as 78 ppb during a period with elevated temperature (>305 K) and BB plumes (smoke
> 100 mg m$^{-3}$). With a similar mixing ratio sum to those of the typical VOCs, analysis of unaccounted-for VOCs is
necessary to perform a realistic investigation of the relevant atmospheric processes at the MOFLUX site. The O:C and
H:C ratios of all the VOCs, as well as their volatility, provided insight into their response to future climate scenarios.
During BB plume transport, less oxygenated compounds with high volatility were enhanced. Elevated temperature in
the forest induced the formation and emission of more unsaturated compounds. Hydrocarbons (CxHy) and $C_xH_yN_z$
compounds dominated the total VOC mixing ratio during the initial and later hours of biomass transport, respectively,
whereas oxygenated hydrocarbons persisted consistently during periods of elevated temperature. Furthermore, the
analysis of the entire spectra pinpointed an additional 40 compounds that have at least 100% enhancement in mixing



ratio at extreme temperatures. Two of them are formic (0.89 ppb) and acetic acid (3.29 ppb), which have a vital impact
on atmospheric acidity and cloud formation.

The highly variable profiles of the extended list of VOCs measured at MOFLUX clearly indicated that species were
impacted by a variety of emissions and processes. NNMF was applied to the VOC mixing ratios, and five factors were
identified: two BB factors and one each for secondary chemistry, biogenic, and others. The two BB factors were
resolved based on the chemical composition of the compounds contributing to each factor. With the high reactivity of
such compounds to oxidants such as OH and $NO_3$, it is expected that BB altered the normal forest-dominated
atmospheric processes.

The comprehensive analysis of the whole mass spectra performed in this study underscores the importance of
unaccounted-for VOCs in the total chemical reactivity of the atmosphere. The results of this study highlight the
possible unaccounted modifications in VOC distribution that might be expected in future climate scenarios with
serious impacts on aerosol–climate interaction. With the growing but still limited insights on the effect of mixed
precursors on aerosol formation, more information regarding the overall distribution and transformation of AVOCs
and BVOCs and their response to different future climate scenarios are needed to realistically account for the climate
forcing of organic aerosols.



**Data availability**

The data used in this publication are available to the community and can be accessed by request to the corresponding author.

**Author contribution:**

CMS, JDW, EGC, HAS, BDK, and SSO conducted the measurements. CMS, JDW, MAM, KRB, and GL designed the project, coordinated the measurements, and supervised the study. MAM, GL, and KRB obtained funding for the project. CMS and KRB carried out the data curation and analysis. CMS prepared the manuscript. All co-authors contributed to the discussion and the interpretation of the results.

**Competing interests:**

The authors declare that they have no conflict of interest.

**Acknowledgements**

This research was funded through Oak Ridge National Laboratory (ORNL)'s Directed Research and Development (LDRD) program. ORNL is managed by the University of Tennessee-Battelle, LLC, under contract DE-AC05-00OR22725 with the U.S. Department of Energy. The U.S. Drought Monitor, which provided the drought data of Boone County, MO, is jointly produced by the National Drought Mitigation Center at the University of Nebraska-Lincoln, the United States Department of Agriculture, and the National Oceanic and Atmospheric Administration. The authors gratefully acknowledge the NOAA Air Resources Laboratory (ARL) for the provision of the HYSPLIT transport and dispersion model (https://www.ready.noaa.gov) used in this publication.

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
