# Peer review of "Measurement Report: Extreme Heat and Wildfire Emissions Enhance Volatile Organic Compounds in a Temperate Forest"

_EGUsphere, 2024_

## Author Comment (AC1)

**Extreme Heat and Wildfire Emissions Enhance Volatile Organic Compounds: Insights on Future Climate**

Christian Mark Salvador et al.

We appreciate the referee's detailed review of our manuscript  and we provided here a detailed response that addresses the referee's concerns. Our point-by-point responses to the Reviewer's general and specific comments are presented below. The referee's comments are in black, and our answers are in red.  Modified or new statements integrated into the revised manuscript are indented. All changes can be seen in the revised version of the manuscript in red font.

**General Comment (GC)**

**GC 1:** The work presented here reports the change in mixing ratio and distribution of several biogenic and anthropogenic VOCs (measured by CIMS) in a forest as a response to increased temperature and transported biomass burning plumes. The authors underscore the variability of VOCs as a result of heat and wildfire and claim to present a comprehensive analysis of the whole mass spectra performed in this study. While the subject matter of how the VOC distribution changes in response to extreme wildfire and temperature events in the context of future climate scenarios is highly relevant, the surface level results presented here unfortunately do not do well to support the conclusions and claims made by the authors. In its current form, the (sometimes incorrectly) drawn conclusions do little to further current knowledge. On this basis, and described in more detail in specific comments below, I recommend rejection.

**Response:** We appreciate the thorough evaluation of our work and understand the initial decision provided by the reviewer. In response, we carefully considered each comment and suggestion. The manuscript has been substantially revised to improve the readability and enhance the scientific impact of our research. Below are some of the major additions and revisions made to the manuscript:

1. **Chemical Reactivity** –  The original manuscript made several claims regarding the modification of atmospheric chemical reactivity due to changes in the overall distribution of VOCs caused by elevated temperatures and transported combustion plumes. However, the initial version lacked the supporting data or calculations for those claims. In response, we calculated the OH reactivity of each compound with an identified molecular formula to assess the alteration of chemical reactivity due to extreme atmospheric events. More details regarding the calculation of OH reactivity is provided in our response to specific comment (SC) 20. Statements about reactivity in the revised manuscript are now based on the results of the reactivity calculations. We added the time series profile of the total reactivity in Figure 5.

[Figure]

**Figure R1.** Time series profile total reactivity between July 8 to 17.

2. **Linear fit of Temperature and VOC Concentration** – One significant error in the earlier version of the manuscript was the fitting of a linear trend line to the correlation between ambient temperature and BVOC mixing ratios. We apologize for this oversimplification, and the revised version of the manuscript now correctly demonstrates the exponential relationship between temperature and VOCs, consistent with several previous studies. We have also presented the statistical analysis of the exponential fitting, highlighting the merits of this approach. The coefficients derived from this analysis can be used by modelling studies to evaluate the current and future influence of temperature on temperate forests.

[Figure]

3. **Interpretation of O:C ratio During Transport of Combustion Plume –** Changes in the O:C ratio was observed during biomass burning transport to MOFLUX, but the initial explanation provided was inaccurate. We appreciate the reviewer for suggesting a more plausible explanation, which has now been incorporated into the revised manuscript.

4. **Missing Supporting Data for Claims and Conclusions** – Several statements in the original manuscript were made without sufficient supporting evidence. These include the assertion that mass 93 is a fragment of monoterpene and that benzene enhancement is due to biomass burning. Plots have now been included in the supplement to substantiate these claims in the main text. Any statements that could not be supported by our data, calculations, or results have been completely removed from the revised version of the manuscript.

5. **Interpretation of Smoke** – The Smoke parameter was extensively used in this research to indicate the extent of biomass burning's influence on the atmosphere in the

temperature forest. However, the original manuscript lacked an in-depth explanation and data basis for this parameter. The revised manuscript now provides a clearer explanation of the Smoke parameter and includes a reference study for readers interested in further investigating or utilizing this biomass burning parameter.

6. **Formatting and composition guideline** – The original version of the manuscript has been carefully revised to comply with the formatting and composition guidelines of ACP. Specifically, the figures were updated to include letter notations that are referenced in the corresponding captions. Additionally, uncertainties in some graphs were addressed by integrating error bars to provide a clearer representation of the data.

These are just some of the major revisions made to our research work. We sincerely hope that the reviewer appreciates our comprehensive response and the modifications outlined in this document and reconsider the decision regarding the publication of our work in ACP. We very much appreciate the amount of time and effort the reviewer has spent in bringing these issued to our attention. We remain grateful for their contribution to improve the quality of our research product.

**GC 2:** Despite the above, the manuscript has potential to be highly novel and impactful should the authors take steps to further the analysis and provide additional clarification in the methods. A refinement of the results would also allow for comparison to regional and global assessments, which would make the work far more reaching and useful. A review of recent literature would be beneficial in relating the relevance (and significance) of the work presented here to work already published.

**Response:** Several new statements were added in the recent version manuscript to compare our results with similar prior studies. For instance the following text was incorporated to compare our O:C ratios and acetonitrile levels with previous studies that showcased the influence of biomass burning:

*This is consistent with a prior study that reported low O:C ratio (0.25) during intense biomass burning plume, compared to the measurement period when the smoke became diluted and impact of biogenic emission was enhanced (O:C =0.7) (Brito et al., 2014).*

*Additionally, the observed acetonitrile levels are beyond the mixing ratio range (0.047 to 1.08 ppb) recorded in Asian, US, and European regions (Huangfu et al., 2021), indicating the severe impact of biomass burning (BB)*

**GC 3:** Broadly speaking, the difference in quantitative metrics (for example estimated OH reactivity, estimated reactive organic carbon, change in species abundance, etc) between typical conditions and high temperature and smoke impacted conditions are lacking and could be better analyzed and presented. Several of the figures are missing estimates of error and need to be further refined. Results presented and implied conclusions are missing context or relevance. The comparison to previous studies and literature is lacking and needs to be elaborated on. For instance, it would be useful to know how many of the species measured here by the CIMS are not included in climate predictions (e.g models). Presumably, it's quite a few given that global prediction models lack the complexity of emissions measured here. For models that discuss the importance in certain VOCs, it would be useful to know what those VOCs are and if they were also measured here. These are just a few examples of how the manuscript can be improved and resubmitted. Additional comments are provided below.

**Response:** We concur with the reviewer and we integrated the OH reactivity data as an important quantitative metric to showcase the impact of future climate events. The comprehensive calculation was performed for all 275 compounds with assigned molecular formulas. A total reactivity value is now presented in the manuscript, which highlights the potential influence of biomass burning and elevated temperatures. Additionally, key conclusions are supported by evidence such as correlation plots between VOCs and linear trends of tracers, paired with observations of atmospheric events. Insights without appropriate proof have been removed to prevent confusion and misinformation for the readers.

While we recognize that major compounds such as isoprene, monoterpenes and methanol are commonly included in climate projections, we are not familiar with previous studies that extensively list all species that could be referenced in our study.

**Specific Comments (SC):**

**SC 1:** Prior to resubmission I highly encourage and recommend the author review formatting and composition guidelines presented by ACP.

**Response:** The revised manuscript has been formatted according to the guidelines of ACP, with particular attention to the panel labels in the figures. Multi-plot figures are now labeled with letters, and the corresponding notations have been included in the figure captions.

**SC 2:** The background is fairly general and could benefit from more specificity when referencing literature. Specifically, elaboration on how the reviewed literature fits into the context of this study could help strengthen the results presented.

**Response:** The introduction has been revised to provide insights into the future climate conditions, particularly extreme temperatures and the increased frequency of global wildfires. During our measurement period, the MOFLUX site experienced such events, providing relevant data that can be used to explore the variability of atmospheric parameters and components. This was emphasized in the first and second paragraphs of the introduction.

The third paragraph highlights previous studies that investigated the impact of future conditions on VOCs. We also included reaction schemes and discussed the contribution of these compounds in the atmosphere. While we provided information on the influence of temperature on VOC emission and transformation, the section lacked a discussion on the impact of increasing wildfire events on VOCs. To address this, the following statements regarding previous studies on the influence of BB on VOCS have been added in the revised manuscript.

> *Moreover, BB events such as wildfires are considered as the second-largest source of VOCs globally, further influencing air quality and climate (Jin et al., 2023; Yokelson et al., 2008). Benzene, a common compound emitted during wildfire events, has been found to be more than ten times the typical concentration in metropolitan areas, thereby posing elevated health risks (Ketcherside et al., 2024).*

**SC 3:** Coordinates need a degree sign and space when naming the direction. There are a few spots in the methods where the authors only added the lat, long in parentheses. This needs to be corrected.

**Response:** We concur with the reviewer. All coordinates presented in the manuscript have been converted to degrees minutes seconds (DMS) format. Here are the updated texts:

*Measurements were conducted at the Missouri Ozark AmeriFlux (MOFLUX) site (latitude: 38° 44' 38.76" N, longitude: 92° 12' 0" W) in central Missouri, United States*

*Figure 1 shows the time series profile of hourly averages of temperature and relative humidity collected from Columbia Regional Airport (latitude: 38° 49' 1.2" N, longitude: 92° 13' 15.6" W) approximately 8.5 km from the MOFLUX site.*

*Global solar radiation data were measured at a weather site in Ashland, MO (latitude: 38° 43' 19.2" N, longitude: 92° 15' 10.8" W), 5.22 km from the MOFLUX tower.*

**SC 4:** For methods, It would be useful to know how tall the tower is and where the instrument was situated.

**Response:** The height of tower and location of the PTR-TOF-MS were added in the methodology section. The following statements were added:

*The PTR-ToF-MS was located in a climate-controlled cabin shed at the base of the MOFLUX tower.*

*Ambient air was sampled from the MOFLUX tower with the height of 32 meters.*

**SC 5:** L44: Not sure what is meant by "components", consider rewording for clarity.

**Response:** The word component was replaced with ***constituents***, which indicates traces gases that are influences by warming of the atmosphere.

**SC 6:** L53: Current wording of this sentence is awkward. Consider rephrasing to, "One potential effect of overall atmospheric warming is the change in global wildfire frequency"

**Response:** Done. Thank you for the suggestion. The first sentence of 2$^{nd}$ paragraph now reads:

*One potential effect of overall atmospheric warming is the change in global wildfire frequency.*

**SC 6:** L55: Cite Juang et al., 2022 as a reference to enhancements in wildfire and soil moisture/aridity: https://agupubs.onlinelibrary.wiley.com/doi/full/10.1029/2021GL097131

**Response:** Done. Juang et al., 2022 was referenced in the discussion soil moisture evaporation.

**SC 7:** L63/64: Do you have a reference that provides evidence of this? Please cite.

**Response:** The atmospheric process indicated in L63/64 were previously discussed in two studies, which are:

Juang, C.S., Williams, A.P., Abatzoglou, J.T., Balch, J.K., Hurteau, M.D., Moritz, M.A. (2022**). Rapid Growth of Large Forest Fires Drives the Exponential Response of Annual Forest-Fire Area to Aridity in the Western United States**. Geophysical Research Letters 49, e2021GL097131. https://doi.org/https://doi.org/10.1029/2021GL097131

Liu, T., Hong, Y., Li, M., Xu, L., Chen, J., Bian, Y., Yang, C., Dan, Y., Zhang, Y., Xue, L., Zhao, M., Huang, Z., Wang, H. (2022). **Atmospheric oxidation capacity and ozone pollution mechanism in a coastal city of southeastern China: analysis of a typical**

**photochemical episode by an observation-based model**. Atmos. Chem. Phys. 22, 2173-2190. https://doi.org/10.5194/acp-22-2173-2022

**SC 8:** L66: Change to say, "the abundance of volatile organic compounds are expected…"

**Response:** We appreciate the suggestion; the statement now reads:

*Among the chemical components of the atmosphere, the abundance of volatile organic compounds (VOCs) is expected to respond to extreme heat and wildfire emissions.*

**SC 9:** L74-75: Can you specify what "future" refers to here, or provide a time frame? The cited study is nearly 20 years old now, so it would be useful in context.

**Response:** We added the years **(2070-2099)** used in the calculation of future emission scenario indicated in the study cited in the introduction section. The statement now reads:

*For instance, a global estimate of isoprene emissions with temperature and land-cover drivers under future scenario (year: 2070-2099) was 889 Tg yr$^{-1}$.*

**SC 10:** L109-110: Not sure what the significance is of including averages from January and July? Consider removing if not relevant

**Response:** We concur with reviewer that January data is not relevant in this section, however, we kept the average temperature of July since it overlaps with our period of measurement (i.e., June 25 to August 12). The new sentence is as follow:

*Long-term measurements of meteorological parameters (1981–2010) at a nearby airport (~10 km) indicated that the average temperature for July was 25.2°C.*

**SC 11:** L111: Specify if this is an annual average, and change units to cm.

**Response:** The precipitation value presented is annual average. The sentence was improved to better convey the information. The unit was changed to cm as well.

*Typical precipitation (annual average: 108.2 cm) is fairly evenly distributed through the yearly cycle.*

**SC 12:** L139: Specify what "calibrated regularly" means. How often and for how long? (e.g every 5 days for 20 minutes, daily for 20 minutes, etc).

**Response:** The mass spectrometer was calibrated every two weeks for 50 minutes. This information is now included in the methodology section.

*The PTR-ToF-MS was calibrated regularly every two weeks for 50 minutes using a 110-ppb mixture of gases.*

**SC 13:** L141: Are the mixing ratios for the entire mixture or for each compound?

**Response:** The standard gas supplier (Airgas) prepared a gas mixture with each compound at concentration of 110 ppb. The range of mixing ratios during the calibration also applies to every compound. The statement was modified to account the reviewer's comment.

*The linear calibration curve for each standard compound consisted of eleven data points, with mixing ratios ranging between 1.89 and 50.9 ppb.*

**SC 14:** L141-142: It's unclear what is meant by the sentence, "The same compounds were used to calculate the mixing ratio of other compounds using the transmission efficiency and first-order kinetic reaction." Please elaborate or reference to a manuscript that explains the method further.

**Response:** PTR-ToF-MS can provide quantitative measurement of compounds without standard gas available using mass dependent transmission analysis. The conversion of raw signals (counts per second) to mixing ratio (ppb) of an uncalibrated gas VOC can be performed using the following equation:

$$[VOC, ppb] = \frac{1}{k\Delta t} \times \frac{I(VOC^+)}{T(VOC^+)} \times \frac{I(H_3O^+)}{T(H_3O^+)}$$

Where $RH^+$ is the protonated gas compound, $k$ is the proton-transfer-reaction rate coefficient, $\Delta t$ is the reaction time, $I(VOC^+)$ and $I(H_3O^+)$ are the measured ion count rates for the $RH^+$ and the hydronium ion ($H_3O^+$), respectively. $T(VOC^+)$ and $T(H_3O^+)$ are the transmission efficiencies for $RH^+$ and $H_3O^+$ ions, respectively (Worton et al., 2023; Taipale et al., 2008). All the values are readily available except for transmission efficiency value, which can be determined by generating a mass dependent transmission curve from compounds with known concentrations and reaction rate. The transmission is an instrument-specific parameter that depends on the transmission efficiencies of the lens system/ion guide, the mass filter (TOF), and the ion detector. Transmission Tool provided by the instrument developer (IONICON) was used to generate the transmission efficiencies of gas standards.

The following statements regarding the conversion of raw signals to mixing ratio were added to the Methods section:

> *PTR-ToF-MS can provide quantitative measurement of compounds without standard gas available using mass dependent transmission analysis. The conversion of raw signals (counts per second) to mixing ratio (ppb) of an uncalibrated gas R can be performed using the following equation*
>
> $$[VOC, ppb] = \frac{1}{k\Delta t} \times \frac{I(VOC^+)}{T(VOC^+)} \times \frac{I(H_3O^+)}{T(H_3O^+)}$$
>
> *Where RH+ is the protonated gas compound, $k$ is the proton-transfer-reaction rate coefficient, $\Delta t$ is the reaction time, $I(VOC^+)$ and $I(H_3O^+)$ are the measured ion count rates for the RH+ and the hydronium ion ($H_3O^+$), respectively. $T(VOC^+)$ and $T(H_3O^+)$ are the transmission efficiencies for $RH^+$ and $H_3O^+$ ions, respectively (Worton et al., 2023; Taipale et al., 2008). All the values are readily available except for transmission efficiency value, which can be determined by generating a mass dependent transmission curve from compounds with known concentrations and reaction rate. The transmission is an instrument-specific parameter that depends on the transmission efficiencies of the lens system/ion guide, the mass filter (TOF), and the ion detector. Transmission Tool provided by the instrument developer (IONICON) was used to generate the transmission efficiencies of gas standards.*

**SC 15:** L150-152: How tall was the tower? How long was the measurement sample line or inlet? Was a filter applied for particles?

**Response:** Ambient air was sampled from the MOFLUX tower with the height of 32 meters. The air was drawn at the top of the tower using a 65-meter overall length with ½ in. OD PFA tube. A Teflon filter with 47 mm diameter was attached to inlet to prevent particles from entering the sampling line. All of this information was added to the recent version of the manuscript.

> *Ambient air was sampled from the MOFLUX tower with the height of 32 meters. The air was drawn at the top of the tower using a 65-meter overall length with ½ in. OD PFA tube (McMaster-Carr) and a GAST compressor/vacuum pump with a mass flow controller (Alicat Scientific, Inc) set at 20 L min⁻¹. A Teflon filter with 47 mm diameter was attached to inlet to prevent particles from entering the sampling line.*

**SC 16:** Figure 2: Please specify the time resolution of the averaged data. Is this hourly, daily, etc? It would be useful to also have some estimate of error or variability on the graphs of diurnal cycles.

**Response:** The VOC data were collected with a 100-millisecond time resolution and averaged into hourly data for processing and reporting. This statement has been added to the manuscript. Moreover, the authors added error bars, represented by standard error, to graphs in Figure 2.

[Figure]

***Figure 2.*** *(A-C) Average mixing ratio in ppb and (E-M) average diurnal profile of major VOCs at MOFLUX. Also included here is the diurnal profile of (D) temperature for reference. Time reported here is the local daylight time. The center lines of the box and whisker plots are the mean mixing ratio. Box edges are quartiles, and lower (upper) corresponds to 25th (75th). Whiskers represent 1.5 times the interquartile range. Symbols outside the box plot are outliers. Diurnal profiles have a unit of ppb mixing ratio. MVK and MACr are methyl vinyl ketone and methacrolein. The error bars are represented by standard error.*

**SC 17:** L190: How many is several? Define.

**Response:** The number of VOCs (n=275) was added in the text.

> *Many VOCs (n = 275) were detected in the ambient air throughout the three-month measurement period*

**SC 18:** L206: What is BB? Don't think the abbreviation has been defined yet.

**Response:** We thank the reviewer for noticing this error. We have defined BB in this section of the manuscript

> *Besides the photochemical oxidation of isoprene, MVK and MACr have other sources, such as Biomass Burning (BB) and gasoline vehicular emissions.*

**SC 19:** L236: It's unclear if the particle diameter is supposed to be greater than or less than 50 nm.

**Response:** This error was corrected by adding the appropriate mathematical sign in the text.

> *Relatively large particles (i.e., particle diameter > 50 nm) were observed with no apparent aerosol growth.*

**SC 20:** L264-267: While acetonitrile was more enhanced at this site during the BB event compared to other studies, it is not necessarily fair to say that the increase in acetonitrile alone 'highlights the severe impact of BB on atmospheric VOC distribution and reactivity.' It would instead be more appropriate to change the language to "implying the severe impact of BB…" or even better, providing a metric to confirm this. One idea is to compare the estimate average OH reactivity (OHr) of the measured VOC species during ambient times and compare it to the OHr during BB impacted times when acetonitrile is elevated. This would also help to strengthen the message of the manuscript.

**Response:** We concur with the reviewer regarding the addition of OH reactivity data, which provides supporting evidence to our claim about the impact of VOCs on the overall atmospheric reactivity in the temperate forest. To calculate the total OH reactivity (R) based on the measured concentration of VOCs, we used the following equation similar to a prior study (Wang et al., 2021):

$$R = \Sigma k_{VOC_i + OH}[VOC_i]$$

where $[VOC_i]$ is the concentration of the volatile organic compounds measured by the PTR-ToF-MS and $k_{VOC+OH}$ ($cm^3$ molecule$^{-1}$ s$^{-1}$) is the rate constant of the reaction between the OH and VOC. The rate constants were obtained from the National Institute of Standards and Technology (NIST) Chemical Kinetics Database which compiles kinetics data on gas phase reactions (https://kinetics.nist.gov/kinetics/). All molecular formulas identified from more than 250 ions were subjected to Reaction Database Quick Search Form. The calculation of the reaction constant accounted for the hourly temperature conditions measured during the field campaign. Only records with temperature range (20-36 °C) similar to the observed conditions in the temperate forest were considered in the calculation. The kinetic search procedure yielded 428 record matches for 82 ions. The median value of rate constants was used for molecular formulas with multiple records.

The authors added the following statement in the methodology and results and discussion section of the revised manuscript.

> *The total calculated OH reactivity (R) was obtained from the measured concentration of the VOCs using the following equation similar to a prior study (Wang et al., 2021):*

$$R = \Sigma k_{VOC_i+OH}[VOC_i]$$

> *where [VOC$_i$] is the concentration of the volatile organic compounds measured by the PTR-ToF-MS and $k_{VOC+OH}$ (cm$^3$ molecule$^{-1}$ s$^{-1}$) are the rate constant of the reaction between the OH and VOC. The rate constants were obtained from the National Institute of Standards and Technology (NIST) Chemical Kinetics Database which compiled kinetics data on gas phase reactions (https://kinetics.nist.gov/kinetics/). All molecular formulas identified from more than 250 ions were subjected to Reaction Database Quick Search Form. The calculation of the reaction constant accounted the hourly temperature conditions measured during the field campaign. Only records with temperature range (20-36 °C) similar to the observed conditions in the temperate forest were considered in the calculation The median value of rate constants was used for molecular formulas with multiple records.*

We also rephrased the statement in focus to the suggested wording by the reviewer.

> *Such values are beyond the mixing ratio range (0.047 to 1.08 ppb) of acetonitrile recorded in Asian, US, and European regions (Huangfu et al., 2021), implying the severe impact of BB.*

**SC 21:** L268-272: Is it possible to estimate an age of the BB event using a back trajectory or the abundance of compounds? For instance, furan containing species are often associated with fresh combustion and can be used to estimate smoke age. This would help to contextualize the compounds reported here.

**Response:** Due to the sporadic nature of the biomass burning events observed during the field campaign, we were unable to estimate the age of the combustion plumes that arrived at our sites.

**SC 22:** L274: Instead of saying "During some parts" it would be useful to have a quantitative measure (e.g number of days/hours). You could reference the supplemental histogram here.

**Response:** The number of hours when the temperature exceeded over 32 °C occurred more than 100 times during the measurement in the temperate forest. This was mentioned in the following statement in the revised text.

> *The extreme temperature, defined by an hourly mean temperature exceeded 32 °C, was based on the projected climate scenarios that temperature will increase by 2–4°C by 2100 (Collins et al., 2013). The extreme temperature occurred for **more than 100 hours** (see Figure S1 for histogram). The strong impact of the elevated temperature in the region ultimately altered the vegetation's physiological functions*

**SC 23:** Figure 3: Error bars should be added to the bar charts. It's unclear why a linear regression was used to fit the correlation between isoprene and temperature when the relationship is known to be exponentially related. This needs to be corrected. Equations should

be added to the correlation analysis figures and errors. Units are needed on the y axis. For linear regression, consider orthogonal regression over linear regression, as orthogonal distance regression takes into account error in both the x and y axis.

**Response:** The authors agree with the reviewer. The bar charts now include error bars, which represents the calculated standard errors. The major flaw of the manuscript regarding the linear relationship between isoprene and temperature has been corrected by the fitting exponential curve instead. Additionally, the statistical merits of the fitting, which include the equation of the line and correlation coefficient (r), are provided in Figure 3. The authors have retained the linear regression fitting for benzene. The caption of Figure 3 has been updated to reflect the changes in the figure.

The new Figure 3 and the corresponding caption are as follow:

[Figure]

*Figure 3. (A-B) Comparison of VOC mixing ratios for temperatures below and above 32°C. Catechol, not shown here, showed no evident difference between the two conditions (~30 ppt). The error bars in the bar chart are represented by standard error. (C-F) The correlation analysis of temperature with biogenic VOCs and benzene mixing ratios (in ppb). Correlation analysis of other major VOCs is provided in the supplement. Black symbols are the hourly data, whereas the red lines indicate the best-fit line of the binned mixing ratio of VOCs according to 1.0°C of temperature. The equation of the exponential fit line and correlation coefficients are given on the right side of the plot.*

**SC 24:** L290-L291: It is inappropriate to use a linear regression for isoprene and temperature. The relationship is known to be exponential. There several papers in the literature that show this. (Guenther at al., 1993, 2006; Rasulov et al., 2010; Hu et al., 2015; Selimovic et al., 2022; etc).

**Response:** We apologize for the error of using linear regression between temperature and isoprene, thank you kindly for pointing this out. The statements in this paragraph were updated to reflect the evident exponential relationship between temperature and the BVOCs.

> *The major BVOCs, isoprene and monoterpene, responded well to variations in temperature, as shown in Figure 3. Under extreme temperatures, the isoprene and monoterpene mixing ratios were 23 and 0.32 ppb, respectively, which were three times higher than the concentrations observed at temperatures below 32°C. The enhancement of isoprene and monoterpene also increased the reactivity of the atmosphere in the temperate forest, based on the calculated 8.31 $s^{-1}$ increase in OH reactivity. Furthermore, Figure 4 shows the evident exponential relationship between temperature and the major BVOCs, consistent with previous studies (Hu et al., 2015; Selimovic et al., 2022; Guenther et al., 2012). The empirically determined coefficients (β) for isoprene and monoterpene are 0.13 and 0.12.*

**SC 25:** L292: It's simpler to just say "was three time higher than conditions…"

**Response:** Done. Statement was revised accordingly.

**SC 26:** L293-297: See my earlier comments about which fit to apply for monoterpenes. Also the authors state, "the tenfold increase … had several implications for the distribution and chemical reactivity in the forest," but then provide no evidence or metric for reactivity to support this. An estimate of the change to reactive organic carbon (ROC) or OHr as a result of the increase would support this statement.

**Response:** The authors calculated the changes in OH reactivity due to enhancement of isoprene and monoterpene at elevated temperature conditions. We added the following statement

> *The enhancement of isoprene and monoterpene also increased the reactivity of the atmosphere in the temperate forest, based on the calculated 8.31 $s^{-1}$ increase in OH reactivity.*

**SC 27:** L300-301: Why? What is the significance of calculating this ratio? The wording could be changed.

**Response:** A comprehensive plant chamber analysis indicated that the suppression of new particle formation was dependent on the ratio of isoprene carbon to monoterpene carbon (Kiendler-Scharr et al., 2009). This impacts the aerosol formation events in the temperate forest. However, due to limited information and supporting data to explain our statements at L300-303, the authors have completely removed the aforementioned statements and the corresponding plot in the supplementary information.

**SC 28:** L302-L303: This statement lacks specificity. What values needs to be exceeded in which aerosol formation is suppressed? What is the optimum temperature? Why is it interesting or relevant that this occurs?

**Response:** See our response to the prior comment.

**SC 29:** L307-L311: What is the enhancement range in values for the temperature increase reported here? Would be useful to report so that a direct comparison can be made to previous literature.

**Response:** In the main text, the authors indicated that the measured MVK/MACr during the enhanced temperature conditions doubled compared at low temperatures. To explicitly highlight the impact of temperature on the concentration of MVK/MACR, the following statement was added to the main text.

> *As shown in Figure 3, the concentration of MVK/MACr doubled during extreme temperature conditions compared at low temperatures.*

**SC 30:** L306-L314: Can you report the ratio of isoprene to MACR+MVK during "low" temperatures and elevated temperature? This would provide an assessment of the lifetime and a metric for how oxidation changes between the two events. It would also be useful to compare during BB and non-BB events. See Hu et al., 2015, and Selimovic et al., 2022 for a discussion of this metric.

**Response:** We calculated the ratio of isoprene to monoterpene to assess the impact of degree of oxidation of isoprene to its primary oxidation products. During elevated temperature conditions (>32 °C), the average ratio was 1.21, while the value increased under low temperatures conditions (<32 °C, 1.27). Both values suggest transport time shorter than one isoprene lifetime, as indicated in previous studies (Selimovic et al., 2022; Hu et al., 2015). However, the higher values recorded at low temperatures are in stark contrast to the expected trend of decreasing ratio due to enhancement of MVK/MACr production from isoprene. As a result, the following statements were removed from the manuscript

> *Moreover, several of the reaction mechanisms during the oxidation of isoprene are temperature dependent (e.g., 1,6- and 1,5-H shift isomerization reactions of isoprene), which further augmented the formation of the first-generation products of isoprene (Navarro et al., 2013).*

Instead, the elevated concentration of the primary oxidation products was primarily attributed to enhancement of concentration of isoprene. The following statements were added in the manuscript.

> *The ratio between isoprene and MACR + MVK indicates the lifetime of the isoprene and degree of oxidation of isoprene to MVK and MACr. At elevated temperature (>32 °C), the average ratio was 1.21 while it increased at low temperatures (<32 °C, 1.27) conditions. Both values suggest transport time shorter than one isoprene lifetime, as indicated in the previous studies (Selimovic et al., 2022; Hu et al., 2015). However, the higher values recorded at low temperatures are in stark contrast to expected trend, where the ratio should decrease due to enhanced production of MVK/MACr from isoprene. The elevated concentration of the primary oxidation products was primarily attributed to enhancement of concentration of isoprene.*

**SC 31:** L319: AVOCs has not been previously defined.

**Response:** Anthropogenic VOCs (AVOCs) was already defined in the introduction section (L91)

**SC 32:** L319-320: It's not clear why a negative correlation between colder nighttime temperatures and AVOC would exist, especially when in the previous sentence the authors seem to imply the opposite is true, and that AVOCs are enhanced when the boundary layer is reduced (presumably at colder temperatures)?

**Response:** We recognize that the statements were confusing and do not have supporting information. Thus, the said texts were removed from the main text.

**SC 33:** L32gr1-322: I see no direct evidence to confirm this is due to higher temperature. A plot would be useful.

**Response:** Our apologies for the confusion. We meant to clarify that mass 93 is a fragment of monoterpene. We have improved the text and added a correlation plot between monoterpene and mass 93 in the supplement.

> *This result further supports our initial claim that the compound occurring at mass 93 originates from the fragmentation of monoterpene. The correlation plot in the supplement shows a direct relationship between the two compounds.*

The following graph was also added to the supplement:

[Figure]

**Fig S4:** Correlation plot of mass 93 and monoterpene at mass 137.

**SC 34:** L327-328: This needs to be reworked on the basis of extensive literature historically showing exponential relationships with temperature.

**Response:** Agree. The new statement is as follow:

> *On the other hand, BVOCs such as isoprene, MVK+MACr, and monoterpene showed exponential relationship with temperature but at varying rates.*

**SC 35:** L328-L332. It would be useful to know what VOC compounds the literature refers to here and whether or not they were also measured in the results reported in this manuscript.

**Response:** Agreed. We indicated in the main text the specific VOCs investigated in the oxidation of the mixed precursor system performed in previous studies.

> *Recent laboratory chamber studies have shown that unexpected interaction of individual VOCs (e.g., isoprene, monoterpene, toluene, xylene, and trimethylbenzene) during the oxidation process produced intermediates and products that impacted the yields, volatility, and other physiochemical properties of aerosols (Voliotis et al., 2021; Takeuchi et al., 2022; Chen et al., 2022).*

**SC 36:** L335: Do you have a reference for this?

**Response:** We were unable to find a literature that can support our statements regarding the integration of cross-reaction of precursors in regional and climate models. Thank you for pointing this out and we have removed these statements.

**SC 37:** Some of the information in Section 3.3 would be better introduced prior to discussion of BB impacts in Section 3.1 and 3.2. Earlier introduction would help to provide context.

**Response:** We understand the reviewer's concern regarding the proper introduction of biomass burning events and transport observed at our site. However, rather than relocating the introductory statements from section 3.3 to earlier paragraphs, we opted to compile all discussion related to biomass burning tracers in section 3.3 to enhance the readability of our manuscript. The following statements have been moved to section 3.3.

> *At MOFLUX, typical gas phase BB tracers were observed in substantial amounts. Acetonitrile, one of the prominent BB markers (Huangfu et al., 2021), had mean and maximum mixing ratios of 1.56 ppb and 4.45 ppb, respectively. Such values are beyond the mixing ratio range (0.047 to 1.08 ppb) of acetonitrile recorded in Asian, US, and European regions (Huangfu et al., 2021), implying the severe impact of BB. Acetonitrile did not follow a typical daily cycle, which is consistent with the sporadic nature of the emissions from a distant source and subsequent transport. Another prominent BB marker measured at the site was catechol, an aromatic compound directly emitted from combustion processes. At MOFLUX, catechol had a mean level of 30 ppt but increased significantly to 300 ppt on some days. Catechol had a minor peak during the daytime, which can be attributed to the photochemical processing of phenol (Finewax et al., 2018), another aromatic VOC emitted during BB events. Moreover, acetonitrile (r = 0.53) and catechol (r = 0.017) also did not follow the trend of temperature, which is consistent with the infrequent emissions of BB plumes.*

**SC 38:** L343-L345: Specificity on how wildfire smoke pollution periods were determined and separated would be beneficial. Were there times when both temperature was high >32 and BB was present? If so, how did the authors handle these in their comparisons?

**Response:** Wildfire plume transports were estimated using the NOAA's High-Resolution Rapid Refresh (HRRR) model at 3-km resolution, which generates the weather forecast for the entire continental US. The HRRR-Smoke model is based on single smoke tracer, plume rise parametrization, and satellite fire radiative power processing (Chow et al., 2022). Considering the intensive period between July 8 to 17 (see figure 5), there's limited overlap between the two

extreme events. This made the delineation of response of the VOCs to elevated temperature and transport of combustion plume easier.

To provide further clarity regarding the estimation of Smoke, the following statements were added to the Methods section:

> *Smoke concentrations (in mg m$^{-3}$) were estimated from the High-Resolution Rapid Refresh (HRRR) 3 -km weather model for Missouri at 6-hour intervals for the duration of the VOC data measurement period The HRRR model generates weather forecast for the entire continental US. The Smoke model is based on single smoke tracer, plume rise parametrization, and satellite fire radiative power processing (Chow et al., 2022)*

**SC 39:** L345: What defines stronger enhancement between the two? Can you provide some metric (e.g PM2.5, acetonitrile mixing ratios, etc?)

**Response:** The enhancement was based on the estimated Smoke concentrations shown in Figure 4, which revealed substantial concentrations between July 15-18 (~180 mg m$^{-3}$) compared to the observation between June 25 to July 1.

**SC 40:** L348-L351: How old is the air mass based on these trajectories? How long are the trajectories? What inventory was used? What is the resolution? What heights were the runs initialized at?

**Response:** The backward trajectories were calculated based on single trajectory, 500 m above ground level height, 24-hour duration and model vertical velocity as the motion calculation method. The meteorology used for the calculation was based on 1-degree Global Data Assimilation System (GDAS1).

This information was added in the methods section.

> *Backward airmass trajectories estimation was performed using was calculated using the Hybrid Single-Particle Lagrangian Integrated Trajectory (HYSPLIT) Model (Stein et al., 2015). The backward trajectories were calculated based on single trajectory, 500 m above ground level height, 24-hour duration and model vertical velocity as the motion calculation method. The meteorology used for the calculation was based on 1-degree Global Data Assimilation System (GDAS1).*

**SC 41:** Figure 4: What is "smoke" in the figure? Is this a combination of VOCs? Is this PM2.5? There is no definition for what is included in the smoke measurement.

**Response:** Please see our response to SC 38 regarding the estimation of Smoke parameter.

**SC 42:** L360-363: How many non-BB days were compared? How were non-BB days defined? Did non-BB days include extreme temperature events? If so, how was this separated?

**Response:** The figure was based on the recorded values during July 16, when a strong combustion plume was observed , and July 12 with no low concentration of smoke. As shown in Figure 5, the two extreme events did not overlap during these days.

**SC 43:** L363-L364: Not sure how the authors came to this conclusion?

**Response:** Prior to July 15, the typical concentration of benzene was around 0.4 ppb, which was accounted to automobile emissions. Between July 15 to July 17, benzene concentrations

followed the variability of the estimated Smoke (see figure S), indicating the impact of transported combustion plume on benzene levels. The following statement was added in Section 3.3

> *The major source of benzene shifted from vehicular emissions to BB, highlighting the diverse anthropogenic activities influencing the variability of benzene at the temperate forest, as shown in the time series profile of benzene and Smoke (see Figure S in supplement)*

The time series profile of benzene and smoke between July 8 to 17 was also added in the supplement.

[Figure]

**Figure S:** The time series profile of benzene and smoke between July 8 to 17

**SC 44:** L366-L368: This seems to be only for one day? Can you expand this analysis for the observation period to strengthen your results? This is also only benzene and ozone, so it's not fair to say that this one measurement is evidence of change to the overall chemical reactivity in the forest. On this note, it would be beneficial to have a measure of the regional applicability based on landscape and emission sources. Further, transported smoke plumes can also reduce the amount of sunlight getting to vegetation, impacting photolysis and potentially altering emissions of BVOC (notably isoprene) due to light and temperature reduction. This is an important consideration in the context of changing BVOC profiles due to changes in environmental factors. Based on Figure 5 it looks like the peak of smoke occurred during an extreme temperature event. Given the known relationship between ozone and temperature, how were the authors able to separate increases in ozone due to temperature versus the increase due to enhancements in VOC and BVOC precursors (e.g isoprene?)

**Response:** The statements regarding the increase of ozone formation potential of benzene was removed in the revised manuscript based on the suggestion of the other reviewer. The reviewer indicated that the benzene is not a key factor for ozone formation in a temperate forest, where biogenic VOCs such as isoprene are more abundant.

We agree with the reviewer that the transported plume can modify the incoming radiation that can influence the emissions of BVOCs. However, we don't have the necessary supporting data

to assess the changes in solar radiation due to transported biomass burning plume and the subsequent alteration of the variability of BVOCs.

The separation of the impact of elevated temperature and biomass burning was explained in-detail in our response to specific comment (SC) 50. Briefly, the influence of enhanced temperature was evident between July 8 at 1:00 to 15 at 6:00 while the contribution of combustion processes attributed primarily for the measurements between July 15 at 7:00 to July 17 at 20:00. The BB period only recorded one hour of extreme temperature conditions (>32°C), which was excluded during the calculations.

**SC 45:** An expansion of Section 3.4 to include references to previous literature would be beneficial in contextualizing the results. As it is currently written it's unclear what the significance of the reported results is. Previous studies (Brito et al., 2014) have utilized the O:C and H:C ratios as a marker for aging and to characterize organic aerosol.

**Response:** Done. We referenced a prior study that showcased the variability of O:C during the transport of combustion plume.

> *This is consistent with a prior study that reported low O:C ratio (0.25) during intense biomass burning plume compared during the measurement period when smoke became diluted and impact of biogenic emission enhanced (O:C =0.7) (Brito et al., 2014).*

**SC 46:** L396-397: It seems disadvantageous to exclude these compounds from the analysis, considering their global abundance and the importance that was placed on them in the earlier part of the manuscript

**Response:** We completely agree with the reviewer regarding the atmospheric significance of the acetone, isoprene, and MVK + MACr. However, integrating these VOCs to the multivariate analysis skews the results of the factorization methodology, which limits the contribution of the other VOCs with substantially lower concentrations.

**SC 47:** L397-L398: Is that the average VOC mixing ratio excluding those compounds? What is the standard deviation in the average VOC mixing ratio?

**Response:** The calculated average concentration of VOCs, including isoprene, acetone, and MVK + MACr is 97.8 ppb with standard deviation of 45.5 ppb. The three major VOCs contribute more than 35% of the total VOC concentration, which skews and impacts the matrix factorization analysis. The standard deviation of the average mixing ratio without acetone, isoprene and MVK + MACr is 31.5. This information was added in the main text.

> *During this period, the average VOC mixing ratio was 78 ppb with standard deviation of 31.5*

**SC 48a:** L401-L402: This conclusion is fundamentally incorrect. VOCs become more oxygenated as they are aged away from the biomass burning source. The oxidation of VOCs is what produces ozone and secondary organic aerosol. Multiple studies show this. The more likely explanation for a decrease in the O:C ratio is the increase in reactive organic carbon as a result of enhancements in VOC abundance due to BB and BB aerosol, which is overwhelmingly organic in nature. Additionally, the higher temperatures noted during the smoke period likely induce gas-particle portioning of transported BB aerosol, particularly VOC and IVOC compounds (classified in the manuscript) further contributing to a decrease in the O:C ratio.

**Response:** The authors agree. The statements were modified based on the suggestion of the reviewer. The new statement now reads:

> *However, the apparent transport of the combustion plume to the site decreased the ratio to less than 0.3 due to increase of reactive organic carbon.*

**SC 48b:** L401-L402Additionally, the higher temperatures noted during the smoke period likely induce gas-particle portioning of transported BB aerosol, particularly VOC and IVOC compounds (classified in the manuscript) further contributing to a decrease in the O:C ratio

**Response**: We believe that the change in O:C ratio was primarily due to biomass burning as temperature during transport of the combustion plume on July 16 did not exceed the criterion set for extreme temperature conditions.

**SC 49:** Figure 6: It would be more useful to plot the VOC types as a fraction of the total, to assess the distribution and how it changes, rather than the total abundance. Does the analysis presented in Figure 6 exclude the compounds previously mentioned?

**Response:** The authors concur. The time series of the percent contribution of each VOC class was added in the supplement. Moreover, the analysis of the presented in Figure 6 excluded the concentration of isoprene, monoterpene, and MVK + MACr.

[Figure]

***Figure S7:*** *Time series profile of percent contribution of each VOC class to total concentration during the intensive observation period with enhanced temperature and combustion plume transport at MOFLUX. $C_xH_yO_wS_v$ compounds were not included due to low mixing ratio compared to other categories.*

**SC 50:** L421-L423: The change in distribution in the extended list alone does not validate the substantial influence of temperature and BB on the overall chemical reactivity. A more appropriate measure of reactivity would be to calculate how the distribution of total reactive organic carbon (ROC) and OH reactivity (OHr) as a result of temperature and biomass burning influence.

**Response:** The authors agree with the reviewer and sincerely appreciate your feedback. Figure R2 presents the time series of calculated OH reactivity during the intensive period from July 8 to 17, which was influenced by extreme heat and combustion plumes. For reference, the time

series of smoke and temperature are also included. Note that the calculation of reactivities included isoprene, acetone, and MVK+MACr.

During this period, the average OH reactivity was 91.30 s⁻¹, with notable contributions from isoprene, acetone, ethylamine, and ethenone. To assess the impact of elevated temperature and biomass burning on atmospheric reactivity, the data were categorized based on recorded ambient temperature and smoke concentration. The influence of biomass burning was evident from July 15 at 07:00 to July 17 at 20:00. Only one hour within this period had a temperature above 32°C, in which that data point was excluded from the average reactivity calculation. Conversely, the effect of extreme temperatures was evaluated using data recorded from July 8 at 01:00 to July 15 at 06:00. Within this timeframe, 30 hours met the extreme temperature criteria, allowing an assessment of the potential impact of future warming on atmospheric reactivity.

The calculated OH reactivity for low and extreme temperatures was 98.92 s⁻¹ and 106.37 s⁻¹, respectively, indicating that elevated temperatures enhanced atmospheric reactivity. Additionally, transported combustion plumes increased reactivity to 106.00 s⁻¹ due to elevated concentrations of biomass burning tracers such as acetonitrile and benzene.

[Figure]

**Figure R1.** Time series profile total reactivity during the intensive operational period between July 8 to 17

The total reactivity plot was added in figure 5 and the following statements were added in section 3.4 to highlight the impact of extreme temperature conditions and transported plumes on reactivity.

> *Also in Figure 5 is the time series of the calculated OH reactivity during the intensive period influenced by extreme heat and combustion plumes between July 8 to 17. Note that the calculation of reactivities included isoprene, acetone, and MVK+MACr. During*

*this period, the average OH reactivity was 91.30 s $^{-1}$, with notable contributions from isoprene, acetone, ethylamine, and ethenone. To assess the impact of elevated temperature and biomass burning on atmospheric reactivity, the data were categorized based on recorded ambient temperature and smoke concentration. The influence of biomass burning was evident from July 15 at 07:00 to July 17 at 20:00. Only one hour within this period had a temperature above 32°C, in which that data point was excluded from the average reactivity calculation. Conversely, the effect of extreme temperatures was evaluated using data recorded from July 8 at 01:00 to July 15 at 06:00. Within this timeframe, 30 hours met the extreme temperature criteria, allowing an assessment of the potential impact of future warming on atmospheric reactivity. The calculated OH reactivity for low and extreme temperatures was 98.92 s $^{-1}$ and 106.37 s $^{-1}$, respectively, indicating that elevated temperatures enhanced atmospheric reactivity. Additionally, transported combustion plumes increased reactivity to 106.00 s $^{-1}$ due to elevated concentrations of biomass burning tracers such as acetonitrile and benzene.*

**SC 51:** L426-L428: Many of the compounds listed here are tracers for (typically fresh <1 day old) wildfire emissions. That it increased with temperature is likely a result of concurrent "smoke" enhancements as well (evident in Figure 5). Given this fact it cannot be stated that they increased 100% a result of enhanced temperature conditions alone, especially given the concurrence of the two events.

**Response:** The statements were modified to reflect the possible impact of biomass burning in the enhancement of the compounds listed in these statements.

*Besides the major compounds such as isoprene and monoterpene, VOCs such as formic acid ($CH_2O_2$), acetic acid ($C_2H_4O_2$), isocyanic acid (HCNO), acrolein ($C_3H_4O$), furan ($C_4H_4O$), methylglyoxal ($C_3H_4O_2$), glycolic acid ($C_2H_4O_3$), and propanethiol ($C_3H_8S$) exhibited enhancement at the extreme temperature conditions, although it is equally possible that these compounds were also associated with transport of the combustion plumes.*

**SC 52:** L488: This is presumably an average mixing ratio? Earlier the manuscript stated that isoprene reached a maximum of 75 ppb.

**Response:** *The value presented here is the average mixing ratio of isoprene. This was explicitly indicated in the new manuscript.*

*Among the VOCs, isoprene had one of the highest recorded average mixing ratios (10 ppb), next to methanol (23 ppb) and acetone (22 ppb).*

**SC 53:** L489-490: There are no measurements of light or photosynthetically active radiation to support this conclusion that temperature had a greater effect than UV.

**Response:** The statement was removed in the revised manuscript.

**SC 54:** L496-497: There is no metric (SOA formation potential, OHr, change to ROC) that supports this conclusion.

**Response:** Our new results regarding the chemical reactivity now supports this conclusion. However, no calculation of SOA formation potential was provided, thus it was removed from the statement. The new text now reads:

*The varying enrichment of the major VOCs and their response to extreme temperatures influenced the atmospheric reactivity in the temperate forest.*

**SC 55:** L497-498: It would be useful to know what VOCs these are and if there is overlap with the ones presented here.

**Response:** We agree with the reviewer. We added statements in this section regarding the VOCs studied during the oxidation of multiple coexisting VOC precursors.

*For instance, the coexistence of isoprene and monoterpene led to reduced hydroxyl radical availability, leading to a limited oxidation process (Mcfiggans et al., 2019). The interactions of anthropogenic VOCs (AVOCs) such as toluene, xylene, and trimethylbenzene produced more secondary organic aerosols, but the addition of biogenic VOCs (BVOCs) reduced the yield through cross-reactions between the intermediates (Chen et al., 2022).*

**SC 56:** L510: It's unclear why the authors all of a sudden switched to units of Kelvin? And what is smoke? How is it defined?

**Response:** Our apologies for the mistake. The temperature was converted to Celsius in the latest version of the manuscript. Smoke is defined earlier in this response.

*The mixing ratios of which sum to as much as 78 ppb during a period with elevated temperature (>32 °C ) and BB plumes (smoke > 100 mg $m^{-3}$).*

**SC 57:** L514-L515: There is no correlation analysis presented to support this conclusion.

**Response:** The statement regarding the formation of more unsaturated compounds due to enhanced temperature was disregarded in the revised manuscript.

**SC 58:** L518-L519: Some of these increases are likely associated with enhancement of wildfire emissions rather than temperature.

**Response:** The authors added the impact of the wildfire emissions on concentration of some VOCs in this statement.

*The analysis of the entire spectra pinpointed an additional 40 compounds that have at least 100% enhancement in mixing ratio at extreme temperatures and/or during the transport of wildfire emissions.*

**SC 59:** L525-527: This is likely broadly true but there is no evidence presented in the manuscript to support this conclusion. It would be useful to compare how the reactivity changes for each oxidant based on available kinetics data for the species measured.

**Response:**  The total reactivity calculated during the intense biomass burning plume transport observed between July 15 to 17 increased the reactivity to 106.00 $s^{-1}$ compared at non-BB period (98 78 $s^{-1}$), based on our calculations in section 3.4. However, the calculation was only based on reaction with OH oxidant, thus $NO_3$ was removed from the statement. The statement now reads:

*With the high reactivity of such compounds to OH radicals, it is expected that BB altered the normal forest-dominated atmospheric processes.*

**REFERENCES**

Brito, J., Rizzo, L. V., Morgan, W. T., Coe, H., Johnson, B., Haywood, J., Longo, K., Freitas, S., Andreae, M. O., and Artaxo, P.: Ground-based aerosol characterization during the South American Biomass Burning Analysis (SAMBBA) field experiment, Atmos. Chem. Phys., 14, 12069-12083, 10.5194/acp-14-12069-2014, 2014.

Chen, T., Zhang, P., Chu, B., Ma, Q., Ge, Y., Liu, J., and He, H.: Secondary organic aerosol formation from mixed volatile organic compounds: Effect of RO2 chemistry and precursor concentration, npj Climate and Atmospheric Science, 5, 95, 10.1038/s41612-022-00321-y, 2022.

Chow, F. K., Yu, K. A., Young, A., James, E., Grell, G. A., Csiszar, I., Tsidulko, M., Freitas, S., Pereira, G., Giglio, L., Friberg, M. D., and Ahmadov, R.: High-Resolution Smoke Forecasting for the 2018 Camp Fire in California, Bulletin of the American Meteorological Society, 103, E1531-E1552, https://doi.org/10.1175/BAMS-D-20-0329.1, 2022.

Guenther, A. B., Jiang, X., Heald, C. L., Sakulyanontvittaya, T., Duhl, T., Emmons, L. K., and Wang, X.: The Model of Emissions of Gases and Aerosols from Nature version 2.1 (MEGAN2.1): an extended and updated framework for modeling biogenic emissions, Geosci. Model Dev., 5, 1471-1492, 10.5194/gmd-5-1471-2012, 2012.

Hu, L., Millet, D. B., Baasandorj, M., Griffis, T. J., Turner, P., Helmig, D., Curtis, A. J., and Hueber, J.: Isoprene emissions and impacts over an ecological transition region in the U.S. Upper Midwest inferred from tall tower measurements, J. Geophys. Res. Atmos., 120, 3553-3571, https://doi.org/10.1002/2014JD022732, 2015.

Jin, L., Permar, W., Selimovic, V., Ketcherside, D., Yokelson, R. J., Hornbrook, R. S., Apel, E. C., Ku, I. T., Collett Jr, J. L., Sullivan, A. P., Jaffe, D. A., Pierce, J. R., Fried, A., Coggon, M. M., Gkatzelis, G. I., Warneke, C., Fischer, E. V., and Hu, L.: Constraining emissions of volatile organic compounds from western US wildfires with WE-CAN and FIREX-AQ airborne observations, Atmos. Chem. Phys., 23, 5969-5991, 10.5194/acp-23-5969-2023, 2023.

Ketcherside, D. T., Miller, D. D., Kenerson, D. R., Scott, P. S., Andrew, J. P., Bakker, M. A. Y., Bundy, B. A., Grimm, B. K., Li, J., Nuñez, L. A., Pittman, D. L., Uhlorn, R. P., and Johnston, N. A. C.: Effects of Wildfire Smoke on Volatile Organic Compound (VOC) and PM2.5 Composition in a United States Intermountain Western Valley and Estimation of Human Health Risk, Atmosphere, 15, 1172, 2024.

McFiggans, G., Mentel, T. F., Wildt, J., Pullinen, I., Kang, S., Kleist, E., Schmitt, S., Springer, M., Tillmann, R., Wu, C., Zhao, D., Hallquist, M., Faxon, C., Le Breton, M., Hallquist, Å. M., Simpson, D., Bergström, R., Jenkin, M. E., Ehn, M., Thornton, J. A., Alfarra, M. R., Bannan, T. J., Percival, C. J., Priestley, M., Topping, D., and Kiendler-Scharr, A.: Secondary organic aerosol reduced by mixture of atmospheric vapours, Nature, 565, 587-593, 10.1038/s41586-018-0871-y, 2019.

Selimovic, V., Ketcherside, D., Chaliyakunnel, S., Wielgasz, C., Permar, W., Angot, H., Millet, D. B., Fried, A., Helmig, D., and Hu, L.: Atmospheric biogenic volatile organic compounds in the Alaskan Arctic tundra: constraints from measurements at Toolik Field Station, Atmos. Chem. Phys., 22, 14037-14058, 10.5194/acp-22-14037-2022, 2022.

Taipale, R., Ruuskanen, T. M., Rinne, J., Kajos, M. K., Hakola, H., Pohja, T., and Kulmala, M.: Technical Note: Quantitative long-term measurements of VOC concentrations by PTR-MS – measurement, calibration, and volume mixing ratio calculation methods, Atmos. Chem. Phys., 8, 6681-6698, 10.5194/acp-8-6681-2008, 2008.

Wang, N., Zannoni, N., Ernle, L., Bekö, G., Wargocki, P., Li, M., Weschler, C. J., and Williams, J.: Total OH Reactivity of Emissions from Humans: In Situ Measurement and Budget Analysis, Environ. Sci. Technol., 55, 149-159, 10.1021/acs.est.0c04206, 2021.

Worton, D. R., Moreno, S., O'Daly, K., and Holzinger, R.: Development of an International System of Units (SI)-traceable transmission curve reference material to improve the quantitation and comparability of proton-transfer-reaction mass-spectrometry measurements, Atmos. Meas. Tech., 16, 1061-1072, 10.5194/amt-16-1061-2023, 2023.

Yokelson, R. J., Christian, T. J., Karl, T. G., and Guenther, A.: The tropical forest and fire emissions experiment: laboratory fire measurements and synthesis of campaign data, Atmos. Chem. Phys., 8, 3509-3527, 10.5194/acp-8-3509-2008, 2008.

---

## Author Comment (AC2)

**Extreme Heat and Wildfire Emissions Enhance Volatile Organic Compounds: Insights on Future Climate**

Christian Mark Salvador et al.

We are grateful for the efforts of the Reviewer of our study. The comments of the Referee substantially improved our study, and the authors addressed each statement carefully. Our point-by-point responses to the Reviewer's general and specific comments are presented below. The referee's comments are in **black,** and our answers are in red. Modified or new statements integrated into the revised manuscript are indented. All changes can be seen in the revised version of the manuscript in red font

**Reviewer #1**

**General Comments (GC)**

The authors reported the volatile organic compound (VOC) concentrations measured at a temperate forest site. The research highlights the impact of high temperatures and wildfires on VOC concentrations from various sources. This topic is important due to the increasing frequency of heatwaves and fire events, and it fits within the scope of ACP. The results from this study are interesting and could contribute to understanding the impact of these events on air quality and climate. The paper is generally well-written, and the analysis conducted is reasonable and solid. However, the manuscript still requires additional adjustment in the format and clarification before it is ready for publication.

**Response:** The authors appreciate the kind words from the reviewer.

**GC 1:** According to the requirements of ACP (https://www.atmospheric-chemistry-and-physics.net/submission.html#figurestables), figure panel labels should be included. This is currently not the case throughout the manuscript, including the figures in the supplementary materials.

**Response:** Done. All figures with panels, including in the supplementary file, have been modified with alphabetically labeled panels. The corresponding captions were also modified according to the changes in the figures.

**GC 2:** The method section (Section 2.3) is not clear enough. The content in Lines 173-177 should be expanded with more detailed explanations. For example, the authors mentioned, "The NNMF was applied for a 10-factor series with 30 replicates, 1000 iterations, and a multiplicative update algorithm," which is too vague for readers.

**Response:** We added the following statements in section 2.3 to explain the parameters and algorithm used in the NNMF procedure.

> *Replicates are the number of times the program will perform the factorization, with every replicate starting with random values for W and H. Iteration is the input value for the maximum replicate performed in the optimization for convergence purposes. The NNMF at Matlab can be performed either as alternating least square (als) or multiplicative update algorithm (mult). This study implemented the mult factorization algorithm as it has faster iterations and is more sensitive to starting values.*

**GC 3:** Some content from the first paragraph of Section 3.4 should be placed in the methods section, as it should not be mixed with the results.

**Response:** Done. We appreciate the reviewer for raising the concern. We moved the following statements from section 3.4 to section 2.2 in the methodology.

*The volatility of the extended list of VOCs was assessed by estimating the effective saturation mass mixing ratio ($C_{sat}$). The parameterization of the volatility, based on the number of carbon, oxygen, and nitrogen atoms (Donahue et al., 2011; Mohr et al., 2019), was calculated using the following equation:*

$$log(C_{sat}) = (n_{O*} - n_C)b_C - (n_O - 3n_N)b_O - 2\frac{(n_O - 3n_N)n_C}{(n_C + n_O - 3n_N)}b_{CO} - n_N b_N \qquad (3)$$

*where $n_{0*} = 25$, $b_C = 0.475$, $b_O = 0.2$, $b_{CO} = 0.9$, and $b_N = 2.5$. The terms $n_C$, $n_O$, and $n_N$ are the number of carbon, oxygen, and nitrogen atoms, respectively.*

**Specific Comments (SC)**

**SC 1:** Line 231: When monoterpene shows a peak during the day, it implies that the emission of monoterpenes from plants is also light-dependent, similar to isoprene (e.g., Kuhn et al., 2004). This is not necessarily related to the isomers of monoterpenes.

**Response:** Recent studies have shown that isomers of the monoterpenes can be categorized as light-dependent or independent. In an extensive measurement in a coniferous forest in Germany, isomers of monoterpene showed different diurnal profiles. Both pinene isomers (α- and β-) recorded daytime peaks while sabinene had an apparent enhancement during nighttime as shown in Figure R1 (Borsdorf et al., 2023). This observation was consistent across different months/seasons of the year.

[Figure]

**Figure R1.** Seasonal and diurnal of isoprene and monoterpenes in a coniferous forest in Germany. Adapted from: Borsdorf, H., Bentele, M., Müller, M., Rebmann, C., and Mayer, T.: Comparison of Seasonal and Diurnal Concentration Profiles of BVOCs in Coniferous and Deciduous Forests, Atmosphere, 14, 1347, 2023.

Similar results were obtained in a Forest Research Lab in Virginia, USA where pinene isomers had minima during the daytime while sabinene had a clear peak around 15:00 (Mcglynn et al., 2023). Evidently, isomers of monoterpene had different diurnal profiles, which can be utilized as identifiers. Our measurements at a temperate forest in the Midwestern USA indicated elevated monoterpene concentrations during daytime, which implies that the dominant monoterpene might be sabinene and/or ocimene.

**SC 2:** Line 274: Align the paragraph.

**Response:** Done. The paragraph was aligned.

**SC 3.1:** Line 289: It is well-known that BVOC emissions increase exponentially with temperature. Why did you choose linear regression here? Could you try using the traditional exponential equation to fit the data?

**Response:** We thank the reviewer for the suggestion. The temperature-BVOC relationships were fitted with the exponential equation. The following statements were added to the main text.

> *The major BVOCs, isoprene and monoterpene, responded well to variations in temperature, as shown in Figure 3. Under extreme temperatures conditions, the isoprene and monoterpene mixing ratios were 23 and 0.32 ppb, respectively, which were three times higher than the concentrations observed at temperatures below 32°C. The enhancement of isoprene and monoterpene also increased the reactivity of the atmosphere in the temperate forest, based on the calculated 8.31 $s^{-1}$ increase in OH reactivity. Furthermore, Figure 4 shows the evident exponential relationship between temperature and the major BVOCs, consistent with previous studies (Hu et al., 2015; Selimovic et al., 2022; Guenther et al., 2012). The empirically determined coefficients (β) for isoprene and monoterpene are 0.13 and 0.12.*

**SC. 3.2** One interesting point I noticed in Figure 3 is that monoterpenes respond differently across different temperature ranges. This might be related to the stress response of monoterpenes. For example, monoterpene emissions may increase dramatically after surpassing a certain temperature threshold (Nagalingam et al., 2023). Therefore, I wonder whether the varying responses observed here are indicators of a stress response in plants or simply an artificial effect caused by the choice of fitting equation.

**Response:** Indeed, plants have mechanisms to adapt to heat stress and the emission of BVOCs is one of the physiological responses to mitigate heat stress. We agree with the reviewer that the varying responses of monoterpene to different temperature ranges in our temperate forest might be linked to plant thermotolerance.

We added the following statements to the main text.

> *The non-linear response, particularly the enhanced emission of monoterpene at elevated temperatures, can be linked to the thermotolerance physiological activities of plants during heat stress events. A previous study indicated heat stress evidently enhanced the emission of monoterpenes from sunflower, western redcedar, and American sweetgum by at least a factor of 22 (Nagalingam et al., 2023).*

**SC 4:** Line 364-368: From the perspective of ozone formation, the interactions between fire plumes and BVOCs could be more significant. Since rural regions are usually VOC-limited due

to the lack of NOx, the transportation of fire plumes could bring NOx or PANs to the site and promote ozone formation (Xu et al., 2021). In this case, the increase in benzene is not a key factor for ozone formation at this site, which is abundant in isoprene. Additionally, even though the benzene concentration increased significantly, I wonder if the OFP of benzene could be as high as that of isoprene. I suggest either removing this part or providing a more comprehensive discussion.

**Response:** We agree with the reviewer that benzene might not be the primary contributor to ozone formation, even during the fire plume transport. The section, including the calculation of OFP in the supplementary material, was removed.

**REFERENCES**

Borsdorf, H., Bentele, M., Müller, M., Rebmann, C., and Mayer, T.: Comparison of Seasonal and Diurnal Concentration Profiles of BVOCs in Coniferous and Deciduous Forests, Atmosphere, 14, 1347, 2023.

Guenther, A. B., Jiang, X., Heald, C. L., Sakulyanontvittaya, T., Duhl, T., Emmons, L. K., and Wang, X.: The Model of Emissions of Gases and Aerosols from Nature version 2.1 (MEGAN2.1): an extended and updated framework for modeling biogenic emissions, Geosci. Model Dev., 5, 1471-1492, 10.5194/gmd-5-1471-2012, 2012.

Hu, L., Millet, D. B., Baasandorj, M., Griffis, T. J., Turner, P., Helmig, D., Curtis, A. J., and Hueber, J.: Isoprene emissions and impacts over an ecological transition region in the U.S. Upper Midwest inferred from tall tower measurements, J. Geophys. Res. Atmos., 120, 3553-3571, https://doi.org/10.1002/2014JD022732, 2015.

McGlynn, D. F., Frazier, G., Barry, L. E. R., Lerdau, M. T., Pusede, S. E., and Isaacman-VanWertz, G.: Minor contributions of daytime monoterpenes are major contributors to atmospheric reactivity, Biogeosciences, 20, 45-55, 10.5194/bg-20-45-2023, 2023.

Nagalingam, S., Seco, R., Kim, S., and Guenther, A.: Heat stress strongly induces monoterpene emissions in some plants with specialized terpenoid storage structures, Agricultural and Forest Meteorology, 333, 109400, https://doi.org/10.1016/j.agrformet.2023.109400, 2023.

Selimovic, V., Ketcherside, D., Chaliyakunnel, S., Wielgasz, C., Permar, W., Angot, H., Millet, D. B., Fried, A., Helmig, D., and Hu, L.: Atmospheric biogenic volatile organic compounds in the Alaskan Arctic tundra: constraints from measurements at Toolik Field Station, Atmos. Chem. Phys., 22, 14037-14058, 10.5194/acp-22-14037-2022, 2022.

---

## Referee Report (RR1)

**General Comments**

Thanks to the authors for their efforts in addressing my concerns as well as those raised by the other reviewer. The manuscript quality has improved following the authors' revisions, and many changes have been incorporated into the updated manuscript. However, there are still some remaining issues that need to be addressed.

- 1. I think the current Introduction is quite scattered and lacks focus. The topic of this paper is the impact of heatwaves and wildfires on VOC concentrations. Because the measurement site is a rural temperate mixed forest, one focus of the paper is biogenic VOCs, and the other is wildfires. However, the authors did not clearly present what has already been investigated or identify the specific knowledge gaps their paper could address. For instance, BVOCs are temperature-dependent, which is why understanding the impact of high temperatures on BVOCs is important. Additionally, two field campaigns focusing on BVOCs have previously been conducted at the MOFLUX site (Potosnak et al., 2014, and Seco et al., 2015). However, the authors did not include this background information in their Introduction. The same issue applies to their discussion of wildfires.
- 2. I don't think the definition of an extreme temperature threshold as 32 °C is appropriate. A temperature of 32 °C is quite common. I understand that temperature conditions can vary significantly by region. Typically, heatwave studies define thresholds using the 95th or 99th percentile of historical temperature records. Therefore, I suggest the authors use the 95th percentile of hourly July temperatures over the past 10 years as the threshold for defining heatwayes.
- 3. The authors insisted that "Regions dominated by emissions of  $\alpha$ -pinene,  $\beta$ -pinene, and limonene typically have a nighttime peak, whereas daytime enhancements are observed for areas with sabinene and ocimene." I cannot agree with this statement. The diurnal cycles of monoterpenes largely depend on whether their emissions are light-dependent. De novo monoterpenes typically peak during the day because their emissions are driven by light emissions (Jardine et al., 2015). Only temperature-dependent monoterpenes consistently show nighttime peaks due to the lower boundary layer height at night. The monoterpenes mentioned by the authors can be either light-dependent or only temperature-dependent. For example,  $\alpha$ -pinene emissions from Scots pine is light-dependent (Tarvainen et al., 2005). Furthermore, the paper (Borsdorf et al., 2023), which the authors cited in response to my comment, also mentioned that "The atmospheric concentrations of monoterpenes, which are emitted in a temperaturedependent manner, are often greatest during nighttime hours due to the shrinking of the planetary boundary layer," and also noted that "Sabinene is obviously emitted in a light-dependent manner comparable with isoprene, while all the other monoterpenes are emitted by volatilization." Therefore, assigning diurnal emission patterns based on monoterpene isomer type is not appropriate, as each isomer may exhibit either emission pathway.

- 4. Figures 2 and 3 contain obvious errors. Figure 2 lacks a label for the first panel and has no axis labels for panel (C). In Figure 3, the standard deviations shown in the histogram plots appear too small compared to the actual distribution of data points displayed on the right side of the figure.
- 5. According to the NNMF results, does this imply that there is always a fire plume near the site? Lines 501–503 state: "oxygenated hydrocarbons (CxHyOw) and CxHyNz compounds were persistent in the early hours of the combustion plume." However, as the second reviewer mentioned, "VOCs become more oxygenated as they age away from the biomass burning source," which does not support the statement made here.

**Specific comments:**

- Figure 1. The term "Global solar radiation" used here is incorrect; the authors should use "Downward solar radiation" instead. In addition, UV refers sto ultraviolet radiation, which is a portion of the whole solar radiation spectrum. Based on the values provided (around 1000 W m-2), I believe the authors are referring to total solar radiation rather than UV radiation, as such a high UV radiation level would not be survivable for any life forms. In addition, did author consider the smoke impact on BVOC emission during by affecting the solar radiation?
- Line 252: The word "global budget" is not accurate. Guenther et al. (2012) calculated the emission, which is just part of the "global budget".
- Line 322: What kind of vegetation's physiological functions?
- Line 354: I think that biogenic VOCs are temperature dependent.

**Reference**

- Borsdorf, H., Bentele, M., Müller, M., Rebmann, C., and Mayer, T.: Comparison of Seasonal and Diurnal Concentration Profiles of BVOCs in Coniferous and Deciduous Forests, Atmosphere, 14, 1347, 2023.
- Jardine, A. B., Jardine, K. J., Fuentes, J. D., Martin, S. T., Martins, G., Durgante, F., Carneiro, V., Higuchi, N., Manzi, A. O., and Chambers, J. Q.: Highly reactive light-dependent monoterpenes in the Amazon, Geophysical Research Letters, 42, 1576-1583, https://doi.org/10.1002/2014GL062573, 2015.
- Tarvainen, V., Hakola, H., Hellén, H., Bäck, J., Hari, P., and Kulmala, M.: Temperature and light dependence of the VOC emissions of Scots pine, Atmos. Chem. Phys., 5, 989-998, 10.5194/acp-5-989-2005, 2005.

---

## Referee Report (RR2)

**General Summary:**

The work presented here reports the change in mixing ratio and distribution of several biogenic and anthropogenic VOC measured in a forest, as a response to increased temperature and transported biomass burning plumes. The authors emphasize the variability of VOCs and associated reactivity due to heat and smoke. This study investigating how the VOC distribution changes in response extreme events is highly relevant, and the authors have done quite a bit of work to improve upon the analysis from initial submission. The appropriate temperature responses have been added, and a section on reactivity has been included to expand the discussion of VOC variability and reactivity under extreme conditions. There are some minor adjustments that need to be made, but overall the reviewer is pleased with the additions and corrections the authors have made to help improve the manuscript. I recommend publication of the manuscript, after these small issues (addressed below) are rectified or clarified.

**Minor comments:**

I recommend abstract explicitly state the estimated and increased OH reactivity due to these exceptional events. This is a significant and informative addition that is highly relevant. The authors should do well to emphasize this!

Figure 1 is missing (A), Figure 1C is missing a label for the x-axis, not sure what this is representing?

L285 to L287: Minor point, but I would caution against the authors using "large" to broadly refer to particles >50 nm. Based on the geometric mean diameter shown in the supplementary information Fig. S2B, the particles fall into the accumulation mode and are technically "fine mode" aerosol. Large particles typically refer to particles in the coarse mode (e.g. >2500 nm or 2.5 microns). Suggest removing 'large' in lines 285 to 287, and instead simplify to say "...Particles >50 nm in diameter were observed..." and "...for the presence of these particles..." You could also consider stating the average (+/-) geometric mean of particles throughout the study for reference.

L381-383: It's both interesting and intriguing that catechol also has a sharp peak after 6 PM in the diurnal cycle shown in Fig 2M. Is there some process responsible for this, or is that just an artifact from data averaging smoke impacted days? The lines here mention that it increased significantly to 300 ppt on some days, so it could just be that the timing of the smoke impact corresponded with that sharp increase?

L388: Make sure to specify/reiterate that smoke concentration is derived from HRRR. (E.g. "Figure 4 shows the HRRR-derived smoke concentration..."

Figure 4B: It's difficult to read/see the trajectory as an inset in the figure. Is it possible to make it its own separate panel, or move it to the supplement as part of figure S5?

L437-438: Are there units on this value of 8.5? Is this different from Csat?

L450-451: Presumably "low temperature" means <32 C and extreme is >32 C? Suggest defining/stating this in parenthesis next to the statements to prevent ambiguity. Is the low temperature and corresponding reactivity in reference to the selected time period (July 8 to 17) or is that the average for non-impacted/non-extreme temp outside of that date range as well? Please clarify.

L450-L453: The additional reactivity calculations do a lot to help develop the work and associated impacts! These calculations are a highly useful addition. However, I think this section could be worked on a bit to provide more clarity and help emphasize the significance of what is shown. Is it possible to include OH reactivity during "background" or typical conditions during times not influenced by temperature or combustion? Looks like July 9th to 10th could be a possible time-frame that would work for this? L443 earlier states that average reactivity between July 8 to July 17 was 91.30, but this average would presumably include days that experienced elevated temperature and BB impacts. What does the reactivity look like outside of these times? This would provide a useful benchmark for determining exactly how much the reactivity increased due to these extreme events. If it's not possible to calculate yourself, is there a study you can cite that might provide an estimate? I'm also confused as to how it's possible that the reactivity at low temperature (however that is defined) is higher than the average for this entire "impacted" period (98.92 low temp versus 91.30 s-1 average July 8 to July 17)? Clearer wording and phrasing throughout the paragraph would strengthen this finding.

L451: It would be useful to compare your estimates of OH reactivity to other studies (SOAS, CalNEX, WE-CAN, etc.), for context.

L453, & Figure 5: One thing not emphasized in the text that I think would benefit from more attention is that according to L453 and Figure 5, even though total VOC was lower during the BB event compared to extreme heat, total volatility increased (presumably due to presence of BB compounds like benzene and HCN as mentioned in text). Total reactivity under BB was comparable to reactivity under high temp conditions without any BB influence. Given the lower apparent tVOC, this really highlights how reactive BB gas phase species are! This should be emphasized, especially when considering the projected increase in BB and extreme temp events in the future, and also begs the question—under these future climate scenarios what will have more of an influence in this kind of environment: enhanced temperature, or enhanced BB?

L466-L470: Do you have data that you can add here to support this? (E.g % change enhancement due to increased temperature, or make a table in the supplement). Without the values there the reader is left to take the author's word for it.

Figure 6: The CxHyNz category is a bit washed out and difficult to see. Can you make it a different color or make it darker? Same for the error shading in diurnal plot Figure 6C.

L496: Define what a substantial number is. N=?

L504-505: Can you remind the reader of the significance of LogCsat >6? What do values higher than this imply? Reiterate the message you want to convey.

L512-513: "This is unlikely due to the expected biogenic emission in the forest, although it came second with 34% contribution compared to 66% from the H-BB factor." This sentence is worded strangely... Does "this" refer to the preceding sentence describing monoterpene and its fragment? Is "it" also monoterpene and its fragment? Not sure what This/it refers to here, can you please clarify?

L514-L515: Hmmm, are BB events considered anthropogenic? I think this is the subject of ongoing debate and is a hot topic! The answer depends on where you are in the world. Suggest slight rephrasing to ... originate from anthropogenic sources and BB events.

L525-526: Do you have a reference to support nighttime oxidation of NO3 contributing to the formation of these species?

L528-529: "During the transport of BB plumes, the secondary factor had a relatively low increase in signal compared to both BB factors, which shows that oxidation compounds were generally locally...." I don't think this is true, given that one of your factors is literally called, "oxygenated BB", which implies that there was indeed some contribution from long-range transport... I think maybe what you mean to say is, "the secondary factor had... low increase... showing that secondary formation was predominantly locally generated..."

L530-531: Re-emphasize the change to chemistry during the BB impact to strengthen your messaging. What was the increase in reactivity during BB events compared to non-BB events (see earlier comment)? Add it here.

L533: What is meant by a "critical" VOC? Suggest removing this adjective and just saying, "VOCs have important contributions to several..."

L536: Change to "anthropogenic and fire emissions". Anthropogenic emissions typically refer to things like traffic, shipping, home heating, etc. While it's true that a significant portion of wildfires are started by humans, that's not always the case. Factors contributing to fire ignition are dependent on location and time of year, among others.

L538-539: Some of these are also sourced from anthropogenic emissions (e.g. benzene, toluene), so I suggest removing "in the forest" after typical VOCs, especially when they are referenced as AVOCs later in the conclusions (L545). Add the error to the average total mixing ratio reported at the end of the line in L539.

L541-L542: Add the uncertainty or standard deviation with these averages.

L557-L558: "...Increases could be significantly affected regionally by accompanying changes in atmospheric circulation." I'm not sure what this means or is supposed to imply. Are you saying that the extent of predicted increases could be regionally impacted due to changes? Suggest rewording for clarity.

L565: Not sure what a typical VOC is. I think you mean to say background/typical conditions? "Typical VOC" likely changes based on location and environment.

L567: Important to note that this is for all of the measured VOCs\*. Estimates could vary if additional species are measured to include those beyond the list of 250+ you have.

L568: But you had a whole factor named "oxygenated BB" that showed enhancement so it's not clear to me how it's fair to say, "During BB plume transport, less oxygenated compounds were enhanced..."?

---

## Editor Decision (ED1)

The authors have an important dataset that captures changes in emissions sources and meteorological conditions in a location that has seen previous study. The results provide some insight in changes in atmospheric composition and reactivity driven by changes in source and/or meteorological conditions that generally contribute to our understanding of atmospheric chemistry. The methods are appropriate, with some minor comments and suggestions given below. These strengths demonstrate that this manuscript is appropriate for consideration as a measurement report in ACP. Following minor comments and suggestions on the methods are broader scientific comments and concerns that should be address in the revision stage. Some of these comments and concerns will naturally resolve with the change in format for manuscript type, and a tighter focus on the campaign and results.

**Minor Comments (with a focus on Methods)**

Line 89-Several **hundreds** of compounds have been identified in wildfire smoke (see for example Hatch et al. https://doi.org/10.5194/acp-15-1865-2015, Koss et al. https://doi.org/10.5194/acp-18-3299-2018, Selimovic et al. https://doi.org/10.5194/acp-18-2929-2018, Binte-Shahid et al. https://doi.org/10.5194/gmd-17-7679-2024).

Line 94-While it is true that there are significant uncertainties due to the huge variability inherent in fires and emissions, there are a lot of studies that have looked at the influence of wildfire emissions on atmospheric chemistry and reactivity that should be cited here (see for example Gilman et al. doi:10.5194/acp-15-13915-2015, Liu et al. doi/10.1002/2016JD025040, Kumar et al. <a href="https://www.nature.com/articles/s41598-017-19139-3">https://www.nature.com/articles/s41598-017-19139-3</a>, Permar et al. <a href="https://pubs.rsc.org/en/content/articlehtml/2023/ea/d2ea00063f">https://pubs.rsc.org/en/content/articlehtml/2023/ea/d2ea00063f</a>).

It might be useful to include a map of the measurement location, particularly showing the influence of biogenic emissions and smoke transport during the study period.

It is recommended to review the first sentences of the paragraph starting on line 99-some of these details would be more appropriate in the methods.

Line 111-why are vehicle emissions expected to persist in the forest?

Line 144-while it makes some sense to use the HRRR forecasts to identify periods with smoke influence (which need to be more clearly defined in the methods-i.e., what periods met the established threshold?), it would strengthen the paper to use measurements to confirm impacts of smoke during these periods (e.g., using PM2.5 and/or CO data, or even acetonitrile or other known fire tracers measured during the campaign).

There are a lot of details about operation of the PTRMS that could be moved to the SI (e.g., equations 1-3 and associated text). The authors may want to include additional references

on the use of PTRMS for measurement of monoterpenes and known challenges/interferences if relevant.

It is not clear what value was used for the reaction rate constant of the summed monoterpenes with OH and how that was calculated or what assumptions were made.

Line 430-log saturation "mixing ratio" should be log saturation vapor concentration

**Broader Comments on Science and Scope**

The mechanisms by which climate change and climate extremes affect biogenic, anthropogenic, and wildfire emissions are very different, and that complexity is not clearly or adequately addressed in the manuscript as currently written. For example, in the abstract lines 17-18 describe the "...response of VOCs to future conditions such as extreme heat and wildfire events...". It is well documented that biogenic emissions can be temperature dependent, with compound- and species-specific differences. It is also well documented that some anthropogenic emissions can also be temperature dependent, e.g., asphalt emissions, but that strong temperature dependence is typically more indicative of biogenic emissions. In addition, reaction product formation can also have some dependence on temperature (as temperature can influence oxidant levels and reaction rates). This is addressed to some extent in section 3.2, but some refinements are needed. 1-The authors state the at the BVOCs "respond well" to variations in temperature. What does this mean? They behave as expected/consistent with other studies? 2-It appears the isoprene and monoterpenes have the same exponential response to temperature. I do not think that this is completely consistent with other studies (or for example the temperature response as would be predicted in a biogenic emissions model), as isoprene typically shows a much stronger response to temperature. There may be some studies that show very strong dependence of monoterpene emissions in extreme temperatures, but this is not clear as this section is currently written. It is suggested that the temperature sensitivity of isoprene and monoterpenes be discussed separately in this section, including context from references that are specific to isoprene and to monoterpenes. The authors state that the enhanced emissions of monoterpenes "can also be linked to", suggesting that at least one other mechanistic reason for enhanced emissions has already been discussed, but this is not the case. One suggestion is to more generally summarize (with appropriate citations) the reasons for enhanced BVOC emissions with increased temperatures (and not to get into any specific mechanisms). 3-The discussion of MVK and MACr is confusing as written. The discussion seems to be mixing the effects of temperature on emissions of isoprene (and therefore its oxidation products) with the effects of temperature on lifetime (which could alter the ratio as a function of temperature). These effects need to be more clearly differentiated in this

section and this distinction should be considered throughout the manuscript when discussing temperature effects particularly when the compounds can be oxidation products. 4-Toluene is also emitted from biogenic sources. The discussion around toluene (including attributing everything to interference/fragmentation of monoterpenes) needs to be reconsidered and revised accordingly in this context. 5-"Combustion" is used to describe anthropogenic emissions (as in this section) and wildfire emissions which is confusing and should be revised throughout. It is more typical to refer to wildfire emissions as wildfire emissions, biomass burning emissions, or pyrogenic emissions. It is suggested to choose one of those more commonly used terms and use it throughout (right now there is some use of wildfire and biomass burning but it is inconsistent). Back to the line in the abstract, wildfire emissions don't "respond" to temperature in the same way that biogenic emissions or anthropogenic emissions do. There are complex relationships between ambient temperature and fuel moisture, and therefore fire ignitions, fire spread, fire severity, fuel consumption, etc. that affect emissions. This complexity is not acknowledged in the manuscript (e.g., by saying that wildfire emissions "respond" to temperature) and the rationale for discussing the wildfire emissions in the context of temperature is not well supported. A related comment, on line 193, it is stated that acetonitrile and catechol don't follow the trend of temperature (presumably this means they do not increase with temperature like the BVOCs), which is not unexpected based the complexities noted above. However, in line 394 it is concluded that this is in fact because of the "infrequent emissions of BB plumes". It is not clear what is meant by that.

As noted above, the discussion of wildfire emissions is not sufficiently differentiated from anthropogenic emissions throughout the manuscript. This leads to some confusion about the results and ultimately, the implications of this work. In addition, "monoterpene" is used throughout and described on line 285 as being composed of several organic species. This is not accurate or consistent with existing literature. Monoterpene specifically describes a class of organic compounds with the formula C10H16, of which there are several isomers (some of which are listed on line 285). The analytical technique being used can't differentiate isomers (as noted in the methods), so what is being reported is the sum of monoterpenes (plural) and that needs to be clearer. The monoterpenes in that mixture can have very different reactivities, lifetimes, etc. and the manuscript should be revised with this in mind. For example, what was assumed for the OH reactivity of the monoterpene mixture?

The manuscript has a lot of detail that does not necessarily support or benefit from the measurements and results presented. For example, there is scattered mention of climate, climate forcing, future climate scenarios, and aerosols (including new particle formation). (One of the reviewers notes this of the introduction.) The connections between emissions

composition, chemistry, aerosols, and climate could be made more generally, but the repeated mention does not strengthen the manuscript since the measurements and results are not clearly and sufficiently linked to these complex processes. Refining the focus to the measurements and results, including placing them in the context of similar studies, will help improve this aspect of the manuscript.

---

## Author Response (AR2)

**Extreme Heat and Wildfire Emissions Enhance Volatile Organic Compounds: Insights on Future Climate**

Christian Mark Salvador et al.

We appreciate the referee's detailed review of our manuscript, and we provided here a detailed response that addresses the referee's concern. Our point-by-point responses to the Reviewer's general and specific comments are presented below. The referee's comments are in black, and our answers are in red. Modified or new statements integrated into the revised manuscript are indented. All changes can be seen in the revised version of the manuscript in red font.

**Response to Reviewer 1**

**Major Comment:**

Thanks to the authors for their efforts in addressing my concerns as well as those raised by the other reviewer. The manuscript quality has improved following the authors' revisions, and many changes have been incorporated into the updated manuscript. However, there are still some remaining issues that need to be addressed.

**Response:** We thank the reviewer for the thoughtful feedback and for acknowledging the authors' efforts in revising the manuscript. The reviewers greatly contributed to enhancing the quality and clarity of our work. In this version of the manuscript, we have thoroughly addressed the remaining issues based on the reviewers' suggestions.

Comment 1: I think the current Introduction is quite scattered and lacks focus. The topic of this paper is the impact of heatwaves and wildfires on VOC concentrations. Because the measurement site is a rural temperate mixed forest, one focus of the paper is biogenic VOCs, and the other is wildfires. However, the authors did not clearly present what has already been investigated or identify the specific knowledge gaps their paper could address. For instance, BVOCs are temperature dependent, which is why understanding the impact of high temperatures on BVOCs is important. Additionally, two field campaigns focusing on BVOCs have previously been conducted at the MOFLUX site (Potosnak et al., 2014, and Seco et al., 2015). However, the authors did not include this background information in their Introduction. The same issue applies to their discussion of wildfires.

**Response:** We concur with the reviewer. The third paragraph of the introduction section was modified to offer an overview of volatile organic compounds (VOCs), their atmospheric reactivity, and response to future climate scenarios. The initial introduction also identified knowledge gaps, emphasizing the need to understand the influence of extreme temperatures and wildfires on VOC emissions. To explicitly highlight this scientific information and address these research questions, we have added several statements in the third paragraph. The new statements discuss the exponential dependence of biogenic VOCs (BVOCs) on temperature, the typical VOCs emitted during wildfire activities, and the subsequent impact of the enhanced concentration of these VOCs on atmospheric chemical reactivity. The new third paragraph now reads:

Among the chemical components of the atmosphere, the abundance of volatile organic compounds (VOCs) is expected to respond to extreme heat and wildfire emissions. VOCs, particularly the unsaturated compounds, interact with oxidants such as hydroxyl (OH) and nitrate (NO3) radicals, which subsequently create ozone and oxidized molecules (Hakola et al., 2012; Ramasamy et al., 2016; Spirig et al., 2004; Vermeuel et al., 2023). Further reaction products such as highly oxidized molecules also

participate in the formation of particles that subsequently act as cloud condensation nuclei (Chen et al., 2022; Hallquist et al., 2009). The emission and transformation of VOCs highly depend on environmental parameters such as temperature, relative humidity, and solar radiation. For instance, biogenic volatile organic compounds (BVOCs) exhibit an exponential temperature dependence, whereby an increase in temperature accelerates both their production and release from plant tissues. (Guenther et al., 1993; Rinnan et al., 2020). However, the degree of changes under future climate is still uncertain (i.e., suppression or enhancement) (Daussy and Staudt, 2020). A global estimate of isoprene emissions with temperature and land-cover drivers under future scenario (year: 2070-2099) was 889 Tg yr-1, substantially higher compared to that expected using current climatological and land-cover conditions (522 Tg yr-1) (Wiedinmyer et al., 2006). Moreover, CO2, which is expected to rise in the future climate, can substantially decrease the emission of isoprene from vegetation (Lantz et al., 2019a). On the other hand, empirical results and modeling efforts suggest that future elevated temperatures could suppress the impact of CO2 on isoprene emissions, thus increasing the uncertainty of future climate's influence on the emission of isoprene (Lantz et al., 2019b; Sahu et al., 2023). Moreover, BB events such as wildfires are considered as the second-largest source of VOCs globally, further influencing air quality and climate (Jin et al., 2023; Yokelson et al., 2008). During wildfire events, the combustion of vegetation and other biomass induce the pyrolysis of plant materials which ultimately release several VOCs during the process (Ciccioli et al., 2014). Typical VOCs emitted from wildfires include acrolein, acetonitrile, pyrrole, styrene, guaiacol, toluene, phenol, and catechol (Liang et al., 2022; Jin et al., 2023). Benzene, a common compound emitted during wildfire events, has been found to be more than ten times the typical concentration in metropolitan areas, thereby posing elevated health risks (Ketcherside et al., 2024). Beyond their health impacts, the emitted reactive carbon- and nitrogen-containing compounds can significantly alter several critical atmospheric processes, including ozone formation and particle formation events. However, the relative importance and contribution of VOCs from wildfire activities to atmospheric reactivity remain uncertain. A comprehensive understanding of the interactions between future abiotic factors and VOC emissions is essential for accurately predicting future air quality and climate scenarios.

The initial version of the manuscript included information regarding results from previous measurements at the MOFLUX site. Some of these statements were moved to the introduction section to provide more background information about the site. The fourth paragraph now reads:

The Ozark Plateau (Wiedinmyer et al., 2005), and this site in particular, is a known hotspot for emissions of BVOCs such as isoprene and monoterpene. Drought is a critical event at MOFLUX, as such environmental stress induced the highest ecosystem isoprene emission ever recorded for a temperate forest in 2011 (53.3 mg m $^{-2}$  h $^{-1}$ ) (Potosnak et al., 2014). Field measurement campaign in 2012 in MOFLUX reported isoprene reaching a maximum concentration of 28.9 ppbv, while monoterpenes peaked at 1.37 ppbv over half-hour intervals (Seco et al., 2015). Moreover, the site is about 5 km away from a major highway, thus anthropogenic VOCs (AVOCs) such as benzene and toluene from vehicle exhausts are expected to persist in the forest. Given these strong emitters of BVOCs and the evident transport of AVOCs into the forest, the study area proved to be a good test bed for measurement of the overall response of VOCs to abiotic stress in a way that simulates possible future atmospheric conditions.

**Comment 2:** I don't think the definition of an extreme temperature threshold as 32 °C is appropriate. A temperature of 32 °C is quite common. I understand that temperature conditions can vary significantly by

region. Typically, heatwave studies define thresholds using the 95th or 99th percentile of historical temperature records. Therefore, I suggest the authors use the 95th percentile of hourly July temperatures over the past 10 years as the threshold for defining heatwaves.

**Response:** We thank the reviewer for suggesting the use of heatwaves as a basis for defining extreme temperature, which provides strong complementary evidence for our initial definition. In our study, extreme temperature was defined as an hourly mean temperature exceeding 32 °C, which aligns with projected climate scenarios indicating a temperature increase of 2–4°C by 2100 (Collins et al., 2013). During the measurement period, the average temperature was 26°C; therefore, the defined extreme temperature is at least 5°C above the average temperature.

Heatwave events are defined as periods when the maximum daily temperature exceeds the 90th percentile calculated from a smoothed 15-day moving average (Perkins and Alexander, 2013; Perkins-Kirkpatrick and Gibson, 2017). To qualify as a heatwave, this occurrence must persist for three consecutive days. For our analysis, the calculated 90th percentile temperature for the month of July from 2015 to 2024 was 32°C, which was the same extreme temperature threshold defined for our case.

We added the following statement to substantiate our definition of extreme temperature.

The extreme temperature defined in this study aligns with the heatwave definition (Perkins and Alexander, 2013; Perkins-Kirkpatrick and Gibson, 2017), wherein the 90th percentile temperature for the month of July from 2015 to 2024 is 32°C.

For reference, the 95th percentile of temperatures over the past 10 years is 33°C, which is near our established threshold for extreme temperature conditions (32°C). However, the authors chose to maintain the 32°C cut-off to remain consistent with our initial definition and classification of heatwaves.

Comment 3: The authors insisted that "Regions dominated by emissions of  $\alpha$ -pinene,  $\beta$ -pinene, and limonene typically have a nighttime peak, whereas daytime enhancements are observed for areas with sabinene and ocimene." I cannot agree with this statement. The diurnal cycles of monoterpenes largely depend on whether their emissions are light-dependent. De novo monoterpenes typically peak during the day because their emissions are driven by light emissions (Jardine et al., 2015). Only temperature-dependent monoterpenes consistently show nighttime peaks due to the lower boundary layer height at night. The monoterpenes mentioned by the authors can be either light-dependent or only temperature-dependent. For example,  $\alpha$ -pinene emissions from Scots pine is light-dependent (Tarvainen et al., 2005). Furthermore, the paper (Borsdorf et al., 2023), which the authors cited in response to my comment, also mentioned that "The atmospheric concentrations of monoterpenes, which are emitted in a temperature dependent manner, are often greatest during nighttime hours due to the shrinking of the planetary boundary layer," and also noted that "Sabinene is obviously emitted in a light-dependent manner comparable with isoprene, while all the other monoterpenes are emitted by volatilization." Therefore, assigning diurnal emission patterns based on monoterpene isomer type is not appropriate, as each isomer may exhibit either emission pathway.

**Response:** After careful consideration, the authors have agreed to remove the statements regarding the designation of the monoterpene isomer based on the diurnal profile. We appreciate the reviewer for their insightful suggestion.

**Comment 4:** Figures 2 and 3 contain obvious errors. Figure 2 lacks a label for the first panel and has no axis labels for panel (C). In Figure 3, the standard deviations shown in the histogram plots appear too small compared to the actual distribution of data points displayed on the right side of the figure.

**Response:** Thank you for bringing the error to our attention. We have updated Figure 2 to include the missing labels. The error bars in Figure 3 represent the standard error, which accounts for the total number of observations. This is why the error bars are relatively small.

**Comment 5:** According to the NNMF results, does this imply that there is always a fire plume near the site? Lines 501–503 state: "oxygenated hydrocarbons (CxHyOw) and CxHyNz compounds were persistent in the early hours of the combustion plume." However, as the second reviewer mentioned, "VOCs become more oxygenated as they age away from the biomass burning source," which does not support the statement made here.

**Response:** Figure 7 shows that the oxygenated biomass burning factor experienced a significant enhancement on July 16, which is why the authors initially argued that it was persistent in the early hours of the combustion plume. However, the concentration of oxygenated compounds resulting from the oxidation of hydrocarbons should indeed increase afterward, not prior. To avoid any confusion, the following statement was excluded in the latest version of the manuscript.

This is consistent with the analysis based on atomic content: oxygenated hydrocarbons ( $C_xH_yO_w$ ) and  $C_xH_vN_z$  compounds were persistent in the early hours of the combustion plume.

**Specific comments:**

**Comment 1:** Figure 1. The term "Global solar radiation" used here is incorrect; the authors should use "Downward solar radiation" instead. In addition, UV refers to ultraviolet radiation, which is a portion of the whole solar radiation spectrum. Based on the values provided (around 1000 W m-2), I believe the authors are referring to total solar radiation rather than UV radiation, as such a high UV radiation level would not be survivable for any life forms. In addition, did author consider the smoke impact on BVOC emission during by affecting the solar radiation?

**Response:** The authors appreciate the reviewer's suggestion. The term "UV radiation" has been replaced with "downward solar radiation" throughout the manuscript. Additionally, the authors investigated the impact of smoke on solar radiation and how it subsequently modifies the concentration of biogenic volatile organic compounds (BVOCs). Figure R1 below displays the time series of solar radiation and smoke during the period of significant transport of combustion plumes. It is evident that solar radiation was not substantially affected by the combustion plumes.

**Figure R1.** Time series profile of solar radiation and smoke during the period of evident transport of combustion plumes

**Comment 2:** Line 252: The word "global budget" is not accurate. Guenther et al. (2012) calculated the emission, which is just part of the "global budget".

Response: The word "budget" was replaced with "emission".

**Comment 3:** Line 322: What kind of vegetation's physiological functions?

**Response:** This statement is generally applicable to several physiological functions of plants; however, our results do not support this claim. Therefore, this sentence has been removed from the latest version of the manuscript.

**Comment 4:** Line 354: I think that biogenic VOCs are temperature dependent.

**Response:** The statement was corrected. The new text reads:

Anthropogenic tracers such as benzene and xylene did not show dependence on temperature, unlike some BVOCs.

**References**

- CICCIOLI, P., CENTRITTO, M., and LORETO, F.: Biogenic volatile organic compound emissions from vegetation fires, Plant, Cell & Environment, 37, 1810-1825, https://doi.org/10.1111/pce.12336, 2014.
- Guenther, A. B., Zimmerman, P. R., Harley, P. C., Monson, R. K., and Fall, R.: Isoprene and monoterpene emission rate variability: model evaluations and sensitivity analyses, J. Geophys. Res. Atmos., 98, 12609-12617, 1993.
- Jin, L., Permar, W., Selimovic, V., Ketcherside, D., Yokelson, R. J., Hornbrook, R. S., Apel, E. C., Ku, I. T., Collett Jr, J. L., Sullivan, A. P., Jaffe, D. A., Pierce, J. R., Fried, A., Coggon, M. M., Gkatzelis, G. I., Warneke, C., Fischer, E. V., and Hu, L.: Constraining emissions of volatile organic compounds from western US wildfires with WE-CAN and FIREX-AQ airborne observations, Atmos. Chem. Phys., 23, 5969-5991, 10.5194/acp-23-5969-2023, 2023.
- Liang, Y., Stamatis, C., Fortner, E. C., Wernis, R. A., Van Rooy, P., Majluf, F., Yacovitch, T. I., Daube, C., Herndon, S. C., Kreisberg, N. M., Barsanti, K. C., and Goldstein, A. H.: Emissions of organic compounds from western US wildfires and their near-fire transformations, Atmos. Chem. Phys., 22, 9877-9893, 10.5194/acp-22-9877-2022, 2022.
- Perkins-Kirkpatrick, S. E. and Gibson, P. B.: Changes in regional heatwave characteristics as a function of increasing global temperature, Scientific Reports, 7, 12256, 10.1038/s41598-017-12520-2, 2017.
- Perkins, S. E. and Alexander, L. V.: On the measurement of heat waves, Journal of climate, 26, 4500-4517, 2013.
- Rinnan, R., Iversen, L. L., Tang, J., Vedel-Petersen, I., Schollert, M., and Schurgers, G.: Separating direct and indirect effects of rising temperatures on biogenic volatile emissions in the Arctic, Proc Natl Acad Sci U S A, 117, 32476-32483, 10.1073/pnas.2008901117, 2020.

**Response to Reviewer 2**

**General Summary:**

The work presented here reports the change in mixing ratio and distribution of several biogenic and anthropogenic VOC measured in a forest, as a response to increased temperature and transported biomass burning plumes. The authors emphasize the variability of VOCs and associated reactivity due to heat and smoke. This study investigating how the VOC distribution changes in response extreme events is highly relevant, and the authors have done quite a bit of work to improve upon the analysis from initial submission. The appropriate temperature responses have been added, and a section on reactivity has been included to expand the discussion of VOC variability and reactivity under extreme conditions. There are some minor adjustments that need to be made, but overall the reviewer is pleased with the additions and corrections the authors have made to help improve the manuscript. I recommend publication of the manuscript, after these small issues (addressed below) are rectified or clarified.

**Response:** The authors would like to thank the reviewer for their kind words. We have revised the manuscript in accordance with the reviewer's recommendations.

**Minor comments:**

**Comment 1:** I recommend abstract explicitly state the estimated and increased OH reactivity due to these exceptional events. This is a significant and informative addition that is highly relevant. The authors should do well to emphasize this!

**Response:** Agreed. The following sentence was added in the Abstract.

The calculated OH reactivities during extreme temperature condition and transport of biomass burning plumes were  $106.37 \pm 4.27 \,\mathrm{s}^{-1}$  and  $106.22 \pm 5.15 \,\mathrm{s}^{-1}$ , respectively, which are substantially higher than the background level of  $98.78 \pm 1.16 \,\mathrm{s}^{-1}$ .

**Comment 2:** Figure 1 is missing (A), Figure 1C is missing a label for the x-axis, not sure what this is representing?

**Response:** Thank you for bringing the error to our attention. We have updated Figure 2 to include the missing labels. The error bars in Figure 3 represent the standard error, which takes into account the total number of observations. This explains the relatively small size of the error bars.

Comment 3: L285 to L287: Minor point, but I would caution against the authors using "large" to broadly refer to particles >50 nm. Based on the geometric mean diameter shown in the supplementary information Fig. S2B, the particles fall into the accumulation mode and are technically "fine mode" aerosol. Large particles typically refer to particles in the coarse mode (e.g. >2500 nm or 2.5 microns). Suggest removing 'large' in lines 285 to 287, and instead simplify to say "..Particles >50 nm in diameter were observed..." and "...for the presence of these particles..." You could also consider stating the average (+/-) geometric mean of particles throughout the study for reference.

**Response:** We agree with the reviewer's feedback and have revised these statements. The updated statements now read:

Particles >50 nm in diameter were observed with no apparent aerosol growth (see Figure S2). The average geometric mean diameter in MOFLUX site was  $85.53 \pm 16.68$  nm.

The most probable reason for the presence of these particles was the isoprene-rich condition of the temperate forest that impacted the aerosol nucleation, even with enough monoterpene and ozone available for particle formation.

**Comment 4:** L381-383: It's both interesting and intriguing that catechol also has a sharp peak after 6 PM in the diurnal cycle shown in Fig 2M. Is there some process responsible for this, or is that just an artifact from data averaging smoke impacted days? The lines here mention that it increased significantly to 300 ppt on some days, so it could just be that the timing of the smoke impact corresponded with that sharp increase?

**Response:** The authors are uncertain about the reason for the sharp peak observed after 6 PM. These enhancements did not occur during the days impacted by smoke. The research team has attempted to

identify a likely explanation for the elevated concentration, but we are unable to provide a definitive suggestion at this time.

**Comment 5: Comment:** L388: Make sure to specify/reiterate that smoke concentration is derived from HRRR. (E.g. "Figure 4 shows the HRRR-derived smoke concentration..."

**Response:** The authors concur. The new statement reads:

Figure 4 shows the HRRRR- derived smoke concentration measured at MOFLUX.

**Comment 6:** Figure 4B: It's difficult to read/see the trajectory as an inset in the figure. Is it possible to make it its own separate panel, or move it to the supplement as part of figure S5?

**Response:** The authors have provided a larger image with higher pixel resolution in Figure S5. The following statement has been added to the caption:

A larger image with higher resolution of the trajectory is presented in Figure S5.

Comment 7: L437-438: Are there units on this value of 8.5? Is this different from Csat?

Response: This value is saturation concentration (Csat). Unit (µg m-3) was added to the statement.

**Comment 8:** L450-451: Presumably "low temperature" means <32 C and extreme is >32 C? Suggest defining/stating this in parenthesis next to the statements to prevent ambiguity. Is the low temperature and corresponding reactivity in reference to the selected time period (July 8 to 17) or is that the average for non-impacted/non-extreme temp outside of that date range as well? Please clarify.

**Response:** Agreed. The numerical values for the temperature conditions have been added to the text. Additionally, the atmospheric reactivity mentioned in these statements is based solely on the selected time period from July 8 to 17. We have rephrased this statement for clarity. See new statements in our response to Comment 9.

Comment 9: L450-L453: The additional reactivity calculations do a lot to help develop the work and associated impacts! These calculations are a highly useful addition. However, I think this section could be worked on a bit to provide more clarity and help emphasize the significance of what is shown. Is it possible to include OH reactivity during "background" or typical conditions during times not influenced by temperature or combustion? Looks like July 9th to 10th could be a possible time-frame that would work for this? L443 earlier states that average reactivity between July 8 to July 17 was 91.30, but this average would presumably include days that experienced elevated temperature and BB impacts. What does the reactivity look like outside of these times? This would provide a useful benchmark for determining exactly how much the reactivity increased due to these extreme events. If it's not possible to calculate yourself, is there a study you can cite that might provide an estimate? I'm also confused as to how it's possible that the reactivity at low temperature (however that is defined) is higher than the average for this entire "impacted" period (98.92 low temp versus 91.30 s-1 average July 8 to July 17)? Clearer wording and phrasing throughout the paragraph would strengthen this finding.

**Response:** We concur with the reviewer. We have redefined the periods with low temperature conditions as "background" since these durations experienced temperatures below  $32^{\circ}$ C and were unaffected by biomass burning. The average reactivity during the background conditions was  $98.78 \pm 1.16 \, \text{s}^{-1}$ . Additionally,

we revisited the calculation of average reactivity from July 8 to 17, and the correct mean value during this period was  $100.53 \pm 10.80 \, \text{s}^{-1}$ .

We modified the statement to clarify the "background" condition and corrected the average reactivity values. Furthermore, we included a comparison of our average reactivity with findings from other studies in response to the reviewer's subsequent comment.

During this period, the average OH reactivity was 100.53 ± 10.79 s-1, which was evidently higher compared to previous measurements in an urban environment in California, USA (Hansen et al., 2021), a sub-urban site in Shanghai, China (Yang et al., 2022), and forest environments in Finland (Sinha et al., 2010) and France (Bsaibes et al., 2020). The elevated reactivity calculated in this study was accounted to the notable contributions from isoprene, acetone, ethylamine, and ethenone. To assess the impact of elevated temperature and biomass burning on atmospheric reactivity, the data were categorized based on recorded ambient temperature and smoke concentration. The influence of biomass burning was evident from July 15 at 07:00 to July 17 at 20:00. Only one hour within this period had a temperature above 32°C, in which that data point was excluded from the average reactivity calculation. Conversely, the effect of extreme temperatures was evaluated using data recorded from July 8 at 01:00 to July 15 at 06:00. Within this timeframe, 30 hours met the extreme temperature criteria (>32°C), allowing an assessment of the potential impact of future warming on atmospheric reactivity. Periods with temperature conditions below 32°C and not influenced with combustion plume were tagged as background. The average OH reactivity for periods with enhanced temperatures (>32 °C) and with transported plumes was  $106.37 \pm 4.27 \text{ s}^{-1}$  and  $106.22 \pm 5.15 \text{ s}^{-1}$ . respectively, which are substantially elevated compared to background conditions (98.78 ±1.16 s- 1). The comparable OH reactivities of the two future climate scenarios highlight the reactive nature of the BB gas phase species such as benzene and acetonitrile. Overall, the calculated averages during the extreme heat and wildfires clearly modified the atmospheric reactivity in the forest.

**Comment 10:** L451: It would be useful to compare your estimates of OH reactivity to other studies (SOAS, CalNEX, WE-CAN, etc.), for context.

Response: See previous response.

Comment 11: L453, & Figure 5: One thing not emphasized in the text that I think would benefit from more attention is that according to L453 and Figure 5, even though total VOC was lower during the BB event compared to extreme heat, total volatility increased (presumably due to presence of BB compounds like benzene and HCN as mentioned in text). Total reactivity under BB was comparable to reactivity under high temp conditions without any BB influence. Given the lower apparent tVOC, this really highlights how reactive BB gas phase species are! This should be emphasized, especially when considering the projected increase in BB and extreme temp events in the future, and also begs the question—under these future climate scenarios what will have more of an influence in this kind of environment: enhanced temperature, or enhanced BB?

**Response:** Indeed, the similarity in the calculated OH reactivity during extreme temperature conditions and during the transport of the combustion plume highlights the reactive nature of biomass burning gas species. This will be included in the new version of the manuscript.

Regarding which future climate scenario will have a greater influence, a more comprehensive characterization of VOCs over an extended measurement period, supplemented by modeling procedures,

will be necessary. The current limited dataset, which shows similar values of OH reactivity, is insufficient to draw a valid conclusion about which scenario will exert a greater influence.

We have added the following statement to the latest version of the manuscript:

The comparable OH reactivities of the two future climate scenarios highlight the reactive nature of the BB gas phase species such as benzene and acetonitrile.

**Comment 12:** L466-L470: Do you have data that you can add here to support this? (E.g % change enhancement due to increased temperature, or make a table in the supplement). Without the values there the reader is left to take the author's word for it.

**Response:** The authors agree. Percent enhancement values calculated using average concentrations at low (<32°C) and high (>32°C) temperature conditions are now provided in this section. The statement now reads:

VOCs such as formic acid ( $CH_2O_2$ , 8%), acetic acid ( $C_2H_4O_2$ , 83%), acrolein ( $C_3H_4O$ , 62%), furan ( $C_4H_4O$ , 62%, 51%), methylglyoxal ( $C_3H_4O_2$ , 51%), and glycolic acid ( $C_2H_4O_3$ , 68%) exhibited enhancement at the extreme temperature conditions, although it is equally possible that these compounds were also associated with transport of the combustion plumes. Values inside the parentheses are percent enhancement calculated using average concentrations at low (<32°C) and high (>32°C) temperatures conditions

**Comment 13:** Figure 6: The CxHyNz category is a bit washed out and difficult to see. Can you make it a different color or make it darker? Same for the error shading in diurnal plot Figure 6C.

**Response:** We updated figure 6 to enhance the readability of the plot, particularly for CxHyNz category. Here is the new figure 6 integrated into the new version.

Comment 14: L496: Define what a substantial number is. N=?

**Response:** The number of compounds that contribute substantially to factor "others" is now provided. Here is the new statement.

The "others" factor, with a substantial number (N = 137) of contributing compounds, remains unidentified.

**Comment 15:** L504-505: Can you remind the reader of the significance of LogCsat >6? What do values higher than this imply? Reiterate the message you want to convey.

**Response:** We apologize for the confusion. The statement was modified to indicate that compounds under O-BB and H-BB factors are volatile based on saturation mixing ratio.

Also, the O-BB and H-BB factors are classified as volatile (log Csat > 6  $\mu$ g m-3), based on the saturation mixing ratio values of 7.27 and 8.45  $\mu$ g m-3.

**Comment 16:** L512-513: "This is unlikely due to the expected biogenic emission in the forest, although it came second with 34% contribution compared to 66% from the H-BB factor." This sentence is worded strangely... Does "this" refer to the preceding sentence describing monoterpene and its fragment? Is "it" also monoterpene and its fragment? Not sure what This/it refers to here, can you please clarify?

**Response:** The statement was rephrased to provide more clarity. The new sentence now reads:

The inclusion of monoterpene to the H-BB factor is unlikely due to the expected biogenic emission in the forest, although the biogenic factor accounted for the second-largest contribution at 34%.

**Comment 17:** L514-L515: Hmmm, are BB events considered anthropogenic? I think this is the subject of ongoing debate and is a hot topic! The answer depends on where you are in the world. Suggest slight rephrasing to ... "originate from anthropogenic sources and BB events"

**Response:** The statement was rephrased according to the suggestion of the reviewer.

**Comment 18:** L525-526: Do you have a reference to support nighttime oxidation of NO3 contributing to the formation of these species?

**Response:** Currently, there is no data available to support this statement; therefore, it has not been included in the latest version of the manuscript.

**Comment 19:** L528-529: "During the transport of BB plumes, the secondary factor had a relatively low increase in signal compared to both BB factors, which shows that oxidation compounds were generally locally...." I don't think this is true, given that one of your factors is literally called, "oxygenated BB", which implies that there was indeed some contribution from long-range transport... I think maybe what you mean to say is, "the secondary factor had... low increase... showing that secondary formation was predominantly locally generated..."

**Response:** We agree with the reviewer. The statement was modified to explicitly indicate that secondary formation is primarily generated locally. The new statement reads:

During the transport of BB plumes, the secondary factor had a relatively low increase in signal compared to both BB factors, which shows that secondary formation was predominately locally generated with little to no contribution from long-range transport

**Comment 20:** L530-531: Re-emphasize the change to chemistry during the BB impact to strengthen your messaging. What was the increase in reactivity during BB events compared to non-BB events (see earlier comment)? Add it here.

**Response:** Agree. We added the following sentence in this section.

This was corroborated by the enhanced reactivity (106.22 s-1) during the transport of combustion plume compared to background conditions (98.92 s-1).

**Comment 21:** L533: What is meant by a "critical" VOC? Suggest removing this adjective and just saying, "VOCs have important contributions to several..."

Response: The word "Critical" was removed in this sentence.

**Comment 22:** L536: Change to "anthropogenic and fire emissions". Anthropogenic emissions typically refer to things like traffic, shipping, home heating, etc. While it's true that a significant portion of wildfires are started by humans, that's not always the case. Factors contributing to fire ignition are dependent on location and time of year, among others.

**Response:** The authors concur. The new statement now reads:

...the forest included several sources of biogenic compounds and was influenced by short- and long-range transport of anthropogenic and fire emissions

**Comment 23:** L538-539: Some of these are also sourced from anthropogenic emissions (e.g. benzene, toluene), so I suggest removing "in the forest" after typical VOCs, especially when they are referenced as AVOCs later in the conclusions (L545). Add the error to the average total mixing ratio reported at the end of the line in L539.

**Response:** Agree. The phrase "in the forest" was removed and error was added to the sum of VOCs. New statement reads:

Typical VOCs, consisting of methanol, acetone, isoprene, monoterpene, MVK+MACr, benzene, toluene, acetonitrile, and catechol, had an average total mixing ratio of  $69 \pm 34$  ppb.

Comment 24: L541-L542: Add the uncertainty or standard deviation with these averages.

Response: Uncertainty values were added in this sentence.

Among the VOCs, isoprene had one of the highest recorded average mixing ratios ( $10 \pm 9$  ppb), next to methanol ( $23 \pm 10$  ppb) and acetone ( $22 \pm 9$  ppb).

**Comment 25:** L557-L558: "...Increases could be significantly affected regionally by accompanying changes in atmospheric circulation." I'm not sure what this means or is supposed to imply. Are you saying that the extent of predicted increases could be regionally impacted due to changes? Suggest rewording for clarity.

**Response:** Instead of rewriting, the authors decided to remove the statement in the recent version, thank you for pointing this out.

**Comment 26:** L565: Not sure what a typical VOC is. I think you mean to say background/typical conditions? "Typical VOC" likely changes based on location and environment.

**Response:** The word "typical" was removed in this sentence avoid confusion.

**Comment 27:** L567: Important to note that this is for all of the measured VOCs\*. Estimates could vary if additional species are measured to include those beyond the list of 250+ you have.

**Response:** Agree. The word "all" was replaced with "measured" to constrain the statement to observed VOCs using our current instrumentation. The new statement reads:

The O:C and H:C ratios of the measured VOCs, as well as their volatility, provided insight into their response to future climate scenarios

**Comment 28:** L568: But you had a whole factor named "oxygenated BB" that showed enhancement so it's not clear to me how it's fair to say, "During BB plume transport, less oxygenated compounds were enhanced..."?

**Response:** The authors apologize for the mistake. The research team meant hydrocarbons ( $C_xH_y$ ), instead of less oxygenated compounds. The new statement reads:

During BB plume transport, hydrocarbons ( $C_xH_y$ ) with high volatility were enhanced.

**References**

- Bsaibes, S., Al Ajami, M., Mermet, K., Truong, F., Batut, S., Hecquet, C., Dusanter, S., Léornadis, T., Sauvage, S., Kammer, J., Flaud, P. M., Perraudin, E., Villenave, E., Locoge, N., Gros, V., and Schoemaecker, C.: Variability of hydroxyl radical (OH) reactivity in the Landes maritime pine forest: results from the LANDEX campaign 2017, Atmos. Chem. Phys., 20, 1277-1300, 10.5194/acp-20-1277-2020, 2020.
- Hansen, R. F., Griffith, S. M., Dusanter, S., Gilman, J. B., Graus, M., Kuster, W. C., Veres, P. R., de Gouw, J. A., Warneke, C., Washenfelder, R. A., Young, C. J., Brown, S. S., Alvarez, S. L., Flynn, J. H., Grossberg, N. E., Lefer, B., Rappenglueck, B., and Stevens, P. S.: Measurements of Total OH Reactivity During CalNex-LA, J. Geophys. Res. Atmos., 126, e2020JD032988, https://doi.org/10.1029/2020JD032988, 2021.
- Sinha, V., Williams, J., Lelieveld, J., Ruuskanen, T. M., Kajos, M. K., Patokoski, J., Hellen, H., Hakola, H., Mogensen, D., Boy, M., Rinne, J., and Kulmala, M.: OH Reactivity Measurements within a Boreal Forest: Evidence for Unknown Reactive Emissions, Environ. Sci. Technol., 44, 6614-6620, 10.1021/es101780b, 2010.
- Yang, G., Huo, J., Wang, L., Wang, Y., Wu, S., Yao, L., Fu, Q., and Wang, L.: Total OH Reactivity Measurements in a Suburban Site of Shanghai, J. Geophys. Res. Atmos., 127, e2021JD035981, https://doi.org/10.1029/2021JD035981, 2022.

---

## Editor Decision (ED2)

**Scientific:**

Lines 198-208, calculation of OH reactivity, is somewhat confusing. One suggestion is to be consistent with language and use "reaction rate constant" throughout. Line 202...does this mean that reaction rate constants were calculated for all compounds with hourly resolution using measured temperature? Please just reread for clarity.

Lines 280-282: There are counter-examples of measured terpenes being highest at night, with clear daytime peaks. Bouvier-Brown et al., 2009; Pfannerstill et al., 2024

Line 302: Do you mean that benzene can initiate new particle formation or just contribute to SOA formation? I am unfamiliar with the former (contribution of benzene to NPF).

Lines 350-354: Note that toluene also has a biogenic source, and appears as an emission in biogenic emissions models (e.g., MEGAN). This does not mean your conclusion is wrong, but you may want to rethink given that toluene can be biogenic.

**Editorial:**

It is suggested to remove all exaggerated/non-quantitative/qualitative language. Some examples include: "grave implications" (line 49), "imminent implications" (line 359), remove "clearly" in Fig 1 caption.

Line 66: suggestion to change "..ozone enhancement will lead to..." to "ozone enhancement may lead to..."

Line 88: "several VOCs" should be changed to "hundreds of VOCs"

Line 156: needs revision

Line 166: cabin or shed? (not clear what a cabin shed is)

Line 167: "can be" missing "found"

Line 371-378: This section describes "sporadic...emission and transport" and "infrequent emissions". During a fire, emissions and transport will be ongoing (and not sporadic or infrequent). It is more than the variability in emissions and transport lead to infrequent interception of the smoke at any given location. Suggestion to reword to make that clearer.

Line 439: "accounted for" should be changed to "attributed to"

Lines 549-554: This paragraph is talking about biomass burning/wildfire emissions, but concludes with talking about "such AVOCs". BB/wildfire emissions are not considered AVOCs. The specific compound, benzene, that you are describing has both pyrogenic and anthropogenic sources. Last sentence should be revised to make that clearer.

---

## Author Response (AR3)

**Measurement Report: Extreme Heat and Wildfire Emissions Enhance Volatile Organic Compounds in a Temperate Forest**

Christian Mark Salvador et al.

We appreciate additional comments of the handling editor of our manuscript, and we provided here a detailed response that addresses the handling editor 's concern. Our point-by-point responses to the Editor's general and specific comments are presented below. The Editor's comments are in black, and our answers are in red. Modified or new statements integrated into the revised manuscript are indented. All changes can be seen in the revised version of the manuscript in red font.

**General Comment:**

The authors have an important dataset that captures changes in emissions sources and meteorological conditions in a location that has seen previous study. The results provide some insight in changes in atmospheric composition and reactivity driven by changes in source and/or meteorological conditions that generally contribute to our understanding of atmospheric chemistry. The methods are appropriate, with some minor comments and suggestions given below. These strengths demonstrate that this manuscript is appropriate for consideration as a measurement report in ACP. Following minor comments and suggestions on the methods are broader scientific comments and concerns that should be address in the revision stage. Some of these comments and concerns will naturally resolve with the change in format for manuscript type, and a tighter focus on the campaign and results.

**Comment:** We welcome the suggestion of the editor to convert the manuscript into a measurement report type under the ACP publications. The authors have revised the manuscript to place greater emphasis on the results of the field measurements rather than the broader implications of the research. In addition to addressing the editor's comments and suggestions, we have also changed the title of the manuscript to meet the requirements for measurement reports. The new title is as follows:

**Measurement Report: Extreme Heat and Wildfire Emissions Enhance Volatile Organic Compounds in a Temperate Forest**

Another requirement for measurement reports is the accessibility of the data presented. The data availability section was modified according to the data policy and regulations of ACP:

The data used in this publication are available to the community and can be accessed in ORNL's Terrestrial Ecosystem Science Scientific Focus Area - Data Products and Tools website under the Volatile Organic Compounds and Meteorological Conditions in the Missouri Ozark AmeriFlux (MOFLUX) Site, 2023 data product which can be accessed via the DOI: <a href="https://doi.org/10.25581/ornlsfa.033/2409393">https://doi.org/10.25581/ornlsfa.033/2409393</a>

The DOI is already functional and can be accessed by anyone without the need for registration or a license.

Several statements have been modified or removed to better highlight the results of the measurements. These changes are detailed in response to Broad Comments (BC) number 9 (BC 9).

**Minor Comments, MC (with a focus on Methods)**

**MC 1:** Line 89-Several **hundreds** of compounds have been identified in wildfire smoke (see for example Hatch et al. https://doi.org/10.5194/acp-15-1865-2015, Koss et al. https://doi.org/10.5194/acp-18-3299-2018, Selimovic et al. <a href="https://doi.org/10.5194/acp-18-2929-2018">https://doi.org/10.5194/acp-18-3299-2018</a>, Selimovic et al. <a href="https://doi.org/10.5194/acp-18-2929-2018">https://doi.org/10.5194/acp-18-2929-2018</a>, Binte-Shahid et al. <a href="https://doi.org/10.5194/gmd-17-7679-2024">https://doi.org/10.5194/acp-18-2929-2018</a>, Binte-Shahid et al. <a href="https://doi.org/10.5194/gmd-17-7679-2024">https://doi.org/10.5194/acp-18-2929-2018</a>, Binte-Shahid et al. <a href="https://doi.org/10.5194/gmd-17-7679-2024">https://doi.org/10.5194/acp-18-2929-2018</a>, Binte-Shahid et al. <a href="https://doi.org/10.5194/gmd-17-7679-2024">https://doi.org/10.5194/gmd-17-7679-2024</a>).

**Response:** The authors cited the studies that detailed the compounds identified in wildfire smoke. The text now reads:

During wildfire events, the combustion of vegetation and other biomass induce the pyrolysis of plant materials which ultimately release several VOCs during the process (Ciccioli et al., 2014; Hatch et al., 2015; Koss et al., 2018; Selimovic et al., 2018; Binte Shahid et al., 2024).

**MC 2:** Line 94-While it is true that there are significant uncertainties due to the huge variability inherent in fires and emissions, there are a lot of studies that have looked at the influence of wildfire emissions on atmospheric chemistry and reactivity that should be cited here (see for example Gilman et al. doi:10.5194/acp-15-13915-2015, Liu et al. doi/10.1002/2016JD025040, Kumar et al. <a href="https://www.nature.com/articles/s41598-017">https://www.nature.com/articles/s41598-017</a> 19139-3, Permar et al. <a href="https://pubs.rsc.org/en/content/articlehtml/2023/ea/d2ea00063f">https://pubs.rsc.org/en/content/articlehtml/2023/ea/d2ea00063f</a>).

**Response:** The authors are grateful for the suggestion. The studies were cited in the new version of the manuscript.

However, the relative importance and contribution of volatile organic compounds (VOCs) from wildfire activities to atmospheric reactivity remain uncertain, even with several studies tackling the problem(Gilman et al., 2015; Kumar et al., 2018; Permar et al., 2023).

**MC 3:** It might be useful to include a map of the measurement location, particularly showing the influence of biogenic emissions and smoke transport during the study period.

**Response:** We agree. The manuscript now includes a map that displays the measurement location along with a 50 km radius surrounding the site. This figure clearly shows the interstate and forested areas, which may contribute to the sources of anthropogenic and biogenic VOCs in the temperate forest. Additionally, the impact of transported wildfire smoke is highlighted in Figure S5 in the supplementary file.

**Figure 1.** Map of the MOFLUX site located in Missouri. The circle indicates a 50 km radius from the site. The figure clearly illustrates the interstate and forested areas, which may contribute to the sources of anthropogenic and biogenic VOCs in the temperate forest.

**MC 4:** It is recommended to review the first sentences of the paragraph starting on line 99-some of these details would be more appropriate in the methods.

**Response:** The last paragraph of the introduction section was modified according to the editor's recommendation. The last paragraph of the introduction now reads:

In this work, we conducted a field campaign in the summer of 2023 to quantify the variability of VOCs over a temperate oak—hickory—juniper (Quercus—Carya—Juniperus) forest in the Ozark Border Region of central Missouri. The primary goal of the campaign was to examine the influence of temperature on VOCs. We deployed a high-resolution chemical ionization mass spectrometer to continuously measure VOC concentrations. We were also able to incorporate opportunistic analyses of the impact of wildfire emissions on the variability of VOCs due to the smoke that reached our site because of extreme forest biomass burning activity in Canada.

The following statements were moved to method sections: Site Description and Meteorological Data:

The Ozark Plateau (Wiedinmyer et al., 2005), and this site in particular, is a known hotspot for emissions of BVOCs such as isoprene and monoterpene. Drought is a critical event at MOFLUX, as such environmental stress induced the highest ecosystem isoprene emission ever recorded for a temperate forest in 2011 (53.3 mg m-2 h-1) (Potosnak et al., 2014). Field measurement campaign in 2012 in MOFLUX reported isoprene reaching a maximum concentration of 28.9 ppbv, while monoterpenes peaked at 1.37 ppbv over half-hour intervals (Seco et al., 2015). Moreover, the site is about 5 km away from a major highway, thus anthropogenic VOCs (AVOCs) such as benzene and toluene from vehicle exhausts are expected to be transported into the forest. Given these strong emitters of BVOCs and the evident

transport of AVOCs into the forest, the study area proved to be a good test bed for measurement of the overall response of VOCs to abiotic stress in a way that simulates possible future atmospheric conditions.

Information regarding the instrument was also added in section 2.2 VOC Measurement and Identification

The mass resolution of the technique (6000 m/ $\Delta$ m) provided an extended list of VOCs, beyond the usual routinely evaluated compounds (e.g., methanol, isoprene, and monoterpene).

MC 5: Line 111-why are vehicle emissions expected to persist in the forest?

**Response:** Indeed, the word "persist" was inappropriate for the statement. Thus, it was replaced with "transported", consistent with the proximity of the interstate highway to measurement site. The statement now reads:

Moreover, the site is about 5 km away from a major highway, thus anthropogenic VOCs (AVOCs) such as benzene and toluene from vehicle exhausts are expected to be transported into the forest .

**MC 6:** Line 144-while it makes some sense to use the HRRR forecasts to identify periods with smoke influence (which need to be more clearly defined in the methods-i.e., what periods met the established threshold?), it would strengthen the paper to use measurements to confirm impacts of smoke during these periods (e.g., using PM2.5 and/or CO data, or even acetonitrile or other known fire tracers measured during the campaign).

**Response:** Acetonitrile was already utilized to confirm the impacts of wildfire emissions during our measurements. This was highlighted in statements in section 3.3.

Among the major VOCs, acetonitrile and benzene appeared to be associated with the transport of the combustion plumes. These two VOCs had day average mixing ratios of 2.15 (acetonitrile) and 0.34 (benzene) ppb on July 16, corresponding to increases of 139% and 269%, respectively, compared to July 12, which is non-BB day.

This was also highlighted in Figure S6, which showed the time series profile of benzene and smoke between July 8 to 17, when enhanced transport of wildfire emissions were observed in MOFLUX.

Figure S6: The time series profile of benzene and smoke between July 8 to 17.

MC 7: There are a lot of details about operation of the PTRMS that could be moved to the SI (e.g., equations 1-3 and associated text). The authors may want to include additional references on the use of PTRMS for measurement of monoterpenes and known challenges/interferences if relevant

**Response:** Agreed. Several statements regarding the operation of the PTR-ToF-MS were moved to supplementary information.

**Operation of PTR-ToF-MS 6000X2, Conversion of Raw Signals, and Definition of Mixing Ratio**

In PTR-ToF-MS, hydronium ions are utilized to charge the VOCs through a non-dissociative proton transfer in the reaction chamber of the instrument. This technique can identify a wide range of compounds (e.g., carboxylic acids, carbonyls, and aromatic hydrocarbons) if the target compound has a proton affinity higher than water (691 kJ/mol). The protonation occurs as follows:

$$H_3O^+ + VOC \rightarrow H_2O + VOC - H^+ (1)$$

The conversion of raw signals (counts per second) to mixing ratio (ppb) of an uncalibrated gas VOC can be performed using the following equation:

$$[VOC, ppb] = \frac{1}{k\Delta t} \times \frac{I(VOC^{+})}{T(VOC^{+})} \times \frac{I(H_{3}O^{+})}{T(H_{3}O^{+})}$$
(2)

Where  $RH^+$  is the protonated gas compound, k is the proton-transfer-reaction rate coefficient,  $\Delta t$  is the reaction time,  $I(VOC^+)$  and  $I(H_3O^+)$  are the measured ion count rates for the  $RH^+$  and the hydronium ion  $(H_3O^+)$ , respectively.  $I(VOC^+)$  and  $I(H_3O^+)$  are the transmission efficiencies for  $RH^+$  and  $H_3O^+$  ions (Worton et al., 2023; Taipale et al., 2008). All the values are readily available except for transmission efficiency value, which can be determined by generating a mass dependent transmission curve from compounds with known concentrations and reaction rate. The transmission is an

instrument-specific parameter that depends on the transmission efficiencies of the lens system/ion guide, the mass filter (TOF), and the ion detector. Transmission Tool provided by the instrument developer (IONICON) was used to generate the transmission efficiencies of gas standards. The PTR-ToF-MS was operated with 2.6 mbar and 80°C drift tube pressure and temperature, with an E/N value of ~119 Townsend. The mass range was set up to 500 m/z. The single spectrum time was set to calculate the fluxes of the VOC, the results of which will be reported in subsequent works. One of the limitations of the PTR-ToF-MS technique is that it cannot distinguish isomers (e.g.,  $\alpha$ -pinene,  $\beta$ -pinene, and limonene) because of their identical exact mass (Blake et al., 2009).

The term mixing ratio used in this manuscript is defined as the ratio of the moles of target analyte to the moles of all of atmospheric gases (i.e.,  $N_2$  and  $O_2$ ). This can be expressed as the following equation:

$$R_i = \frac{n_i}{n_{\Sigma} - n_i} \approx \frac{n_i}{n_{\Sigma}} \quad (3)$$

Where  $R_i$  is the mixing ratio,  $n_i$  is the moles of gas analyte, and  $n_{\Sigma}$  is the total moles of atmospheric gases. The amount of organic gases in the atmosphere is significantly lower than the total gases.

**MC 8:** It is not clear what value was used for the reaction rate constant of the summed monoterpenes with OH and how that was calculated or what assumptions were made.

**Response:** The reaction rate constant for monoterpenes was determined using literature values for  $\alpha$ -pinene,  $\beta$ -pinene,  $\beta$ -carene, limonene, camphene, and  $\beta$ -ocimene. The median reaction constant for all these monoterpenes was then used to calculate the OH reactivity. This information has been incorporated into the updated version of the manuscript.

To address molecular formulas with multiple records, the median value of the rate constants was employed. For example, the reaction rate constant for the class of monoterpenes was derived from literature values of several compounds, including a-pinene,  $\beta$ -pinene, 3-carene, limonene, camphene, and  $\beta$ -ocimene. The median reaction constant across all these monoterpenes was then utilized to calculate the OH reactivity.

MC 9: Line 430-log saturation "mixing ratio" should be log saturation vapor concentration

**Response:** Thank you for noticing the error. The new text now reads:

The average log saturation vapor concentration for all the compounds was 7.50  $\mu$ g m-3, and 100 and 136 compounds were classified as intermediate and volatile organic compounds, respectively.

**Broader Comments on Science and Scope, BC**

The mechanisms by which climate change and climate extremes affect biogenic, anthropogenic, and wildfire emissions are very different, and that complexity is not clearly or adequately addressed in the manuscript as currently written. For example, in the abstract

lines 17-18 describe the "...response of VOCs to future conditions such as extreme heat and wildfire events...". It is well documented that biogenic emissions can be temperature dependent, with compound- and species-specific differences. It is also well documented that some anthropogenic emissions can also be temperature dependent, e.g., asphalt emissions, but that strong temperature dependence is typically more indicative of biogenic emissions. In addition, reaction product formation can also have some dependence on temperature (as temperature can influence oxidant levels and reaction rates). This is addressed to some extent in section 3.2, but some refinements are needed.

**BC 1:** The authors state the at the BVOCs "respond well" to variations in temperature. What does this mean? They behave as expected/consistent with other studies?

**Response:** The authors modified the statement to be consistent with our results. The new statement now reads:

The major BVOCs, isoprene and monoterpene, increased with temperature, as shown in Figure 4.

**BC 2:** It appears the isoprene and monoterpenes have the same exponential response to temperature. I do not think that this is completely consistent with other studies (or for example the temperature response as would be predicted in a biogenic emissions model), as isoprene typically shows a much stronger response to temperature. There may be some studies that show very strong dependence of monoterpene emissions in extreme temperatures, but this is not clear as this section is currently written. It is suggested that the temperature sensitivity of isoprene and monoterpenes be discussed separately in this section, including context from references that are specific to isoprene and to monoterpenes. The authors state that the enhanced emissions of monoterpenes "can also be linked to", suggesting that at least one other mechanistic reason for enhanced emissions has already been discussed, but this is not the case. One suggestion is to more generally summarize (with appropriate citations) (and not to get into any specific mechanisms).

**Response:** We welcome the suggestion of the editor to separate the discussion of temperature sensitivity of isoprene and monoterpene. We also concur with providing a more general summary. New texts were added in the manuscript:

Furthermore, Figure 4 shows the evident exponential relationship between temperature and the major BVOCs, consistent with previous studies in which temperature controls the emission of isoprene and monoterpene (Hu et al., 2015; Selimovic et al., 2022; Guenther et al., 2012). The empirically determined coefficients (β) for isoprene and monoterpene are 0.13 and 0.12, respectively. The emission of isoprene is linked to plant thermotolerance, which is the ability of plants to endure and adapt to high temperatures without experiencing detrimental effects on their growth (Sharkey et al., 2007; Duncan et al., 2009). While the dependence of monoterpene emissions on temperature appears similar, based on the empirically determined coefficients calculated in this study, the mechanisms for monoterpene release from vegetation differ from those of isoprene. This variation is primarily due to

plants' ability to store monoterpenes and their high- water solubility and elevated temperature leads to vaporization of stored monoterpenes. (Loreto and Schnitzler, 2010; Malik et al., 2019; Malik et al., 2023)

**BC 3:** The discussion of MVK and MACr is confusing as written. The discussion seems to be mixing the effects of temperature on emissions of isoprene (and therefore its oxidation products) with the effects of temperature on lifetime (which could alter the ratio as a function of temperature). These effects need to be more clearly differentiated in this section and this distinction should be considered throughout the manuscript when discussing temperature effects particularly when the compounds can be oxidation products.

**Response:** The authors agree. We simplified the discussion on the impact of temperature of yields of MVK/MACr from Isoprene. The new paragraph now reads:

MVK and MACr produced from the oxidation of isoprene showed a strong association with temperature. MVK and MACr reached a 20-ppb average mixing ratio during extreme temperature conditions. As shown in Figure 4, the concentration of MVK/MACr doubled during extreme temperature conditions compared at low temperatures. This is consistent with a prior study that showed the yield of MVK increased with temperature (Navarro et al., 2013).

The discussion on lifetime of isoprene based on the ratio was moved to section 3.1. These statements are now included in the general results of MVK/MACr

Furthermore, The ratio between isoprene and MACR + MVK indicates the lifetime of the isoprene and degree of oxidation of isoprene to MVK and MACr. Greater than value of 1.0 observed in MOFLUX suggests transport time shorter than one isoprene lifetime, as indicated in the previous studies (Selimovic et al., 2022; Hu et al., 2015).

**BC 4:** Toluene is also emitted from biogenic sources. The discussion around toluene (including attributing everything to interference/fragmentation of monoterpenes) needs to be reconsidered and revised accordingly in this context.

**Response:** We agree. The possible contribution of toluene from biogenic emission is now added in the new version of the manuscript.

Interference of para-cymene fragmentation in the drift tube of the PTR-ToF-MS at mass 93 (Ambrose et al., 2010) might have impacted the observed concentrations at MOFLUX although we also do not discount the emission of toluene from vegetation (Heiden et al., 1999).

Remarkably, the toluene mixing ratio (0.73 ppb) doubled at higher temperatures, unlike the benzene and xylene. This result further supports our initial claim that the compound occurring at mass 93 originates from the fragmentation of monoterpene or from the emission of toluene from biogenic activities.

**BC 5:** "Combustion" is used to describe anthropogenic emissions (as in this section) and wildfire emissions which is confusing and should be revised throughout. It is more typical to

refer to wildfire emissions as wildfire emissions, biomass burning emissions, or pyrogenic emissions. It is suggested to choose one of those more commonly used terms and use it throughout (right now there is some use of wildfire and biomass burning but it is inconsistent). Back to the line in the abstract, wildfire emissions don't "respond" to temperature in the same way that biogenic emissions or anthropogenic emissions do. There are complex relationships between ambient temperature and fuel moisture, and therefore fire ignitions, fire spread, fire, severity, fuel consumption, etc. that affect emissions. This complexity is not acknowledged in the manuscript (e.g., by saying that wildfire emissions "respond" to temperature) and the rationale for discussing the wildfire emissions in the context of temperature is not well supported.

**Response:** We agree with editor. The word combustion, all 24 of them, was replaced with wildfire to prevent confusion. Moreover, we concur with editor the word "respond" was unfit to describe the changes of the general physiochemical properties of the gases in the forest. The statement in abstract was modified into the following sentence:

Extreme heat and presence of wildfire plumes modified the overall volatility, reactivity, O:C, and H:C ratios of the extended list of VOCs.

**BC 6:** A related comment, on line 193, it is stated that acetonitrile and catechol don't follow the trend of temperature (presumably this means they do not increase with temperature like the BVOCs), which is not unexpected based the complexities noted above. However, in line 394 it is concluded that this is in fact because of the "infrequent emissions of BB plumes". It is not clear what is meant by that.

**Response:** The statement in question was indeed confusing, thus the 2nd part of the sentence was removed. The new statement is now:

Moreover, acetonitrile (r = 0.53) and catechol (r = 0.017) also did not follow the trend of temperature.

**BC 7:** As noted above, the discussion of wildfire emissions is not sufficiently differentiated from anthropogenic emissions throughout the manuscript. This leads to some confusion about the results and ultimately, the implications of this work.

**Response:** We concur with the editor. We modified several statements in the manuscript to differentiate anthropogenic emissions from wildfire activity.

The major source of benzene shifted from vehicular emissions to BB, highlighting the diverse activities influencing the variability of benzene at the temperate forest, as shown in the time series profile of benzene and Smoke

However, several prior studies have shown that monoterpene can also originate from anthropogenic sources and wildfire events

During the measurement period, the forest included several sources of biogenic compounds and was influenced by short- and long-range transport of anthropogenic and wildfire emissions

**BC 8:** In addition, "monoterpene" is used throughout and described on line 285 as being composed of several organic species. This is not accurate or consistent with existing literature. Monoterpene specifically describes a class of organic compounds with the formula C10H16, of which there are several isomers (some of which are listed on line 285). The analytical technique being used can't differentiate isomers (as noted in the methods), so what is being reported is the sum of monoterpenes (plural) and that needs to be clearer. The monoterpenes in that mixture can have very different reactivities, lifetimes, etc. and the manuscript should be revised with this in mind. For example, what was assumed for the OH reactivity of the monoterpene mixture?

**Response:** We agree with the editor. The current wording of monoterpene indicates a sum of all species instead of a class of compounds with molecular formula of  $C_{10}H_{16}$ . The new statement in the text are as follows:

Monoterpene, a critical contributor to ozone and secondary aerosol formation (Salvador et al., 2020a; Salvador et al., 2020b), is a class of organic compounds with the formula  $C_{10}H_{16}$  such as  $\alpha$ -pinene,  $\beta$ -pinene, limonene,  $\delta$ -carene, ocimene, and sabinene, and its distribution varies significantly based on the vegetation species.

The limitation of the PTR-ToF-MS to resolve isomers is included in the discussion of the capabilities of the instrument. The following statement is now in supplementary information:

One of the limitations of the PTR-ToF-MS technique is that it cannot distinguish isomers (e.g.,  $\alpha$ -pinene,  $\beta$ -pinene, and limonene) because of their identical exact mass (Blake et al., 2009).

Moreover, the reactivity value used for monoterpene was based on the median value of the rate constants available for all monoterpenes. This was highlighted in the new statements in the most recent version of the manuscript.

To address molecular formulas with multiple records, the median value of the rate constants was employed. For example, the reaction rate constant for the class of monoterpene was derived from literature values of several compounds, including  $\alpha$ -pinene,  $\beta$ -pinene, 3-carene, limonene, camphene, and  $\beta$ -ocimene. The median reaction constant across all these monoterpenes was then utilized to calculate the OH reactivity.

**BC 9:** The manuscript has a lot of detail that does not necessarily support or benefit from the measurements and results presented. For example, there is scattered mention of climate, climate forcing, future climate scenarios, and aerosols (including new particle formation). (One of the reviewers notes this of the introduction.) The connections between emissions composition, chemistry, aerosols, and climate could be made more generally, but the repeated mention does not strengthen the manuscript since the measurements and results are not clearly and sufficiently linked to these complex processes. Refining the focus to the measurements and results, including placing them in the context of similar studies, will help improve this aspect of the manuscript.

**Response:** Authors concur. Several statements from the original manuscript were removed or modified To shift the focus to the campaign and results instead of broad extended implication of the work. The following statements are either the modified or removed statements in the new version of the manuscript:

(Abstract – Modified) Ultimately, results here underscore the imminent effect of extreme heat and wildfire emissions on the overall chemical properties VOC in a temperate forest.

(Introduction – Removed) The results presented here provide important information to assess possible future feedback loops of vegetation and atmospheric chemistry to regional- and/or global-scale climate changes (Introduction).

(Results and Discussion – Removed) Based on the results here, isoprene at MOFLUX is expected to increase more as the temperature increases compared to monoterpene. Thus, careful consideration of the oxidant chemistry and product speciation will provide valuable new insights into the impact feedback loop between aerosols and climate in temperate forests.

(Results and Discussion – Removed) The interactions of AVOCs such as toluene, xylene, and trimethylbenzene produced more secondary organic aerosols, but the addition of BVOCs reduced the yield through cross-reactions between the intermediates (Chen et al., 2022).

(Results and Discussion – Removed) Given the anticipated increased temperatures in the future due to increasing effects of climate change, addressing the enhancements of formic and acetic acid will be important for predicting future chemistry–climate interactions.

(Summary and Conclusion – Removed) The comprehensive analysis of the whole mass spectra performed in this study underscores the importance of unaccounted-for VOCs in the atmosphere. The results of this study highlight the possible unaccounted modifications in VOC distribution that might be expected in future climate scenarios with serious impacts on aerosol–climate interaction. With the growing but still limited insights on the effect of mixed precursors on aerosol formation, more information regarding the overall distribution and transformation of AVOCs and BVOCs and their response to different future climate scenarios are needed to realistically account for the climate forcing of organic aerosols.

Header of section 4 was modified from Summary and Atmospheric Implications to **Summary and Conclusions.**

The current version of the manuscript compares results with previous studies with similar measurements. These are highlighted in the following statements:

Mean mixing ratios of methanol and acetone were 23 ppb, consistent with a prior study done in MOFLUX, in which half-hour averages of methanol ranged between 1.9 and 26 ppb (Seco et al., 2015)

Observed isoprene mixing ratios were substantially elevated compared to other similar temperate forests in the United Kingdom (~8 ppb) (Ferracci et al., 2020), deciduous forest in Michigan, USA (~1.5 ppb) (Kanawade et al., 2011), and mixed temperate forest in Canada (~0.01 ppb) (Fuentes and Wang, 1999). For MVK+MACr, prior measurements in similar environments reported mixing ratios below 2.0 ppb (Safronov et al., 2019; Shtabkin et al., 2019; Montzka et al., 1995) highlighting the intense production of MVK+MACr at MOFLUX.

Throughout the measurement duration, the maximum mixing ratio of monoterpene was 0.9 ppb. This ambient level is similar to a prior measurement at the MOFLUX site (Seco et al., 2015), as well as observations of monoterpene in other temperate forests in Wisconsin, USA, and Wakayama, Japan (Vermeuel et al., 2023; Ramasamy et al., 2016).

Furthermore, Figure 4 shows the evident exponential relationship between temperature and the major BVOCs, consistent with previous studies in which temperature controls the emission of isoprene and monoterpene (Hu et al., 2015; Selimovic et al., 2022; Guenther et al., 2012).

In MOFLUX, typical gas phase BB tracers were observed in substantial amounts. Acetonitrile, one of the prominent BB markers (Huangfu et al., 2021), had mean and maximum mixing ratios of 1.56 ppb and 4.45 ppb, respectively. Such values are beyond the mixing ratio range (0.047 to 1.08 ppb) of acetonitrile recorded in Asian, US, and European regions (Huangfu et al., 2021), implying the severe impact of BB.

During this period, the average OH reactivity was 100.53 ± 10.79 s-1, which was evidently higher compared to previous measurements in urban environment in California, USA (Hansen et al., 2021), sub-urban site in Shanghai, China (Yang et al., 2022), and forest environments in Finland (Sinha et al., 2010) and France (Bsaibes et al., 2020).

---

## Author Response (AR4)

**Measurement Report: Extreme Heat and Wildfire Emissions Enhance Volatile Organic Compounds in a Temperate Forest**

Christian Mark Salvador et al.

We appreciate the efforts of the handling editor in improving our manuscript, and we provided here a detailed response that addresses the handling editor's remaining minor concerns prior to publication. Our point-by-point responses to the Editor's comments are presented below. The Editor's comments are in black, and our answers are in red. Modified or new statements integrated into the revised manuscript are indented. All changes can be seen in the revised version of the manuscript in red font.

**Scientific Comment (SC)**

**SC 1:** Lines 198-208, calculation of OH reactivity, is somewhat confusing. One suggestion is to be consistent with language and use "reaction rate constant" throughout. Line 202...does this mean that reaction rate constants were calculated for all compounds with hourly resolution using measured temperature? Please just reread for clarity.

**Response:** The authors agree. The term reaction rate constant was used all throughout for clarity in the revised manuscript. The hourly ambient temperature were indeed considered during the calculation of the reaction rate constant. The calculation also incorporated available activation energy (Ea) and pre-exponential factor (A) data. The new paragraph now reads:

...where [VOCi] is the concentration of the volatile organic compounds measured by the PTR-ToF-MS and  $k_{VOC+OH}$  (cm3 molecule-1 s-1) are **the reaction rate constant** between the OH and VOC. The reaction rate constant were calculated based on the available data from the National Institute of Standards and Technology (NIST) Chemical Kinetics Database which compiled kinetics data on gas phase reactions (https://kinetics.nist.gov/kinetics/). All molecular formulas identified from more than 250 ions were subjected to Reaction Database Quick Search Form. The calculation of the reaction rate constant incorporated the hourly temperature conditions observed during the field campaign. Only records with available activation energy (Ea) and pre-exponential factor (A), along with temperature range (20-36 °C) similar to the observed conditions in the temperate forest, were considered in the calculation. To address molecular formulas with multiple records, the median value of the rate constants was employed. For example, the reaction rate constant for the class of monoterpene was derived from literature values of several compounds, including a-pinene,  $\beta$ -pinene, 3-carene, limonene, camphene, and  $\beta$ ocimene. The median reaction rate constant across all these monoterpenes was then utilized to calculate the OH reactivity.

**SC 2:** Lines 280-282: There are counter-examples of measured terpenes being highest at night, with clear daytime peaks. Bouvier-Brown et al., 2009; Pfannerstill et al., 2024

**Response:** The authors concur with the editor that the daytime peak can occur for some monoterpene. Thus, the statement was removed in the revised manuscript.

**SC 3:** Line 302: Do you mean that benzene can initiate new particle formation or just contribute to SOA formation? I am unfamiliar with the former (contribution of benzene to NPF).

**Response:** We refined this statement to specify benzene's role as a contributor to aerosol formation. The revised sentence now reads:

As benzene is a crucial precursor for ozone and a significant contributor to aerosol formation, the variability of such BB VOC should be incorporated into simulations of future atmospheric processes.

**SC 4:** Lines 350-354: Note that toluene also has a biogenic source, and appears as an emission in biogenic emissions models (e.g., MEGAN). This does not mean your conclusion is wrong, but you may want to rethink given that toluene can be biogenic.

**Response:** The authors acknowledge that toluene can have biogenic source. The paragraph includes the following statement:

...emission of toluene from biogenic activities.

**Response:**

**Editorial Comment (EC)**

**EC 1:** It is suggested to remove all exaggerated/non-quantitative/qualitative language. Some examples include: "grave implications" (line 49), "imminent implications" (line 359), remove "clearly" in Fig 1 caption.

**Response:** We agree with the editor. The following non-quantitative words were removed in the revised manuscript.

Ultimately, results here underscore the **imminent** effect of extreme heat and wildfire emissions ...(Line 33)

...which will have **grave** implications for air quality, climate, and human cardiovascular health (Line 49).

The alteration of VOC distribution due to enhanced temperature has **imminent** implications on the formation of secondary aerosols (Line 358)

The figure **clearly** illustrates the interstate and forested areas... (Line 111)

Overall, the calculated averages during extreme heat and wildfires **clearly** altered atmospheric reactivity in the forest (Line 450).

The highly variable profiles of the extended list of VOCs measured at MOFLUX **clearly** indicated that species were impacted by a variety of emissions and processes (Line 567)

**EC 2:** Line 66: suggestion to change "..ozone enhancement will lead to..." to "ozone enhancement may lead to..."

Response: Done. The new statement now reads:

Ozone enhancement may lead to elevated atmospheric oxidation capacity that can initiate more secondary pollutant formation.

EC 3: Line 88: "several VOCs" should be changed to "hundreds of VOCs"

**Response:** Done. The revised statement now reads:

During wildfire events, the burning of vegetation and other biomass induce the pyrolysis of plant materials which ultimately release **hundreds of VOCs** during the process

EC 4: Line 156: needs revision

**Response:** The authors agree. The following statement is included in the revised manuscript.

Backward airmass trajectories were estimated using the Hybrid Single-Particle Lagrangian Integrated Trajectory (HYSPLIT) model.

**EC 5:** Line 166: cabin or shed? (not clear what a cabin shed is)

Response: The term "shed" was removed from the statement

EC 6: Line 167: "can be" missing "found"

Response: Done. The line now reads:

A detailed description of the general mechanism of the PTR-ToF-MS **can be found** in the supplementary file and elsewhere

**EC 7:** Line 371-378: This section describes "sporadic...emission and transport" and "infrequent emissions". During a fire, emissions and transport will be ongoing (and not sporadic or infrequent). It is more than the variability in emissions and transport lead to infrequent interception of the smoke at any given location. Suggestion to reword to make that clearer.

**Response:** The authors agree with the editor. The terms sporadic and infrequent emissions were removed to make the statement clearer. The new statement now reads:

Acetonitrile did not exhibit a typical daily cycle, aligning with the unpredictable emissions and transport dynamics characteristic of biomass burning events.

EC 8: Line 439: "accounted for" should be changed to "attributed to"

**Response:** Done. The line now reads:

The elevated reactivity calculated in this study **was attributed to** the notable contributions from isoprene, acetone, ethylamine, and ethenone.

**EC 9:** Lines 549-554: This paragraph is talking about biomass burning/wildfire emissions, but concludes with talking about "such AVOCs". BB/wildfire emissions are not considered AVOCs. The specific compound, benzene, that you are describing has both pyrogenic and anthropogenic sources. Last sentence should be revised to make that clearer.

**Response:** We agree. The new statement now reads:

As benzene is a crucial precursor for ozone and a significant contributor to aerosol formation, the variability of such BB VOC should be incorporated into simulations of future atmospheric processes.